# The Shape of Attraction in UMAP: Exploring the Embedding Forces in Dimensionality Reduction

## Abstract

Uniform manifold approximation and projection (UMAP) is among the most popular neighbor embedding methods. The method samples pairs of point indices according to similarities in the high-dimensional space, and applies attractive and repulsive forces to their coordinates in the low-dimensional embedding. In this paper, we analyze the forces to reveal their effects on cluster formations and visualization, and compare UMAP to its contemporaries. Repulsion emphasizes differences, controlling cluster boundaries and inter-cluster distance. Attraction is more subtle, as attractive tension between points can manifest simultaneously as attraction and repulsion in the lower-dimensional mapping. This explains the need for learning rate annealing and motivates the different treatments between attractive and repulsive terms. Moreover, by modifying attraction, we improve the consistency of cluster formation under random initialization. Overall, our analysis makes UMAP and similar embedding methods more interpretable, more robust, and more accurate.

## 1 Introduction

Modern applications routinely generate high-dimensional data. Dimensionality reduction (DR) techniques have emerged as tools for exploratory analysis of such data by visualizing the underlying structure. The most popular methods, $t$-distributed stochastic neighbor embedding (Maaten & Hinton, 2008) and uniform manifold approximation and projection (UMAP) (McInnes et al., 2018) are grounded in the attraction-repulsion dynamics that bring similar data points closer while pushing dissimilar ones apart. As unsupervised algorithms, these do not rely on labeled data; instead, they identify and preserve the intrinsic structure of high-dimensional data by leveraging local (attractive) and global (repulsive) relationships (forces). This makes these algorithms particularly well-suited for tasks such as clustering (Becht et al., 2019), exploratory data analysis (Fleischer & Islam, 2020), anomaly detection in semiconductor manufacturing (Fan et al., 2021), visual search (González-Márquez et al., 2024), time series analysis (Altin & Cakir, 2024), studying representation convergence (Huh et al., 2024), and outlier image detection (Islam & Fleischer, 2024), where visualizing hidden patterns in unlabeled data is critical and meaningful. By learning the embeddings in a data-driven, label-free manner, DR exemplifies the power of unsupervised methods to distill complex data into easily interpretable forms.

Building upon the attraction-repulsion principle, newer methods have emerged (Amid & Warmuth, 2019; Agrawal et al., 2021; Wang et al., 2021; Narayan et al., 2021; Yang et al., 2022; Wang et al., 2025; Kury et al., 2025), each designed to emphasize specific aspects of the data. Despite their relevance in diverse applications, these methods often rely on heuristic practices that may fail to give meaningful interpretations. Moreover, DR introduces distortions that are unavoidable (Chari & Pachter, 2023). Thus, it is imperative to have a deeper understanding of the algorithms so that practitioners can provide better interpretations of the embeddings, avoid spurious structures, and optimize performance. In practice, these algorithms achieve compact clusters using a variety of techniques, including specific initialization, learning rate schedule, and kernel function tuning. However, the underlying dynamics of the attractive and repulsive forces, responsible for cluster formation, have not been thoroughly investigated. Furthermore, the essential tunable parameters are concealed within abstract functional forms, making it harder to explain the algorithms.

In this paper, we decompose the forces into their constituent parts and extract the functional shapes of attraction and repulsion. We find that the necessity of learning rate annealing, the challenge of providing consistent output under random initialization, and the origin of cluster formation rely on attraction. Repulsive forces primarily govern inter-cluster distances. Our specific contributions are:

1. We formulate attraction and repulsion shapes from the attractive and repulsive forces, establish the conditions for contraction and expansion of distance, and provide a fresh perspective for these algorithms (Section 4).
2. We show that the attraction shape of UMAP causes the counterintuitive concept of both contraction and expansion of distance. Comparing attraction shapes of different algorithms, we discuss how attraction influences the learning rate annealing scheme (Section 5).
3. We modify the attraction shape to improve the consistency of embedding under random initialization. This indicates the encoding of a unique structure (Section 5.1).
4. Analyzing repulsion shapes, we provide a deeper understanding of cluster formation and regulating inter-cluster distance (Section 5.2).
5. We compare attraction and repulsion shapes of UMAP, Parametric UMAP, and NEG-$t$-SNE, unveil the similarities and distinctions among them, and characterize the stability of the algorithms (Section 6).

We center the main text on UMAP, now a de facto standard in many fields, and bring in other algorithms where relevant. For further details on different algorithms, see the appendix.

## 2 RELATED WORKS

The origin of modern iterative graph-based neighbor embedding algorithms can be traced to stochastic neighbor embedding (SNE) (Hinton & Roweis, 2002) and its extension using the t-distribution ($t$-SNE) (Maaten & Hinton, 2008). Both methods use a dense graph in which each point in a dataset has a pairwise relation with all the others, regardless of whether they are similar to each other or not. Moreover, the weights of the graphs are normalized to give a notion of probability distribution. Other concurrent methods, including locally linear embedding (Roweis & Saul, 2000) and Laplacian Eigenmaps (LE) (Belkin & Niyogi, 2002), used a $k$-nearest neighbor ($k$-NN) graph of pairwise interaction. Known as spectral methods, these algorithms rely on Eigenvalue decomposition. Subsequent work by Tang et al. (Tang et al., 2016) incorporated the $k$-NN graph in the iterative approach and removed normalization in the lower dimension. This approach was further extended by McInnes et al. (2018) in UMAP, where the normalization step was removed altogether (both in high and low dimension) and the embedding was obtained using pairwise interactions alone. The optimization steps use an explicit attractive force to preserve the local neighborhood and a repulsive force to keep dissimilar points apart. Building on these foundations, methods such as PaCMAP (Wang et al., 2021) and NEG-$t$-SNE (Damrich et al., 2023) have been proposed. For a recent survey of methods, see de Bodt et al. (2025).

There has been considerable progress in understanding and explaining the relationship among these algorithms. An early analysis of SNE found that if the data is well-clustered in the original space, then they are well-clustered in the embedding space (Shaham & Steinerberger, 2017). A similar analysis for $t$-SNE by Linderman & Steinerberger (2019) showed that the number of clusters in the embedding space is a lower bound on the number of clusters in the original space. This was followed up in further characterization (Arora et al., 2018; Cai & Ma, 2022; Linderman & Steinerberger, 2022). Since $t$-SNE and UMAP originate from the same underlying framework (but with drastically different visualizations), a major undertaking in the literature has been to find the connection between them (Böhm et al., 2022; Damrich et al., 2023; Draganov et al., 2023). Böhm et al. (2022) theorized that methods like Laplacian Eigenmap, ForceAtlas2 (Jacomy et al., 2014), UMAP, and $t$-SNE are all samples from the same underlying spectrum. Indeed, LE and $t$-SNE's are connected by the early exaggeration phase (Cai & Ma, 2022). The connection between UMAP and $t$-SNE can be related through contrastive estimation (Damrich et al., 2023). Around the same time, Hu et al. (2023) independently discovered the relation of contrastive learning and SNE. Recent approaches offer a probabilistic perspective (Ravuri et al., 2023; Ravuri & Lawrence, 2024), employ kernel techniques (Draganov & Dohn, 2023), and utilize information geometry (Kolpakov & Rocke, 2024) to explain dimensionality reduction.

## 3 UNIFORM MANIFOLD APPROXIMATION AND PROJECTION

UMAP constructs a high-dimensional graph of the dataset $X = \{x_i \in R^n | i = 1, \ldots, N\}$ by using a pairwise relation: $p_{i,j} = f_h(d(x_i, x_j))$, typically, $\in [0, 1]$, **where, $f_h$ is the high dimensional affinity function and $d(\cdot, \cdot)$ is a distance metric** (for details, see Appendix H).

The graph of the low-dimensional representation $Y = \{y_i \in \mathbb{R}^d | i = 1, \ldots, N\}$ is given by a differentiable function

$$q_{ij} = \frac{1}{1 + a(||y_i - y_j||_2^2)^b}, \tag{1}$$

where the parameters $a$ and $b$ determine the density of the mapping and are chosen by fitting $q_{ij}$ to

$$\Psi(||y_i - y_j||_2) = \begin{cases} 1 & \text{if } ||y_i - y_j||_2 < m_d \\ \exp(-(||y_i - y_j||_2 - m_d)) & \text{otherwise} \end{cases}, \tag{2}$$

where $m_d$ regulates the distance between the two nearest low-dimensional points.
UMAP aims to minimize the following cross-entropy loss function:

$$\mathcal{L} = \sum_{i,j} (-p_{ij} \log(q_{ij}) - (1 - p_{ij}) \log(1 - q_{ij})). \tag{3}$$

The first term provides an attractive force and the second term provides a repulsive force. Instead of optimizing every point in each iteration, UMAP takes the negative sampling approach (Mikolov et al., 2013; Tang et al., 2016). For each edge with $p_{ij} > 0$, named a positive edge, several edges are sampled randomly, named negative edges. The attractive force is applied on the positive edge:

$$y_i^{t+1} = y_i^t + \lambda \nabla_{y_i^t} \log(q_{ij}), \tag{4}$$

$$y_j^{t+1} = y_j^t + \lambda \nabla_{y_j^t} \log(q_{ij}), \tag{5}$$

and the repulsive force is applied on the negative edges:

$$y_i^{t+1} = y_i^t + \lambda \nabla_{y_i^t} \log(1 - q_{ij}), \tag{6}$$

where $\lambda (> 0)$ is the learning rate and $t$ is the step number. Note that $y_j$ is not updated for negative edges. For a detailed analysis of UMAP's loss, see Damrich & Hamprecht (2021).

## 4 ATTRACTION AND REPULSION SHAPES

The action of the updates (4-6) can be simplified by decomposing the gradients $\nabla_{y_i^t} \log(q_{ij})$ ($\nabla_{y_i^t} \log(1 - q_{ij})$) into a scalar coefficient dependent on the distance $||y_i - y_j||_2$ acting on the vector $(y_i - y_j)$. We call this scalar coefficient the attraction (repulsion) shape. While we use UMAP as a specific example, this formalism applies to any method that relies on attraction and repulsion.

By writing $\nabla_{y_i^t} \log(q_{ij}) = f_a(\zeta^t)(y_i^t - y_j^t)$, where $\zeta = ||y_i - y_j||_2$, we can update the equations of a positive edge as

$$y_i^{t+1} = y_i^t + \lambda f_a(\zeta^t)(y_i^t - y_j^t), \tag{7}$$

$$y_j^{t+1} = y_j^t - \lambda f_a(\zeta^t)(y_i^t - y_j^t). \tag{8}$$

Here, $f_a : \mathbb{R}_{\geq 0} \to \mathbb{R}_{\leq 0}$ is the attraction shape, and we use the fact that for Euclidean metric, $\nabla_{y_i^t} \log(q_{ij}) = -\nabla_{y_j^t} \log(q_{ij})$. Similarly, we reformulate the update equation of a negative edge as

$$y_i^{t+1} = y_i^t + \lambda f_r(\zeta^t)(y_i^t - y_j^t), \tag{9}$$

$$y_j^{t+1} = y_j^t, \tag{10}$$

where $f_r : \mathbb{R}_{\geq 0} \to \mathbb{R}_{\geq 0}$ is the repulsion shape.

Such decomposition has appeared previously, e.g., in (McInnes et al., 2018; Agrawal et al., 2021; Draganov & Dohn, 2023), but their formulations and utilization vary. The original UMAP paper (McInnes et al., 2018) used it as a computational trick for fast processing, while Draganov

& Dohn (2023) used it for comparing different algorithms from the kernel perspective. In both cases, the primary focus was computing the derivative. Here, we treat the decompositions as independent functions that can take various forms. A similar approach was used earlier by Agrawal et al. (2021), but they expressed the decomposition in terms of a scalar coefficient and a unit vector $((y_i - y_j)/||y_i - y_j||_2)$ to emphasize the magnitude of the forces. The discussion below shows that our shape decomposition is more illuminating than the magnitude alone.

### 4.1 CONDITIONS FOR ATTRACTION AND REPULSION

In this section, we establish the conditions of attraction and repulsion from the update equations (7)-(10). The following proposition characterizes the contraction of distance between the pair $y_i^t$ and $y_j^t$ of a positive edge:

**Proposition 4.1.** *The update Eqs. (7) and (8) provide a contraction of distance ($||y_i^{t+1} - y_j^{t+1}|| < ||y_i^t - y_j^t||$) if $-1 < \lambda f_a < 0$.*

Here, $\lambda f_a$ works as the effective attraction shape. In most cases, however, $f_a$ alone is enough to draw meaningful conclusions about the embeddings. For a negative edge, the following proposition characterizes the expansion of the distance between the pair $y_i^t$ and $y_j^t$:

**Proposition 4.2.** *The update Eqs. (9) and (10) provide an expansion of distance ($||y_i^{t+1} - y_j^{t+1}|| > ||y_i^t - y_j^t||$) if $f_r > 0$.*

Note that the inclusion of a symmetric term $-\lambda f_r(\zeta)(y_i^t - y_j^t)$ in Eq. (10) does not alter the conclusion presented in this proposition. Additionally, these conditions give a per-iteration certificate for guaranteed contraction/expansion, and one should not confuse them with the learning rate tuning/decay mechanism, as we can design the shapes in such a way that the contraction and expansion do not rely on learning rate decay.

Figure 1 (a) shows the effect of different values of $f_a < 0$ and $f_r > 0$ on two points. The latter shape is straightforward, as any positive value increases the distance. The former is much more subtle, as $f_a$ encodes both attractive and repulsive dynamics. For $f_a \in (-1.0, 0)$, the distance decreases, with a sign flip at the value $f_a = -0.5$ of maximum attraction (coincident points). Any value lower than $-1.0$ causes the distance to increase. Although these forces act locally, they collectively shape the global structure.

## 5 ANALYSIS OF UMAP IN TERMS OF ATTRACTION AND REPULSION SHAPE

Using the gradient decomposition and the distance form (1), the attraction and repulsion shapes are given by

$$f_a^U(\zeta) = -\frac{2ab\zeta^{2(b-1)}}{1 + a\zeta^{2b}} \tag{11}$$

and

$$f_r^U(\zeta) = \frac{2b}{\zeta^{2b}(1 + a\zeta^{2b})}, \tag{12}$$

respectively. This section focuses on the default shapes of UMAP, controlled primarily by the parameters $a$ and $b$, and discusses the insights learned by perturbing them. The discussion below is generally valid for $b \leq 1$, where the attraction shape is strictly increasing and thus invertible (For derivation, see Appendix A).

Figure 1 (b) shows the default attraction shape of UMAP ($a = 1.58, b = 0.89$). It becomes unbounded (approaches $-\infty$) as $\zeta \to 0$. As predicted by Proposition 4.1, the transition from contraction to expansion occurs when $\lambda f_a^U$ crosses $-1$ as $\zeta$ approaches 0. Since the attraction shape is invertible, we can identify the distance at this transition, $\zeta_{-1} = (\lambda f_a^U)^{-1}(-1)$, as the minimum distance for contraction due to attractive updates. Effectively, $\zeta > \zeta_{-1}$ causes contraction, and $0 < \zeta < \zeta_{-1}$ causes expansion, contradicting the intuition that attractive updates consistently bring points closer together.

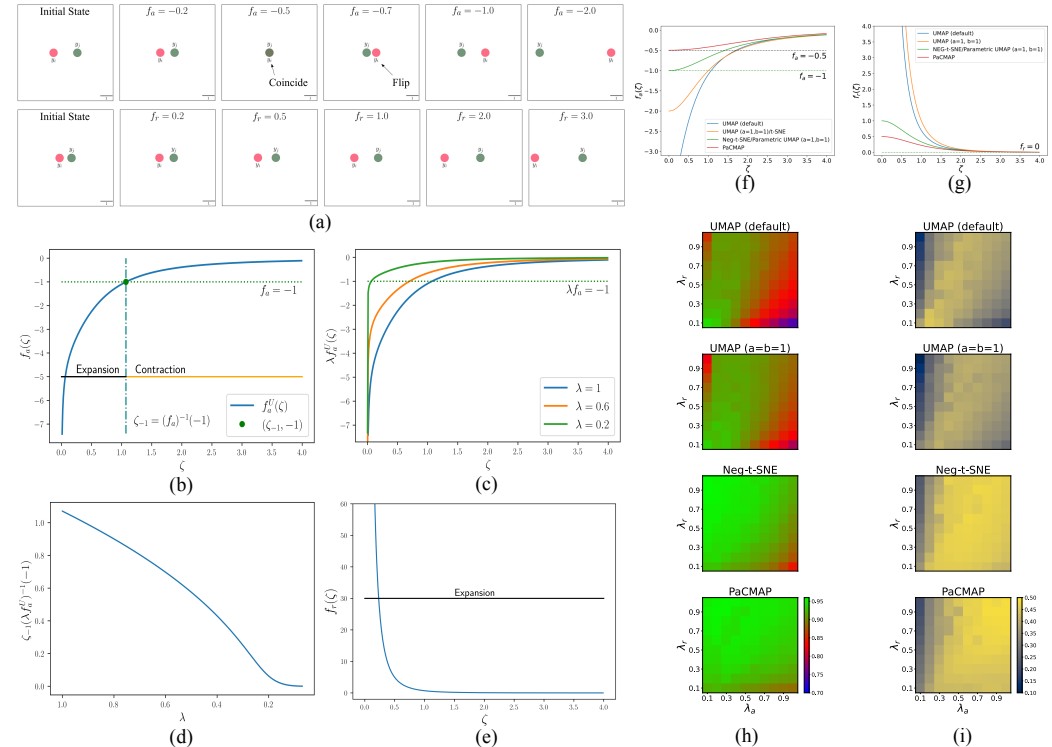

Figure 1: Attraction and repulsion shapes in UMAP. (a) Effect of different values of $f_a$ (top) and $f_r$ (bottom) on a pair. (b) Attraction shape of UMAP. (c) Effective attraction shape ($\lambda f_a$) for various learning rates $\lambda$. (d) Minimum distance for contraction ($\zeta_{-1}$) as $\lambda$ decreases. (e) Repulsion shape of UMAP. (f) Attraction and (g) repulsion shapes of various embedding methods. (h) Trustworthiness and (i) Silhouette Score of various methods as the constant learning rate is varied for attraction ($\lambda_a$) and repulsion ($\lambda_r$) independently. Default UMAP parameters: $a = 1.58$ and $b = 0.89$.

If $\zeta_{-1}$ is high, neighboring points oscillate between contraction and expansion, and the clusters appear fuzzy. For the sharpest boundaries, then, the goal of optimization can be recast as one of achieving the limit $\zeta_{-1} \to 0$. **The default shape oscillates around $\zeta = 1.07$, which is large compared to the expectation.** As a result, UMAP's learning rate schedule requires annealing to zero (Figs. 1 (c,d)). On the other hand, the repulsion shape (Eq. 12) is always positive and satisfies Proposition 4.2. $f_r^U$ approaches 0 as $\zeta \to \infty$ and approaches $\infty$ as $\zeta \to 0$ (Fig. 1 (e)).

Attraction shapes for different algorithms show that only UMAP and t-SNE deal with the issue of having $f_a < -1$ (and consequently $\zeta_{-1} > 0$ at $\lambda = 1$, Fig. 1(f)). t-SNE solves it by weighting the updates with corresponding $p_{ij}$ values, while UMAP relies on learning rate annealing. Methods like Neg-t-SNE and PaCMAP have $f_a$ naturally within $[-1, 0]$ (and thus, satisfies Proposition 4.1 for $\lambda \in [0, 1]$ with $\zeta_{-1} = 0$). Furthermore, PaCMAP's weighted $f_a$ ($< 0.5$ always) even prevents any flips during attraction. On the other hand, the repulsion shapes for different algorithms show that only UMAP deals with large values for small distances (Fig. 1(g); the shape is unbounded, but during optimization, the values are often clipped).

We show the effect of these attraction/repulsion choices for MNIST embedding by applying separate constant learning for attraction ($\lambda_a$) and repulsion ($\lambda_r$), and varying their values (Figs. 1(h,i)). For UMAP, the lower value of $\lambda_a$ ($< 0.5$ and consequently $\zeta_{-1} \simeq 0$) is preferred for better embedding (trustworthiness (T) (Venna & Kaski, 2001), a measure of structure preservation, in Fig. 1(h)) and clustering (silhouette scores (SIL) (Rousseeuw, 1987), a measure of cluster separation, in Fig. 1(i)), whereas the whole range of $\lambda_r$ ($\in [0, 1]$) could be used. For NEG-t-SNE and PaCMAP, for which $\zeta_{-1} = 0$, the whole range of parameters is effective. **This confirms that UMAP relies on making $\zeta_{-1}$ close to 0 for satisfactory embedding and clustering, which it achieves in practice by learning rate annealing, whereas $\zeta_{-1} = 0$ is the default in newer algorithms.**

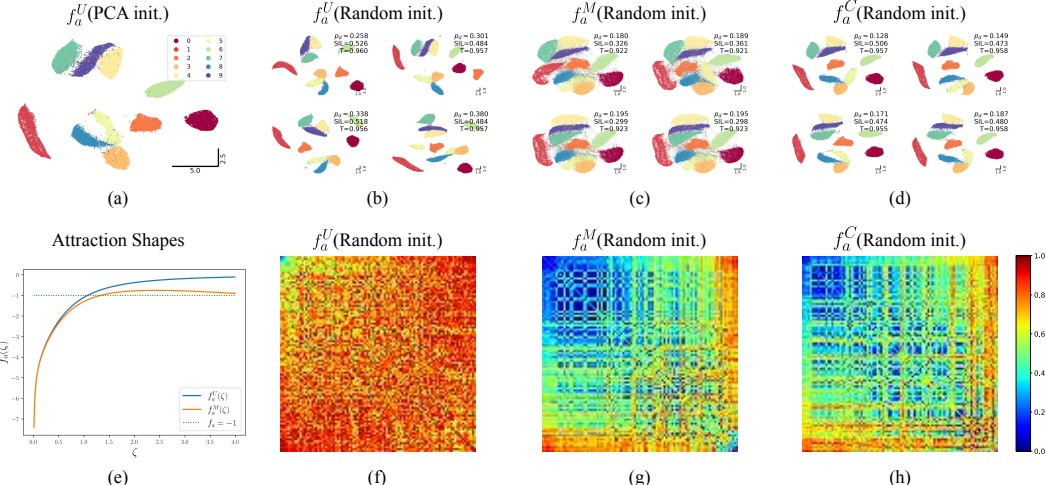

Figure 2: Effect of random initialization on different attraction shapes for the MNIST dataset. (a) Mapping using PCA. (b-d) Four mappings with the lowest Procrustes distance ($p_d$) from the embedding in (a) for (b) UMAP, (c) modified, and (d) composite attraction shapes. (e) Default UMAP and modified attraction shapes. (f-h) Procrustes matrices from 100 runs of (f) UMAP ($0.78 \pm 0.13$), (g) modified ($0.49 \pm 0.21$), and (h) composite attraction ($0.50 \pm 0.20$) shapes. The diagonal $(i, i)$ entries of the Procrustes matrix are sorted by Procrustes distance ($p_d$) from (a), and the off-diagonal, $(i, j)$, entries correspond to $p_d$ between $i^{th}$ and $j^{th}$ mapping. The matrices and (mean $p_d \pm$std) values show that UMAP's embeddings are not self-similar, while the modified and composite attraction shapes encourage initialization-invariant structure.

Future sections and appendices provide additional details. In Section 6 we focus on the case of $a = 1$ and $b = 1$ and compare UMAP to NEG-t-SNE. Appendix B provides additional discussion regarding constant learning rate in UMAP. Appendix G provides details of other dimensionality reduction algorithms (TriMAp, PaCMAP, LocalMAP, t-SNE, SNE and multidimensional scaling) from the perspective of attraction and repulsion shape. Appendix I gives details of the embeddings used in this section as weel as extends results for two other datasets.

## 5.1 Improving UMAP's Consistency under Random Initialization

The consistency of UMAP embeddings depends on proper initialization (Kobak & Linderman, 2021; Wang et al., 2021). Typically, principal component analysis (PCA) of the data or spectral decomposition of the high-dimensional graph initializes the embedding, producing consistent mappings despite various sources of stochasticity. If randomly initialized, clusters often fail to form or form in a random orientation each time the algorithm executes. If the initial distance between two points (nearest neighbors in high dimension) is large, the attractive forces become too low to bring them closer. Known as near-sightedness (Wang et al., 2021), this phenomenon is evident in the attraction shape, where $|f_a^U|$ diminishes towards zero as the distance increases (**since,** $|f_a^U| = o(1/\zeta)$**, and thus,** $\lim_{\zeta \to \infty} |\tilde{f}_a^U(\zeta)|\zeta = 0$).

One can induce "far-sightedness" in the mapping by increasing attraction for large distances, facilitating faraway neighbors to come closer. To test this hypothesis, we modify the attraction shape of UMAP to increase the attractive force:

$$f_a^M = f_a^U - \beta\zeta, \tag{13}$$

where $\beta$ is a parameter that regulates the strength of the added term (we used $\beta = 0.2$, Fig. 2 (e)). This addition in the attraction shape translates to adding a regularizer in the attractive term of the loss function (i.e., $\mathcal{L}^M = \mathcal{L} + \sum_{i,j} \beta/3 \, ||y_i - y_j||_2^3$). In addition to attracting pairs at faraway distances, this technique enables intermixing of points that help convergence under random initialization (akin to early exaggeration in $t$-SNE). In (13), we chose the simplest linear correction; other functions, such as $\log \zeta$ or $\zeta^p$ ($p \in \mathbb{R}_{\geq 0}$), may also be suitable.

Table 1: Effect of random initialization on different datasets using different attraction shapes and comparison to PCA initialization. The metrics (mean±std) are reported based on 100 runs of each. Datasets: MNIST (LeCun et al., 2010), FMNIST (Xiao et al., 2017), Transcriptomes- (Macosko et al., 2015), (Shekhar et al., 2016), and 20NewsGroup (20NG) (Mitchell, 1997)

| Dataset | Shape (initialization) | Embedding Quality | | | Run to Run Consistency | |
|---|---|---|---|---|---|---|
| | | Trustworthiness | Silhouette Score | Spearman | Spearman | Procrustes Distance |
| MNIST | *Default (PCA, standard)* | *0.957 ± 0.001* | *0.51 ± 0.01* | *0.31 ± 0.01* | *0.95 ± 0.06* | *0.13 ± 0.05* |
| | Default (Rand) | 0.958 ± 0.001 | 0.47 ± 0.04 | 0.24 ± 0.06 | 0.44 ± 0.12 | 0.78 ± 0.13 |
| | Modified (Rand) | 0.923 ± 0.003 | 0.33 ± 0.03 | 0.33 ± 0.03 | 0.71 ± 0.12 | 0.49 ± 0.21 |
| | Composite (Rand) | 0.956 ± 0.001 | 0.48 ± 0.02 | 0.29 ± 0.04 | 0.70 ± 0.11 | 0.50 ± 0.21 |
| FMNIST | *Default (PCA, standard)* | *0.975 ± 0.001* | *0.18 ± 0.00* | *0.60 ± 0.00* | *0.99 ± 0.00* | *0.02 ± 0.00* |
| | Default (Rand) | 0.976 ± 0.001 | 0.11 ± 0.05 | 0.42 ± 0.07 | 0.54 ± 0.13 | 0.72 ± 0.15 |
| | Modified (Rand) | 0.959 ± 0.004 | 0.16 ± 0.03 | 0.57 ± 0.04 | 0.82 ± 0.11 | 0.38 ± 0.19 |
| | Composite (Rand) | 0.975 ± 0.001 | 0.18 ± 0.03 | 0.53 ± 0.05 | 0.78 ± 0.10 | 0.43 ± 0.18 |
| Macosko | *Default (PCA, standard)* | *0.950 ± 0.001* | *0.42 ± 0.04* | *0.77 ± 0.01* | *0.95 ± 0.02* | *0.16 ± 0.07* |
| | Default (Rand) | 0.950 ± 0.001 | 0.23 ± 0.06 | 0.75 ± 0.02 | 0.76 ± 0.03 | 0.91 ± 0.06 |
| | Modified (Rand) | 0.935 ± 0.003 | 0.15 ± 0.05 | 0.71 ± 0.02 | 0.82 ± 0.05 | 0.61 ± 0.13 |
| | Composite (Rand) | 0.949 ± 0.001 | 0.28 ± 0.05 | 0.73 ± 0.02 | 0.83 ± 0.04 | 0.64 ± 0.15 |
| Shekhar | *Default (PCA, standard)* | *0.974 ± 0.000* | *0.52 ± 0.01* | *0.51 ± 0.01* | *0.94 ± 0.03* | *0.07 ± 0.03* |
| | Default (Rand) | 0.974 ± 0.000 | 0.41 ± 0.08 | 0.48 ± 0.03 | 0.52 ± 0.09 | 0.70 ± 0.16 |
| | Modified (Rand) | 0.965 ± 0.001 | 0.58 ± 0.03 | 0.40 ± 0.02 | 0.81 ± 0.06 | 0.48 ± 0.19 |
| | Composite (Rand) | 0.973 ± 0.000 | 0.51 ± 0.03 | 0.48 ± 0.01 | 0.73 ± 0.08 | 0.51 ± 0.19 |
| 20NG | *Default (PCA, standard)* | *0.819 ± 0.001* | *−0.17 ± 0.00* | *0.58 ± 0.00* | *0.96 ± 0.01* | *0.05 ± 0.01* |
| | Default (Rand) | 0.818 ± 0.002 | −0.18 ± 0.01 | 0.57 ± 0.01 | 0.92 ± 0.03 | 0.24 ± 0.13 |
| | Modified (Rand) | 0.777 ± 0.001 | −0.16 ± 0.00 | 0.59 ± 0.00 | 0.95 ± 0.04 | 0.20 ± 0.13 |
| | Composite (Rand) | 0.817 ± 0.002 | −0.18 ± 0.01 | 0.58 ± 0.01 | 0.94 ± 0.04 | 0.19 ± 0.15 |

We also consider a composite attraction shape:

$$f_a^C = \begin{cases} f_a^M, & \text{epoch} \leq 100 \\ f_a^U, & \text{otherwise} \end{cases}. \tag{14}$$

The composite shape attempts to remove any distortions introduced by $f_a^M$ by reverting to the original UMAP. Below, we discuss the effects these modified and composite attraction shapes have on DR from random initialization.

We first created a PCA-initialized embedding of the MNIST dataset Figure 2 (a). Then, we produced embeddings using random initialization (Gaussian) for each shape and repeated the experiment 100 times. To quantify the results, we use Procrustes analysis (Gower, 1975) that aligns two point clouds under scaling, translation, rotation, and reflection (for details, see Appendix D.1). Here, we align the randomly initialized embeddings to that of the PCA-initialized one and characterize their separation using the Procrustes distance ($p_d$). Figure 2 (b) shows four embeddings with the lowest $p_d$. While the cluster shapes are consistent, their placements are not. Outputs from the modified and composite attraction shapes (Figs. 2 (c) and (d), respectively) show improved consistency of cluster placements.

To quantify the placements further, we consider the Procrustes matrix: the diagonal of the matrix is sorted by $p_d$ from the PCA-initialized mapping, and the off-diagonal values are $p_d$ between two randomly initialized mappings (for details, see Appendix D.1). This quantification is analogous to the similarity matrix (Foote, 1999). The embeddings due to the default UMAP attraction shape are not similar to each other (Fig. 2 (f)), but the modified (Fig. 2 (g)) and composite (Fig. 2 (h)) shapes show strong similarity to each other.

**Table 1 shows embedding quality and run-to-run variance of five different datasets when randomly initialized and compare it to the corresponding PCA initialized version (standard and the most consistent one). Overall, we can observe that the embedding quality is comparable among different shapes. Overall, modified and composite shapes improve consistency in terms of Spearman's rank correlation and Procrustes distance. This indicates that UMAP, regardless of the initialization, aims to encode a unique structure (in our experiments, PCA initialized embedding is an attractor).** However, attaining that structure in the low dimension may fall short due to small attraction at longer distances.

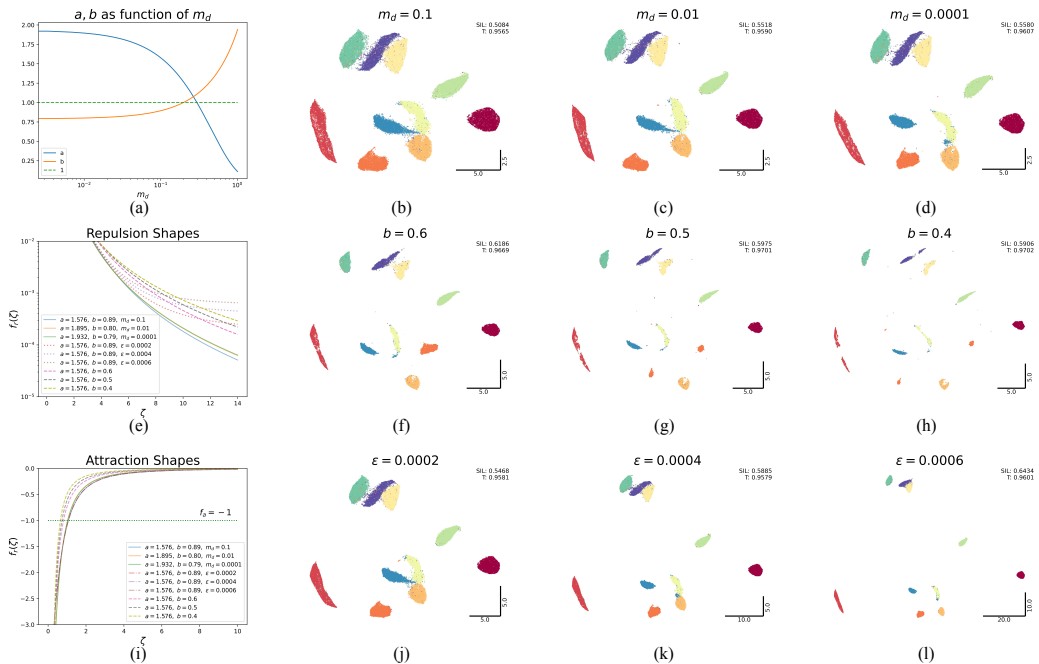

Figure 3: Control of inter-cluster distances on the MNIST dataset. (a) Computing $a, b$ by varying the low-dimensional distance $m_d$ restricts exploration. (b-d) UMAP output by setting $m_d$ to 0.1, 0.01, and 0.0001, respectively, shows little improvement in compactness of clusters. (e) Repulsion shapes for different parameters. (f-h) Increasing repulsion by explicitly varying $b$ results in more compact clusters and forms new ones that were absent otherwise. (i) Attraction shapes by varying parameters. (j-l) Increasing repulsion by adding a small positive value ($\varepsilon$) to the repulsion shape increases inter-cluster distance.

## 5.2 CLUSTER FORMATION AND COMPACTNESS

The primary controllable parameter influencing cluster formation in UMAP is the minimum distance parameter $m_d$ (through Eq. 2). However, varying $m_d$ restricts the exploration of different values of $a$ and $b$ (Fig. 3 (a)). Thus, reducing $m_d$ often results in embeddings that do not provide additional benefit (Figs. 3 (b-d)). The key factor is the limited influence of varying $m_d$ on the repulsion shape (Fig. 3 (e)). Alternatively, we can explicitly vary the values of $a$ and $b$. Decreasing $a$ increases repulsion, but it decreases attraction at a faster rate (causing a worse case of near-sightedness). On the other hand, decreasing $b$ gives a better control (Figs. 3 (e,i)). Figures 3 (f-h) show increasing inter-cluster distance and breaking up of previous clusters by varying $b$ to 0.6, 0.5, and 0.4, respectively. This breaking up occurs due to the increasingly heavy-tailed nature of the kernel as $b$ decreases (heavy-tailed kernels result in smaller and distinct clusters (Van der Maaten & Hinton, 2008; Kobak et al., 2019); see Kobak et al. (2019) for how varying $b$ in UMAP modulates the tail). Although this approach separates all ten MNIST labels into distinct clusters, the relative contributions of attraction and repulsion are hard to disentangle; changing $b$ amplifies repulsion more than changing $m_d$, but it also reshapes the attraction profile.

To quantify the effect of repulsion independently, we can keep attraction fixed and vary the other. To achieve this, we modify the repulsion shape by adding a small positive value ($\varepsilon$):

$$f_r^M = f_r^U + \varepsilon, \tag{15}$$

while keeping the values of $a$ and $b$ constant, which effectively adds a regularizer in the repulsive term of the loss function (i.e, $\mathcal{L}^M = \mathcal{L} - \sum_{i,j} \varepsilon/2 \, \|y_i - y_j\|_2^2$). Using this, we can obtain stronger repulsion than previously (Fig. 3(e)). Figs. 3 (j-l) show that as $\varepsilon$ increases, the inter-cluster distances also increase. However, the clustering properties show similarity to those obtained by varying $m_d$, and we get a loose separation of all the labels as $\epsilon$ increases. Overall, the parameter $\varepsilon$ keeps the

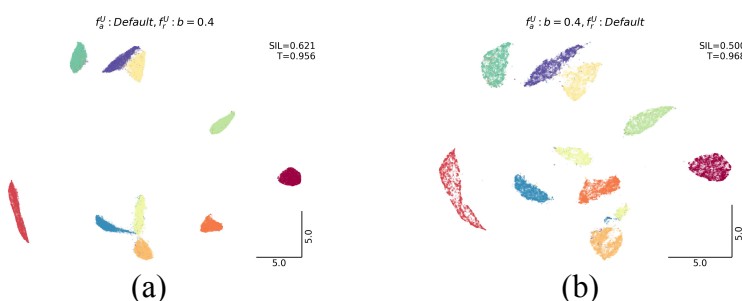

Figure 4: Embedding of the MNIST dataset with (a) default attraction shape but repulsion shape with $b = 0.4$ and (b) default repulsion shape but attraction shape with $b = 0.4$. The former shows the same clusters of default UMAP with increased compactness. The latter develops new structures within clusters and forms new clusters.

attraction shape unaffected, and varying $m_d$ effectively traces similar attraction shapes (Fig. 3 (i)), suggesting cluster formation is governed predominantly by attraction.

To resolve the mystery of cluster formation, we change either the attraction shape or the repulsion shape individually while leaving the other at the default by simply setting $b$ to $0.4$ (Fig. 4). The default attraction is unable to show new structures or clusters in the embedding, but the increased repulsion ($b = 0.4$) gives smaller clusters than the original UMAP (Fig. 4 (a)). On the other hand, when the attraction increases by setting $b = 0.4$ with the default repulsion, the embedding shows additional structures within each cluster (Fig. 4 (b)). Some of the older clusters even separate into smaller ones. This shows that attraction causes cluster formation, while repulsion makes the clusters more compact (depending on the repulsion shape, it does this either by making smaller clusters or increasing inter-cluster distance). **By controlling attraction and repulsion behavior explicitly, we can create embeddings with various levels of granularity.**

**As an aside, one of the claims of LocalMAP is to separate the ten labels of MNIST into individual clusters. Here, we have achieved the same result in UMAP by simply manipulating the attraction and the repulsion shapes. In Appendix G.3, we explore how LocalMAP embeddings exploit this interplay of attraction and repulsion to separate the clusters and relate it to the experiments of this section.**

## 6 COMPARISON TO NEG-$t$-SNE

Setting $a$ and $b$ to $1$ is common for various dimensionality reduction algorithms. This particular setting makes the gradients in many algorithms stable (or bounded) and, in turn, makes optimization easier. Recently, Damrich et al. (2023) explored this for UMAP and proposed Neg-$t$-SNE as a solution, which, in addition to having a stable gradient, provides a more compact clustering even for a constant learning rate of $1$. On the other hand, Parametric UMAP (Sainburg et al., 2021) initially used the UMAP loss formulation, but later it[1] adopted a numerically stable modified cross-entropy loss function (Shi et al., 2023) with a logsigmoid kernel. Analysis of both reveals that the approaches arrive at the same formulation from different starting points, which leads to the following observation:

**Proposition 6.1** (Damrich et al. (2023)). *Neg-t-SNE is Parametric UMAP with $a = 1$ and $b = 1$.*

With $a = 1$ and $b = 1$, the attraction and repulsion shapes of UMAP are given by $f_a^U = -2/(1 + \zeta^2)$ and $f_r^U = 2/(\zeta^2(1 + \zeta^2))$, respectively. The attraction shape becomes bounded within $[-2, 0]$, with $f_a^U(0) = -2$ (Fig. 1 (f)), while the repulsion shape remains essentially unchanged (i.e., unbounded as $\zeta \to 0$, Fig. 1 (g)). Since $f_a^U < -1$ as $\zeta \to 0$, according to Proposition 4.1 and the discussion provided in Section 5, this unity case still requires learning rate annealing.

Damrich et al. (2023) compared UMAP's negative sampling loss function from the perspective of contrastive embedding (CE) and concluded that the effective kernel of UMAP is $1/\zeta^2$. Under

---

[1]https://github.com/lmcinnes/umap/pull/856, merged on April 26, 2022

the CE framework, the authors introduced NEG-$t$-SNE by changing the kernel to $1/(1 + \zeta^2)$. Reverting to UMAP formalism, this results in the low-dimensional affinity function $q_{ij}^N = 1/(2 + \zeta^2)$. Consequently, the attraction and repulsion shapes are $f_a^N = -2/(2 + \zeta^2)$, and $f_r^N = 2/((1 + \zeta^2)(2 + \zeta^2))$, respectively (Figs. 1 (f,g)). The attraction shape is bounded within $[-1, 0]$ and satisfies Proposition 4.1. Any $\lambda \in [0, 1]$ would cause contraction and avoid oscillation of expansion and contraction. Thus, NEG-$t$-SNE is less sensitive to learning rate annealing, and the clusters appear less fuzzy even for constant $\lambda = 1.0$. Moreover, the repulsion shape of NEG-$t$-SNE is also bounded within $[0, 1]$ and does not approach infinity as $\zeta \to 0$, which causes fewer points to leave the clusters when sampled randomly. While Damrich et al. (2023) used only the bounded repulsive forces of NEG-$t$-SNE and unbounded ones of UMAP to explain this disparity, our analysis, in Fig. 1, shows that the attraction shape is equally responsible for stability and compactness of the clusters. **When $a$ and $b$ vary in NEG-t-SNE, it faces the same numerical challenges of UMAP.**

For additional discussion on NEG-$t$-SNE with illustration, see Appendix F. Together with the discussion in Fig. 4, this suggests that attraction and repulsion shapes from different well-behaved methods **(i.e., that guarantee Proposition 4.1) can be mixed to achieve desired embedding properties**, for instance, using the attraction shape of UMAP with the repulsion shape of NEG-$t$-SNE. We show a few examples by experimenting on UMAP, NEG-$t$-SNE, and PaCMAP in Appendix J. **We empirically show that $f_a \in [-1, 0]$ achieves better clustering, while $f_a \in [-0.5, 0]$ (prevents flips) obtains better local structure.**

## 7 DISCUSSION AND CONCLUSION

In this work, we studied the relationship of attractive and repulsive forces to cluster formation. While it is known that attractive forces bring similar points closer together and repulsive forces push dissimilar points further away, the exact mechanisms of such forces have not been well studied. Here, we have analyzed the dynamics underlying cluster formation. While we focused on UMAP and NEG-$t$-SNE in the main text, the results are general to dimensionality reduction (some other algorithms are discussed in the Appendix G).

We treated the attraction and repulsion coefficients as functional mappings and analyzed them in the context of their respective shapes. Characterizing these shapes (Propositions 4.1 and 4.2) revealed a counterintuitive result: the attractive forces of UMAP gave repulsion (expansion of distance) for shorter distances (Fig. 1 (b)), i.e., instead of bringing neighboring points closer together, it pushes them away. We conclude that UMAP's learning rate schedule alleviates this phenomenon.

We also studied the initialization of the low-dimensional embedding. Similar points starting further apart experience low attraction and never contract well enough. Our formalism provides a way to influence attraction at larger distances by adding additional terms to the attraction shape. This modification resulted in outputs that are more consistent under random initialization. This gives us confidence that UMAP encodes a unique structure, with numerical tricks merely helping to achieve it faster. Analyzing the repulsion shape revealed that higher repulsion causes a larger inter-cluster distance without changing the cluster characteristics.

**Overall, the key takeaways for the practitioners are:**

1. **Attraction and repulsion shapes characterize pair-wise dimensionality reduction methods and provide formal guidance for contraction/expansion of points. Various methods can be put on the same graph, and their properties can be described (Fig. 1(f,h)).**
2. **For effective contraction, attraction shapes that follow Proposition 4.1 are desirable. If it doesn't, learning rate tuning is required (UMAP uses annealing).**
3. **Near-sightedness in the attraction force hinders consistent output. Tweaking its shape to include long-range attraction can mitigate this issue.**
4. **To increase inter-cluster distance modifying repulsion shape is enough (Eq. 15). To break up clusters, one may modify just the attraction shape.**

These insights into attraction–repulsion dynamics offer new tools for optimizing dimensionality reduction algorithms. Beyond this, the close connection between dimensionality reduction and contrastive learning (Damrich et al., 2023; Hu et al., 2023) suggests that our approach can also enhance representation learning. Taken together, our work aims to make embeddings and their interpretations more principled, consistent, and reliable, and guide future research.

## SOFTWARE AND DATA

All the data used in this paper are publicly available. The MNIST dataset is available at https://yann.lecun.com/exdb/mnist/. Fashion-MNIST is available at https://github.com/zalandoresearch/fashion-mnist. Single-cell transcriptomes data is available at https://github.com/biolab/tsne-embedding. Additional details are provided in the Implementation Details section in Appendix K. Representative codes for reproducing the results are attached as supplementary material for review.

## LLM USAGE DISCLOSURE

The first author drafted the manuscript and used LLM tools (Grammarly and ChatGPT) for stylistic and grammatical refinement in select sections. All authors subsequently edited and approved the final text. No AI tools were used for code generation, data analysis, or research assistance.

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

APPENDIX

In Appendix A, we provide necessary derivations regarding attraction and repulsion shapes of UMAP. In Appendix B we explore using a constant learning rate following our discussion in the main text. Then, in Appendix C, we discuss the implications of improved attraction and repulsion on large language model embeddings. We formally define Procrustes distance and matrix in Appendix D.1. We explore additional datasets for the random initialization experiment in Appendix E. We extend the discussion of NEG-$t$-SNE and UMAP in Appendix F. After that, in Appendix G, we explore alternate dimensionality reduction methods. Then, for completion, we discuss the construction of the high-dimensional graph in these methods in Appendix H. In Appendix I, we show the detailed result of varying $\lambda_a$ and $\lambda_r$ (from Fig. 1) for UMAP, NEG-$t$-SNE, and PaCMAP on different datasets. We show the effect of mixing and matching the shapes from different algorithms in Appendix J. Finally, we provide implementation details in Appendix K.

# A  PROOFS AND DERIVATIONS

## A.1  PROOF OF PROPOSITION 4.1

*Proof.* We subtract Eq. (7) from Eq. (8) and take a norm:

$$||y_i^{t+1} - y_j^{t+1}|| = |1 + 2\lambda f_a|||y_i^t - y_j^t||. \tag{16}$$

This distance contracts as long as $|1 + 2\lambda f_a| < 1$, i.e., provided

$$-1 < \lambda f_a < 0, \tag{17}$$

$\square$

## A.2  PROOF OF PROPOSITION 4.2

*Proof.* We subtract Eq. (9) from Eq. (10) and take a norm:

$$||y_i^{t+1} - y_j^{t+1}|| = |1 + \lambda f_r|||y_i^t - y_j^t||. \tag{18}$$

This distance increases when $|1 + \lambda f_r| > 1$, i.e.,

$$\lambda f_r < -2 \ \text{ or } \ f_r > 0. \tag{19}$$

From the definition of $f_r$, the latter suffices. $\square$

## A.3  ATTRACTION SHAPE:

We use a general form of the low-dimensional affinity function, i.e., $q_{ij} = (\gamma + a||y_i - y_j||_1^{2b})^{-1}$, to derive the attraction shape. It reduces to UMAP for $\gamma = 1$ and to NEG-$t$-SNE for $\gamma = 2$. The attractive force is given by

$$\nabla_{y_i} \log q_{ij} = -\nabla_{y_i} \log\left(\gamma + a(||y_i - y_j||_2^2)^b\right)$$

$$= -\frac{1}{\gamma + a(||y_i - y_j||_2^2)^b} \nabla_{y_i}(\gamma + a(||y_i - y_j||_2^2)^b)$$

$$= -\frac{1}{\gamma + a(||y_i - y_j||_2^2)^b} ab(||y_i - y_j||_2^2)^{b-1} \nabla_{y_i}||y_i - y_j||_2^2$$

$$= -\frac{2ab(||y_i - y_j||_2^2)^{b-1}}{\gamma + a(||y_i - y_j||_2^2)^b}(y_i - y_j). \tag{20}$$

Defining $\zeta = ||y_i - y_j||_2$, the first term gives the attraction shape as:

$$f_a(\zeta) = -\frac{2ab\zeta^{2(b-1)}}{\gamma + a\zeta^{2b}} \tag{21}$$

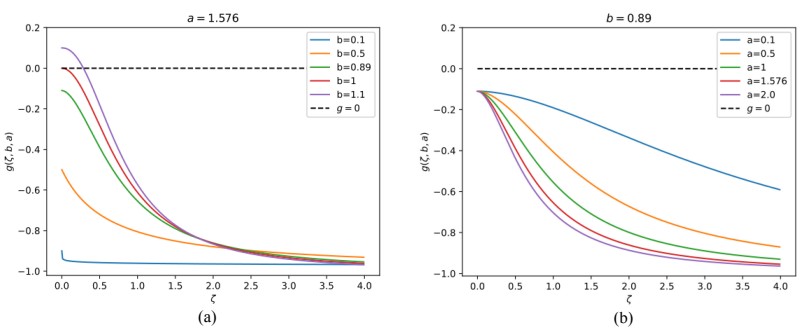

Figure 5: Values of $g(\zeta, b, a)$ for (a) $a$ fixed at $1.576$ and (b) $b$ fixed at $0.89$.

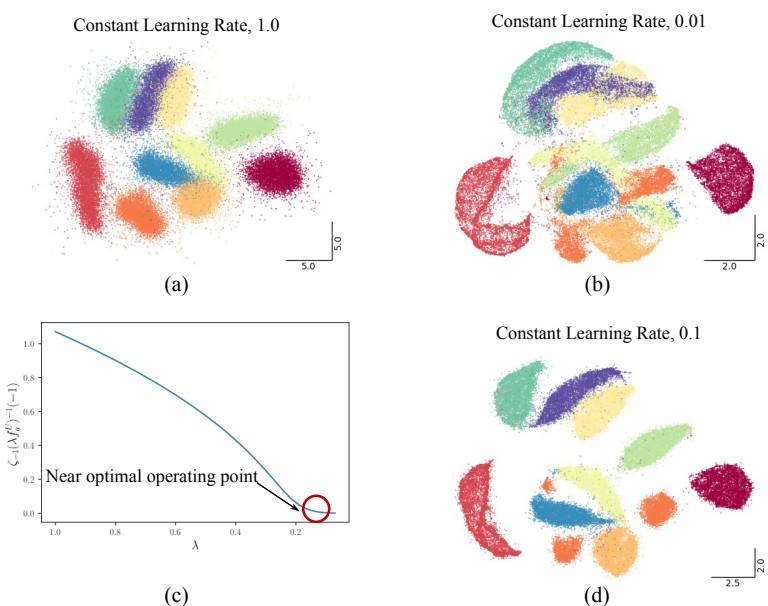

Figure 6: Effect of constant learning rate in embeddings. (a) When the learning rate is too high ($\lambda = 1.0$), the embeddings are diffuse (because of the high value of $\zeta_{-1}$). (b) When the learning rate is too low ($\lambda = 0.01$), clusters don't form (the strengths of attraction and repulsion are too low). (c) $\zeta_{-1}$ decreases nonlinearly as the learning rate decreases. The goal of the algorithm is to reduce $\zeta_{-1}$ while keeping effective levels of attraction and repulsion. (d) Distinct and compact clusters form at a constant, near-optimal learning rate $\lambda = 0.1$.

## A.4 CONDITION FOR STRICTLY INCREASING $f_a^U$:

Its behavior with distance can be characterized by computing the derivative of $f_a^U$:

$$\frac{df_a^U(\zeta)}{d\zeta} = -\frac{2ab\zeta^{2b-3}}{\gamma + a\zeta^{2b}}\left((b-1) - \frac{ab\zeta^{2b}}{\gamma + a\zeta^{2b}}\right). \tag{22}$$

This leads to a strictly increasing condition ($\frac{df_a^U}{d\zeta} > 0$),

$$g(\zeta, b, a) < 0, \tag{23}$$

where $g(\zeta, b, a) = b - 1 - \frac{ab\zeta^{2b}}{\gamma + a\zeta^{2b}}$. This inequality is valid as long as $0 < b \le 1$ (using the derivative and asymptotes of $g$). Figure 5 shows values of $g$ for different $b$ and $a$. As $b$ increases above 1, the inequality (23) no longer holds.

## A.5 REPULSION SHAPE:

The repulsive force, using the general form of the low-dimensional affinity, is given by

$$
\nabla_{y_i} \log (1 - q_{ij}) = \nabla_{y_i} \log \left[ 1 - \frac{1}{\gamma + a(||y_i - y_j||_2^2)^b} \right]
$$

$$
= \frac{\gamma + a(||y_i - y_j||_2^2)^b}{(\gamma - 1) + a(||y_i - y_j||_2^2)^b} \nabla_{y_i} \left[ 1 - \frac{1}{\gamma + a(||y_i - y_j||_2^2)^b} \right]
$$

$$
= \frac{1}{\gamma - 1 + a(||y_i - y_j||_2^2)^b} \frac{ab(||y_i - y_j||_2^2)^{b-1}}{\gamma + a(||y_i - y_j||_2^2)^b} \nabla_{y_i} ||y_i - y_j||_2^2
$$

$$
= \frac{2ab(||y_i - y_j||_2^2)^{b-1}}{(\gamma - 1 + a(||y_i - y_j||_2^2)^b)(\gamma + a(||y_i - y_j||_2^2)^b)} (y_i - y_j). \tag{24}
$$

The first term gives the repulsion shape as:

$$
f_r(\zeta = ||y_i - y_j||_2) = \frac{2ab\zeta^{2(b-1)}}{(\gamma - 1 + a\zeta^{2b})(\gamma + a\zeta^{2b})}. \tag{25}
$$

Generally, $f_r > 0$.

## A.6 LOSS FUNCTIONS DUE TO MODIFIED ATTRACTION AND REPULSION SHAPE:

The cost function of the attractive term with the modification in Eq. (13) is given by

$$
-\int (f_a^U(||y_i - y_j||_2) - \beta ||y_i - y_j||_2)(y_i - y_j) dy_i = -\log(q_{ij}) + \frac{\beta}{3} ||y_i - y_j||_2^3, \tag{26}
$$

whereas the repulsive term due to Eq. (15) is given by

$$
-\int (f_r^U(||y_i - y_j||_2) + \epsilon)(y_i - y_j) dy_i = -\log(1 - q_{ij}) - \frac{\epsilon}{2} ||y_i - y_j||_2^2. \tag{27}
$$

In both cases, the additional term acts as a regularizer, simply using norms. However, when we directly add this term to attraction and repulsion shapes, we can easily explain what each term is doing. For the attractive term, it is increasing far-sightedness, whereas for the repulsive term, it adds a constant repulsive coefficient.

## B   MORE ON UMAP'S LEARNING RATE

As we discussed in Sections 5 and 6 of the main text, it is believed that UMAP requires learning rate annealing (Fig 6). To explain this, in Section 5, we defined the concept of minimum distance for contraction ($\zeta_{-1}$) and established that reducing this value through learning rate annealing results in compact clusters. Later, in Section 6, we compared attraction shapes of UMAP (for $a = 1.0$ and $b = 1.0$) and Neg-$t$-SNE and explained that Neg-$t$-SNE can withstand a constant learning rate of 1.0 better than UMAP because it's attraction shape resides within $[-1, 0]$ while UMAP's is within $[-2, 0]$. Following the same logic, we showed that UMAP can withstand a constant learning of 0.5 by making its attraction shape stay within $[-1, 0]$ (Figs. 11(f,h)).

However, the embeddings are still better if the learning rate anneals (for a wide range of parameters of $a$ and $b$). This is because the goal of the algorithm is to eventually reduce $\zeta_{-1}$ to zero and it occurs when the learning rate reduces close to zero. Otherwise, the embedding becomes diffused (Fig 6(a)). On the other hand, if the learning rate is too low, to begin with, the strength of attraction and repulsion is too low, and thus no clear clusters can form (Fig 6(b)). By analyzing $\zeta_{-1}$ vs $\lambda$ curve (Fig 6(c)), we see that a near optimal point is $\lambda = 0.1$ where the value of $\zeta_{-1}$ is low with considerable attraction and repulsion strength. Setting the constant learning rate to 0.1, we obtain compact clusters with clear boundaries (Fig 6(d)).

## C   LARGE LANGUAGE MODEL EMBEDDING

Increased attraction and repulsion often produce better embeddings. To show this, we randomly selected 300,000 samples of features using PubMedBERT (Gu et al., 2021) with known labels

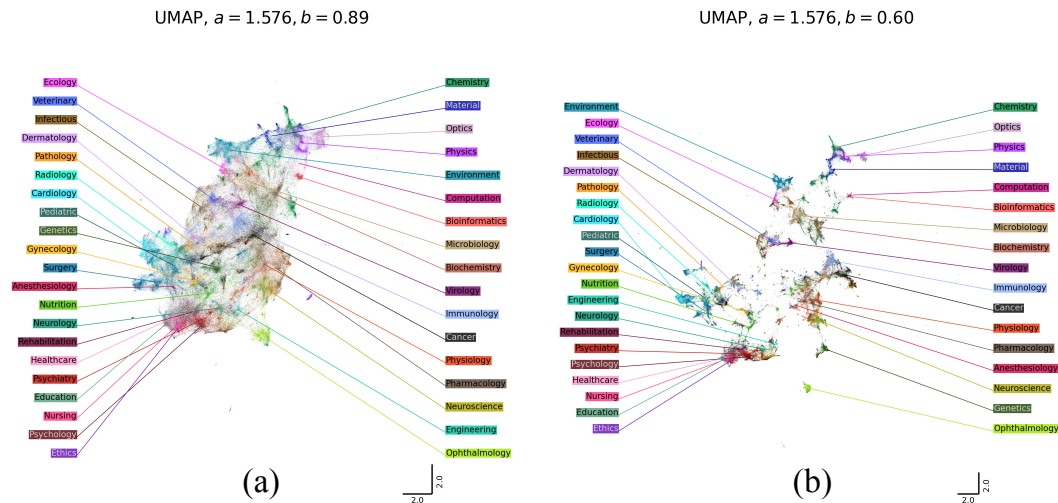

Figure 7: Visualization of 300K abstracts from the PubMed dataset. (a) Default UMAP ($a = 1.576, b = 0.89$) shows labels overlapping each other in the mapping, but (b) UMAP with improved repulsion and attraction ($a = 1.576, b = 0.60$) shows better cluster formation and non-overlapping labels. Both mappings are labeled for easier understanding.

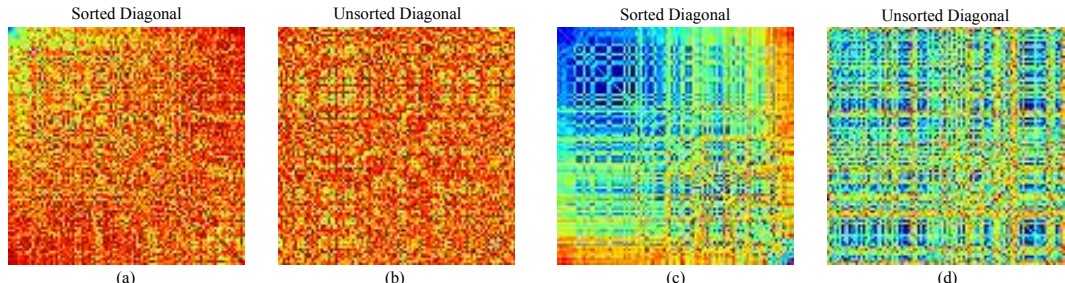

Figure 8: Effect of sorting the diagonal of Procrustes matrix on its visualization. (a) Procrustes matrix reproduced from 2(f) for the default UMAP attraction shape. The diagonal of the matrix is sorted by $p_d$ of a sample embedding with PCA initialization. (b) The same data as in (a), but the diagonal is unsorted (or randomly sorted). (c) Procrustes matrix reproduced from 2(g) for modified attraction shape, where the diagonal of the matrix is sorted by $p_d$ of a sample embedding with PCA initialization. (d) The same data as in (c), but the diagonal is unsorted.

from the PubMed dataset (González-Márquez et al., 2024). The default UMAP provides a crowded structure where labels (colors) overlap (Fig. 7(a)). Without labels, one might be perplexed about the embeddings, as explicit clusters are absent. However, by adjusting $b$ to 0.6 (as discussed in Section 5.2), we increase repulsion and simultaneously facilitate cluster formation through improved attraction. The mapping is explorable even without explicit labeling (Fig. 7(b)). The improvement of the embedding can be quantified by the k-NN accuracy ($k = 10$). On a hold-out data of $10,000$ points, k-NN accuracy has increased from $49.2\%$ to $55.48\%$ by reducing $b$ to 0.6.

## D  METRICS

### D.1  PROCRUSTES DISTANCE AND PROCRUSTES MATRIX

The Procrustes distance (Gower, 1975) measures similarity between two point clouds $\{x\}$ and $\{y\}$ under linear transformations, viz. translation, scaling, and rotation. Operationally, we hold the former fixed and vary the latter until the two sets are in maximum alignment. Let $\{y'\}$ be the transformation

of $\{y\}$ that achieves this objective. Then the Procrustes distance is given by

$$p_d(\{x\}, \{y\}) = \sqrt{\sum_k (x_k - y_k')}. \tag{28}$$

The Procrustes distance is a linear measure that has proven useful in a variety of settings (McInnes et al., 2018; Islam & Fleischer, 2022; Kotlov et al., 2024).

Here, we use the Procrustes distance to measure the consistency of embedding under random initialization. Let $\{x\}_p$ be a reference embedding (using PCA initialization in our experiments), and $X_r = \{\{x\}_i | i = 1, 2, 3, \ldots, N\}$ be a set of $N$ embeddings obtained from random initialization. The similarity of the embeddings can be quantified by taking a mean and standard deviation of the strictly lower triangular values of the matrix $P$ (reported as mean±std in Figs. 2, 9, and 10), with

$$\text{mean} = \frac{2}{N(N-1)} \sum_{i,j(i>j)} P_{i,j}, \ \text{ and } \ \text{std} = \sqrt{\sum_{i,j(i>j)} \frac{2(P_{i,j} - \text{mean})^2}{N(N-1)}} \tag{29}$$

The indexes of $X_r$ can be sorted such that $p_d(\{x\}_i, \{x\}_p) \le p_d(\{x\}_{i+1}, \{x\}_p)$, so that the diagonal values of the Procrustes matrix are given by

$$P_{i,i} = p_d(\{x\}_i, \{x\}_p) \tag{30}$$

and the off-diagonal values are given by

$$P_{i,j} = p_d(\{x\}_i, \{x\}_j). \tag{31}$$

Numerically, the sorting of the diagonal adds little value. Visually, however, the ordering reveals the underlying self-similarity of the embeddings. For example, in Fig. 8(a), the sorted diagonal shows that similar embeddings clump in the upper left region of the matrix. When we use a modified attraction shape, the number of points that are similar to each other increases (as shown by the larger blue region in Fig. 8(c)), indicating the presence of a metastable point in the embedding algorithm. On the other hand, when the diagonal is unsorted, this region disappears, and any sense of similar embeddings is lost (Fig. 8(b,d)).

## D.2 TRUSTWORTHINESS

The trustworthiness metric (Venna & Kaski, 2001) quantifies how well local neighborhoods are preserved after dimensionality reduction:

$$T = 1 - \frac{2}{nk(2n - 3k - 1)} \sum_{i=1}^{n} \sum_{y_j \in \text{KNN}(y_i, k)} \max(0, r(i, j) - k) \tag{32}$$

where $KNN(y_i, k)$ is the $k$-NN graph in the embedding space and $r(i, j)$ is the rank of $x_j$ in the high-dimensional $k$-NN graph. In practice, $k$ is often set to 5, assessing preservation of each point's five nearest neighbors. For computational efficiency, when we report trustworthiness, we randomly sampled 10,000 indices. When comparing different embeddings, we used the same indices.

## D.3 SILHOUETTE SCORE

While the silhouette score (Rousseeuw, 1987) aims to assess clustering algorithms, we use it to evaluate label separation in the embeddings, i.e, how well the ground truth labels have been clustered. The idea is that the embedding algorithms naturally produce clusters and should separate the labels as much as possible. For a point $y_i$ in a point cloud $\{y\}$, let $a_i$ be the mean distance from $y_i$ to other points in its own label, and let $b_i$ be the minimal mean distance from $y_i$ to points in any other label. The pointwise silhouette is thus given by

$$s_i = \frac{b_i - a_i}{\max\{a_i, b_i\}} \tag{33}$$

This value lies within $[-1, 1]$. A value close to 1 means that $y_i$ fits within its own label cluster, near 0 suggests a boundary point, and close to $-1$ indicates failed label separation. The overall silhouette score is

$$\text{SIL} = \frac{1}{N} \sum s_i. \tag{34}$$

We computed the silhouette score for the whole embedding (no random sampling) using Euclidean distances.

## E  EFFECT OF RANDOM INITIALIZATION IN ADDITIONAL DATASETS

In the main text (Section 5.1), we showed results only on the MNIST dataset. Here we perform the same experiment on the Fashion-MNIST (FMNIST) (Xiao et al., 2017) and single-cell tran-scriptomes (Macosko et al., 2015) data (Fig. 9 and 10, respectively). The main conclusion remains unchanged: modified and composite attraction shapes, such as those that increase attraction at large distances, significantly improve the consistency of reconstruction.

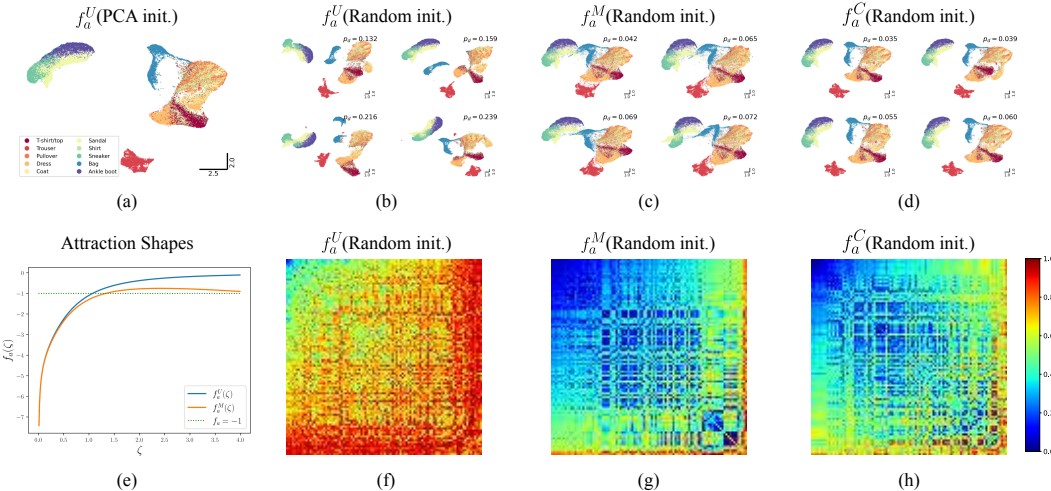

Figure 9: Effect of random UMAP initialization on different attraction shapes on FMNIST data. (a) Mapping using PCA as a standard. (b-d) Four mappings with the lowest Procrustes distance ($p_d$) from the embedding in (a) for (b) default, (c) modified, and (d) composite attraction shapes. (e) Default UMAP and modified attraction shapes. (f-h) Procrustes matrix obtained from 100 runs of (f) default ($0.72 \pm 0.15$), (g) modified ($0.38 \pm 0.19$), and (h) composite ($0.43 \pm 0.18$) attraction shapes.

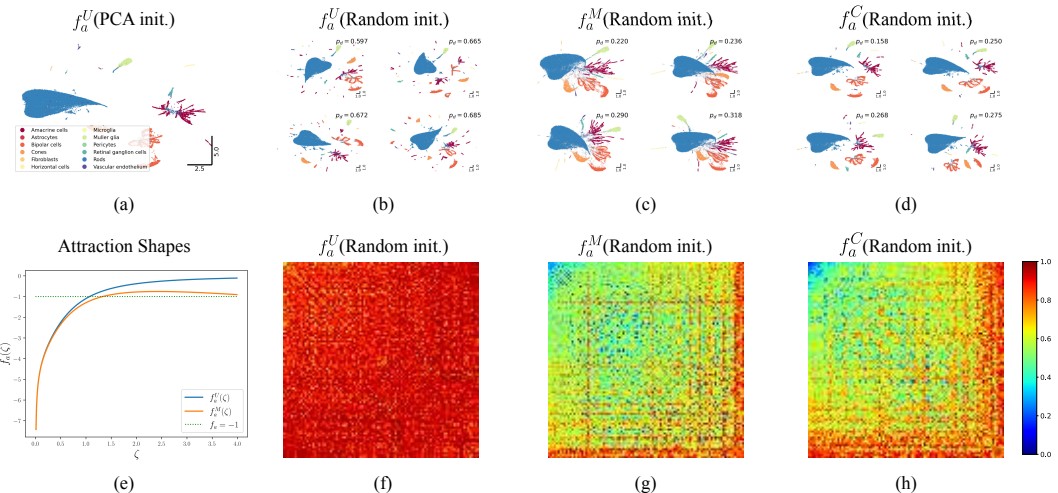

Figure 10: Effect of random UMAP initialization on different attraction shapes on single-cell transcriptomes data. (a) Mapping using PCA as a standard. (b-d) Four mappings with the lowest Procrustes distance ($p_d$) from the embedding in (a) for (b) default, (c) modified, and (d) composite attraction shapes. (e) Default UMAP and modified attraction shapes. (f-h) Procrustes matrix obtained from 100 runs of (f) default ($0.91 \pm 0.06$), (g) modified ($0.61 \pm 0.13$), and (h) composite ($0.64 \pm 0.15$) attraction shapes.

# F MORE ON NEG-$t$-SNE

## F.1 COMPARISON TO UMAP

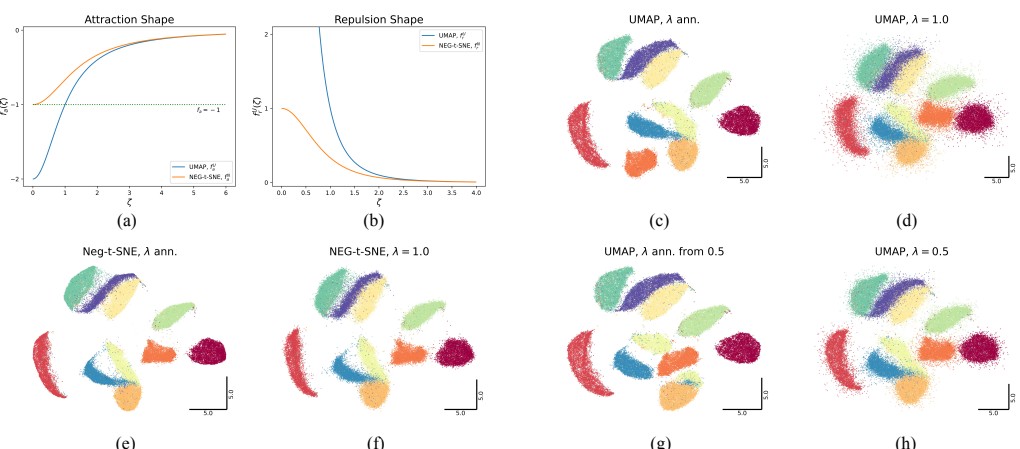

Figure 11: Sensitivity of UMAP and NEG-$t$-SNE to learning rate on the MNIST dataset. (a) Attraction and (b) repulsion shapes for UMAP ($a = 1$, $b = 1$) and NEG-$t$-SNE. (c,d) UMAP is very sensitive to the learning rate $\lambda$, as $f_a^U < -1$ as the separation distance $\zeta$ decreases. Thus, without annealing, the clusters become fuzzy. (e,f) NEG-$t$-SNE is less sensitive to $\lambda$ as $f_a^N \in [-1, 0]$ always, and the clusters are thus less fuzzy even when not annealed. (g,h) Confining $f_a^U$ to $[-1, 0]$ by setting $\lambda = 0.5$ shows less sensitivity to $\lambda$.

Figure 11 shows the shapes of UMAP and NEG-$t$-SNE, along with various MNIST embeddings. When the learning rate is annealed, both UMAP and NEG-$t$-SNE show similar output (Figs. 11(c,e)). However, when the learning rate is a constant value of 1, the UMAP shows a fuzzy structure, while NEG-$t$-SNE shows a structure with much cleaner boundaries (Fig. 11(d,f)). The discussion in Section 6 suggests that constraining $f_a^U$ within $[-1, 0]$ can potentially result in less fuzzy clusters for fixed $\lambda$. We have seen this previously in Fig. 1(h,i) as well, where UMAP provided better embedding and clustering when the learning rate for attraction ($\lambda_a$) was $\lesssim 0.5$. A straightforward way to achieve this is to initialize $\lambda$ to 0.5, which satisfies Proposition 4.1 for all $\zeta$. The resulting embeddings, shown in Figs. 11 (g) and (h), confirm that clusters are similar to those of NEG-$t$-SNE's, and for a constant $\lambda = 0.5$, the clusters are less fuzzy than before as predicted with sharper boundaries (Fig. 11(d,h)). There are still a few points outside the clusters due to the characteristics of UMAP's repulsion shape, which NEG-$t$-SNE solves.

Next, we can introduce the parameters $a$ and $b$ into NEG-$t$-SNE (essentially the formulation of Parametric UMAP; see Proposition 6.1). The affinity function becomes $q_{ij}^N = 1/(2 + a\zeta^{2b})$, and the attraction and repulsion shapes become $f_a^N = -\frac{2ab\zeta^{2(b-1)}}{2+a\zeta^{2b}}$, and $f_r^N = \frac{2ab\zeta^{2(b-1)}}{(1+a\zeta^{2b})(2+a\zeta^{2b})}$, respectively. For $0 < b < 1$, both shapes become unbounded as $\zeta \to 0$. Thus, NEG-$t$-SNE will face similar numerical challenges to UMAP if $a$ and $b$ vary, and corresponding limitations carry over. One notable distinction is that, compared to UMAP, the attraction shape attains a lower minimum distance ($\zeta_{-1}$) for the attraction. While this may enhance cluster formation, it approaches zero faster (increased near-sightedness as distance increases), potentially diminishing its effectiveness for attraction over longer distances.

## F.2 COMPARISON TO PARAMETRIC UMAP

Parametric UMAP was initially trained with the original UMAP objective (Sainburg et al., 2021), but later work adopted a numerically stable, log-sigmoid–based modified cross-entropy loss (Shi et al., 2023). This modification makes Parametric UMAP and Neg-t-SNE Damrich et al. (2023) equivalent (Proposition 6.1). We show the equivalence below.

PROOF OF PROPOSITION 6.1

*Proof.* The kernel function under this modification becomes

$$q_{ij}^P = -\log\left(1 + a||y_i - y_j||_2^{2b}\right). \tag{35}$$

The attractive term is

$$-\text{logsigmoid}(q_{ij}^P) = -\log\left(\frac{1}{1 + \exp(-q_{ij}^P)}\right) \tag{36}$$

$$= -\log\left(\frac{1}{2 + a||y_i - y_j||_2^{2b}}\right) \tag{37}$$

$$= -\log(q_{ij}^N). \tag{38}$$

And the repulsive term is

$$\text{logsidmoid}(q_{ij}^P) - q_{ij}^P = \log\left(\frac{1}{2 + a||y_i - y_j||_2^{2b}}\right) + \log\left(1 + a||y_i - y_j||_2^{2b}\right) \tag{39}$$

$$= \log\left(\frac{1 + a||y_i - y_j||_2^{2b}}{2 + a||y_i - y_j||_2^{2b}}\right) \tag{40}$$

$$= \log\left(1 - \frac{1}{2 + a||y_i - y_j||_2^{2b}}\right) \tag{41}$$

$$= \log(1 - q_{ij}^N). \tag{42}$$

Both these are NEG-t-SNE with explicit parameters $a$ and $b$ (while in Neg-$t$-SNE these are set to 1). $\square$

# G   ALTERNATE DIMENSIONALITY REDUCTION ALGORITHMS

The alternative algorithms we consider use the same kernel function as UMAP (with $a = 1$ and $b = 1$ in their low-dimensional weight):

$$q_{ij} = \frac{1}{1 + ||y_i - y_j||_2^2}. \tag{43}$$

In this section, we first discuss the TriMap (Amid & Warmuth, 2019) algorithm. Even though, this algorithm relies on triplets (and not pairwise interactions), this works as a primer for analyzing attraction and repulsion that are a bit more involved than UMAP. This discussion is followed by Pairwise Controlled Manifold Approximation (PaCMAP) (Wang et al., 2021) and it's extension Pairwise Controlled Manifold Approximation with Local Adjusted Graph (LocalMAP) (Wang et al., 2025) that modify TriMAP's loss function that works for pairwise interactions. We then tackle $t$-SNE (Van der Maaten & Hinton, 2008). Then, we provide a short note on SNE (Hinton & Roweis, 2002) that uses an alternate kernel function and finally end the section by briefly discussing multidimensional scaling (Borg & Groenen, 2007).

## G.1   TRIMAP

TriMap (Amid & Warmuth, 2019) optimizes low-dimensional using a triplet loss

$$\mathcal{L}^T = \sum_{(i,j,k)} w_{ijk} \frac{1}{1 + \frac{q_{ij}}{q_{ik}}}, \tag{44}$$

where $w_{ijk}$ is the weight of the triplet $(y_i, y_j, y_k)$, $y_j$ is in the $k$-nearest neighbor set of $y_i$ in the high dimension, and $y_k$ is a far-away point. When minimized, we expect that $y_i$ and $y_j$ attract each other, while $y_i$ and $y_k$ repel each other. The update equations are

$$y_i^{t+1} = y_i^t + \lambda f_a^T(\zeta_1^t, \zeta_2^t)(y_i^t - y_j^t) + \lambda f_r^T(\zeta_1, \zeta_2)(y_i^t - y_k^t), \tag{45}$$

$$y_j^{t+1} = y_j^t - \lambda f_a^T(\zeta_1^t, \zeta_2^t)(y_i^t - y_j^t), \tag{46}$$

$$y_k^{t+1} = y_k^t - \lambda f_r^t(\zeta_1^t, \zeta_2^t)(y_i^t - y_k^t), \tag{47}$$

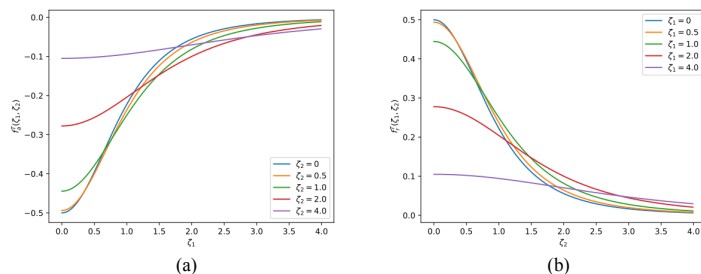

Figure 12: (a) Attraction shapes of TriMap for different $\zeta_2$ and (b) repulsion shapes of TriMap for different $\zeta_1$.

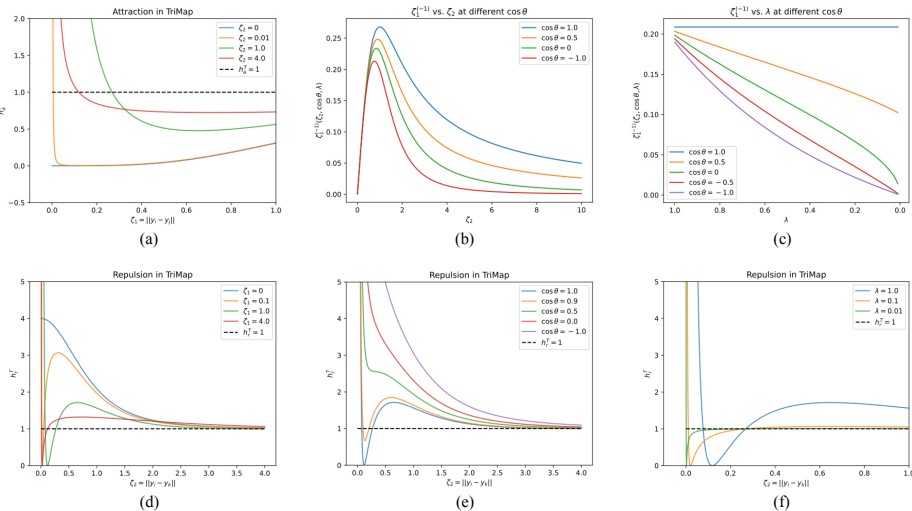

Figure 13: Attraction and repulsion behavior in TriMap. (a) $h_a^T$ vs $\zeta_1$ for different $\zeta_2$. Any values below the dotted line indicate attraction. Like UMAP, TriMAP shows attraction and repulsion for nearest neighbors $(y_i, y_j)$. (b,c) Unlike UMAP, the minimum distance for contraction $(\zeta_1^{(-1)})$ varies due to dependence on (b) $\zeta_2$ and (c) $\lambda$; the function $\cos\theta$ regulates the range of these values. (d-f) Repulsion in TriMap by varying (d) $\zeta_1$, (e) $\cos\theta$, and (e) $\lambda$. Values above (below) the dotted line indicate repulsion (attraction). While the repulsion force of UMAP shows only repulsion, that of TriMap can provide both attraction and repulsion. Unless otherwise labeled, $\zeta_1 = 1.0$, $\zeta_2 = 0.5$, $\cos\theta = 1.0$, and $\lambda = 1.0$.

where $\zeta_1 = ||y_i - y_j||_2$ is the distance between nearest neighbors, $\zeta_2 = ||y_i - y_k||_2$ is the distance from the faraway point, and $f_a^T$ and $f_r^T$ are attraction and repulsion shapes of TriMap, respectively. Unlike UMAP, the attractive and repulsive components are non-separable and the shapes depend on two distance measures (making them 2D). The functional form of the attraction shape is

$$f_a^T(\zeta_1, \zeta_2) = -\frac{2(1 + \zeta_2^2)}{(2 + \zeta_1^2 + \zeta_2^2)^2}, \tag{48}$$

and the repulsion shape is

$$f_r^T(\zeta_1, \zeta_2) = \frac{2(1 + \zeta_1^2)}{(2 + \zeta_1^2 + \zeta_2^2)^2}. \tag{49}$$

The attraction and repulsion shapes (Fig. 12) of TriMap shows similar trends of that of UMAP. However, the minium value of repulsion shape is $-0.5$; thus, unlike UMAP there is no position flipping in TriMAP due to attraction alone. However, since Eqs. (45-47) are not decoupled between attractive and repulsive terms, Propositions 4.1 and 4.2 do not apply. Focusing on attraction first, we show

**Proposition G.1.** *Update equations (45)-(47) provide a contraction if*

$$h_a^T(\zeta_1, \zeta_2, \theta, \lambda) < 1, \tag{50}$$

*where* $h_a^T(\zeta_1, \zeta_2, \theta, \lambda) = (1 + 2\lambda f_a^T)^2 + 2(1 + 2\lambda f_a^T)\lambda f_r^t \frac{\zeta_2}{\zeta_1}\cos\theta + (\lambda f_r^T)^2\frac{\zeta_2^2}{\zeta_1^2}$ *and* $\theta$ *is the angle between the vectors* $(y_i - y_j)$ *and* $(y_i - y_k)$, *i.e.,* $\cos\theta = \frac{(y_i^t - y_j^t)^T(y_i^t - y_k^t)}{||y_i^t - y_j^t||_2 ||y_i^t - y_k^t||_2}$.

*Proof.* We require

$$||y_i^{t+1} - y_j^{t+1}||_2^2 < ||y_i^t - y_j^t||_2^2. \tag{51}$$

From Eq. (45) and (46): $y_i^{t+1} - y_j^{t+1} = (1 + 2\lambda f_a^T)(y_i^t - y_j^t) + \lambda f_r^T(y_i^t - y_k^t)$. Putting this value in Eq. (51), we obtain the desired inequality. $\square$

Inequality (50) depends on $\zeta_1$, $\zeta_2$, $\cos\theta$ and $\lambda$. In particular, the value of $\zeta_1 = ||y_i - y_j||_2$, where we want a contraction, is coupled with additional variables. Figure 13(a) shows the attraction behavior for various values of $\zeta_2$, while $\cos\theta = 1$ and $\lambda = 1$. The values below the dotted line indicate attraction (and thus contraction of distance $\zeta_1$), whereas the values above indicate repulsion (and therefore expansion of distance $\zeta_1$). The value where the dotted line and $h_a^T$ meet gives the minimum distance for contraction ($\zeta_1^{(-1)}$) [analogous to $\zeta_{-1}$ of UMAP], which we define as

$$\zeta_1^{(-1)}(\zeta_2, \theta, \lambda) = \arg\min_{\zeta_1} |h_a^T(\zeta_1, \zeta_2, \theta, \lambda) - 1|, \tag{52}$$

$$\text{s.t. } \zeta_1 \geq 0. \tag{53}$$

$\zeta_1^{(-1)}$ has a finite value and is $> 0$ for most cases (Fig. 13(b,c)). As a result, TriMap can show behavior similar to UMAP and thus require learning rate annealing or a similar approach (in the TriMap implementation, the authors use the delta-bar-delta (Jacobs, 1988) method under appropriate initialization).

**Proposition G.2.** *Update equations (45)-(47) provide expansion if*

$$h_r^T(\zeta_1, \zeta_2, \theta, \lambda) > 1, \tag{54}$$

*where* $h_r^T(\zeta_1, \zeta_2, \theta, \lambda) = (1 + 2\lambda f_r^T)^2 + 2\lambda f_a^T(1 + \lambda f_r^T)\frac{\zeta_1}{\zeta_2}\cos\theta + (\lambda f_a^T)^2\frac{\zeta_1^2}{\zeta_2^2}$.

*Proof.* We require

$$||y_i^{t+1} - y_k^{t+1}||_2^2 > ||y_jt - y_k^t||_2^2. \tag{55}$$

From Eq. (45) and (47): $y_j^{t+1} - y_k^{t+1} = (1 + 2\lambda f_r^T)(y_i^t - y_k^t) + \lambda f_a^T(y_i^t - y_j^t)$. Putting this value in Eq. (55) we obtain inequality (54). $\square$

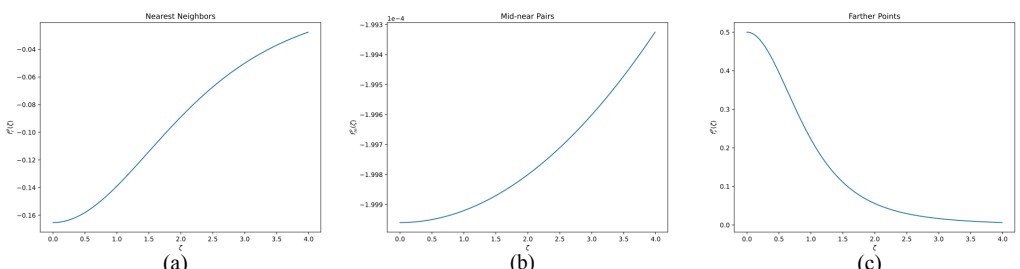

Figure 14: Attraction and repulsion shapes of PaCMAP. (a,b) Attraction shapes for (a) nearest-neighbor and (b) mid-near points (note that the values are on the order of $10^{-4}$). (c) Repulsion shape for farther pairs. $\lambda = 1$ for all figures.

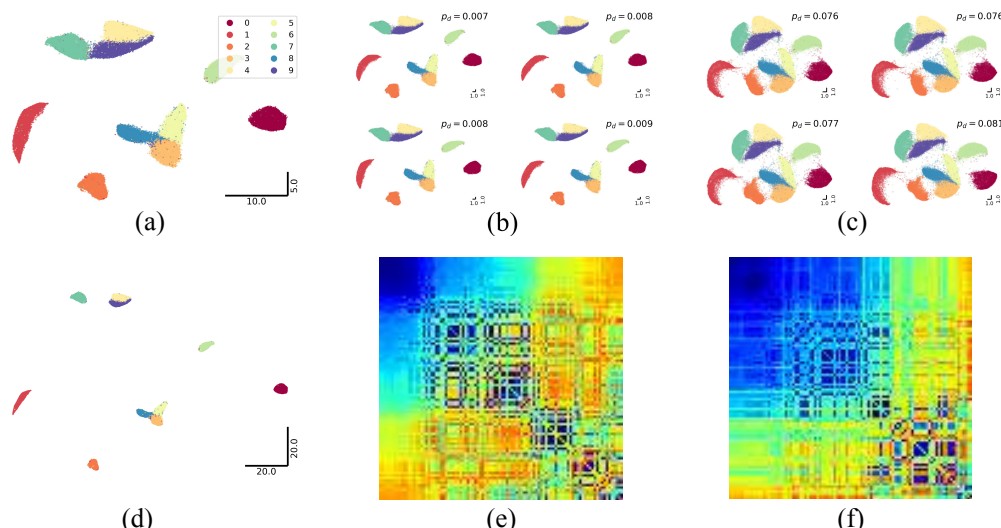

Figure 15: PaCMAP behavior for different conditions. (a) PaCMAP of MNIST data with PCA initialization. (b,c) Four samples that best match with (a) by (b) original PaCMAP and (c) PaCMAP with modified attraction shape, $f_a^M(\zeta) = f_a^P(\zeta) - 0.001\zeta$, when randomly initialized. (d) PaCMAP of MNIST with a modified repulsion shape $f_r^M = f_r^P + 0.00005$. (e,f) Procrustes matrix for (e) original PaCMAP ($0.51 \pm 0.22$) and (f) PaCMAP with modified attraction shape ($0.43 \pm 0.22$). Similar to UMAP, increased attraction at farther distances show improved consistency, while increased repulsion shows smaller clusters and larger inter-cluster distances.

Inequality (54) also depends on the set $\zeta_1$, $\zeta_2$, $\cos\theta$ and $\lambda$. Here, we are interested in the expansion of $\zeta_2 = ||y_i - y_k||_2$. Figures 13(d-f) show the repulsion behavior by varying the other quantities. Any values above the dotted line indicate repulsion (and thus expansion of distance), while the values below indicate attraction. The striking difference compared to UMAP is that repulsion in TriMap can cause contraction instead of expansion. Since this anomaly occurs for small distances, it can be avoided by an appropriate initialization and choice of triplets.

### G.2 PAIRWISE CONTROLLED MANIFOLD APPROXIMATION (PACMAP)

PaCMAP (Wang et al., 2021) optimizes low-dimensional embedding at different scales. The loss function is

$$\mathcal{L}^P = w_{NB} \sum_{(i,j) \in NN} \frac{1}{1 + 10q_{ij}} + w_{MN} \sum_{(i,k) \in MN} \frac{1}{1 + 10000q_{ik}} + w_{FP} \sum_{(i,l) \in FP} \frac{1}{1 + \frac{1}{q_{il}}}, \quad (56)$$

where $w_{NB}$, $w_{MN}$, and $w_{FP}$ are weights of the nearest neighbor (NN) pairs, mid-near (MN) pairs, and further pairs (FP), respectively (details in Appendix H). The first two terms provide attraction, whereas the last term provides repulsion. A closer look at the loss function reveals that the function is a modified form of TriMap's triplet loss. For the attractive terms, it replaces TriMap's affinity for distant points with constant terms ($1/10$ for nearest neighbors, $1/10000$ for mid-near pairs, and $1$ for farther points). This loss function is thus separable into three terms and decouples the update equations. The update equations of the nearest neighbor term are

$$y_i^{t+1} = y_i^t + \lambda f_a^P(\zeta_1^t)(y_i^t - y_j^t), \tag{57}$$

$$y_j^{y+1} = y_j^t - \lambda f_a^P(\zeta_1^t)(y_i^t - y_j^t), \tag{58}$$

of the mid-near pairs are

$$y_i^{t+1} = y_i^t + \lambda f_m^P(\zeta_2^t)(y_i^t - y_k^t), \tag{59}$$

$$y_k^{y+1} = y_k^t - \lambda f_m^P(\zeta_2^t)(y_i^t - y_k^t), \tag{60}$$

and of the farthest pairs are

$$y_i^{t+1} = y_i^t + \lambda f_r^P(\zeta_3^t)(y_i^t - y_l^t), \tag{61}$$

$$y_l^{t+1} = y_l^t - \lambda f_r^P(\zeta_3^t)(y_i^t - y_l^t), \tag{62}$$

where $\zeta_1 = ||y_i - y_j||_2$, $\zeta_2 = ||y_i - y_k||_2$, $\zeta_3 = ||y_i - y_l||_2$ are distances, $f_a^P$ and $f_m^P$ are attraction shapes for nearest neighbors and mid-near pairs, respectively, and $f_r^P$ is the repulsion shape for the farthest pairs. Correspondingly, the functional forms of the shapes are

$$f_a^P(\zeta) = -\frac{20}{(11 + \zeta^2)^2}, \tag{63}$$

$$f_m^P(\zeta) = -\frac{20000}{(10001 + \zeta^2)^2}, \tag{64}$$

$$f_r^P(\zeta) = \frac{2}{(2 + \zeta^2)^2}. \tag{65}$$

$f_a^P$ and $f_m^P$ follow Proposition 4.1, and $f_r$ follows Proposition 4.2 (Fig. 14). The attraction is quite low compared to UMAP, but it is good enough for a wide range of learning rates (modulated by the Adam algorithm (Kingma & Ba, 2015)); with $w_{NB} = 3$ the maximum attraction is always below $0.5$ preventing any flips during attractin update. Typically, PaCMAP initializes the embedding within a small sphere in the low dimension (e.g., in (Wang et al., 2021), the initialization is often on the order of $10^{-3}$) and relies on repulsion to separate the individual clusters. Overall, it mostly recovers UMAP's clustering properties (especially for MNIST) with improved ordering. The consistency under random initialization is better than UMAP (Fig. 15(b,e)) which can be improved further using a modified attraction (Fig. 15(c,f)). Increasing repulsion by adding a small value to the repulsion shape increases compactness of the embedding (by increasing inter-cluster distance).

### G.3 PAIRWISE CONTROLLED MANIFOLD APPROXIMATION WITH LOCAL ADJUSTED GRAPH (LOCALMAP)

LocalMAP (Wang et al., 2025) is an iteration of the PaCMAP algorithm. One of the defining features of LocalMAP is the separation of all 10 clusters of the MNIST data (Fig. 16(a)), with behavior similar to the ones in Figs. 3(g,h,j,k) and Figs. 4(a,b). Here, we explore the interplay of attractive and repulsive forces on the compactness and connectedness of clusters.

LocalMAP performs PaCMAP and then does additional optimization on the attraction to decouple some clusters. To this end, it minimizes the following loss function

$$\mathcal{L}^L = \sum_{(i,j) \in NN} \frac{K}{\frac{1}{\sqrt{q_{ij}}} + C\sqrt{q_{ij}}} + \sum_{(i,l) \in FP} \frac{1}{1 + \frac{1}{q_{il}}}. \tag{66}$$

The first term amalgamates the attractive and repulsive nature of the triplet loss function that works on the same pair. In one regime, this function causes attraction, while in the other, it causes repulsion. The second term is identical to the ones in PaCMAP; the only difference is that the algorithm

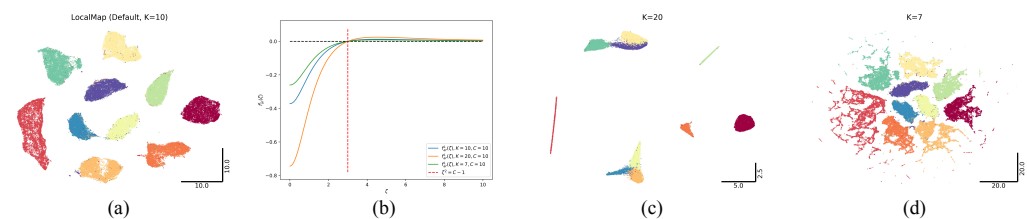

Figure 16: Behavior of LocalMAP on MNIST data. (a) Default embedding. (b) Attraction-repulsion shape of nearest neighbors. A value below (above) the dotted line indicates attraction (repulsion). Transition from attraction to repulsion occurs as $\zeta$ increases and crosses $\sqrt{C-1}$. Following implementation of LocalMAP, we used $C = 10$. (c) When $K$ is large ($= 20$), repulsion dominates and clusters become compact. (d) When $K$ is small ($= 7$), attraction dominates and clusters break up.

resamples further pairs every few iterations. Thus, we analyze only the first term involving nearest neighbors. The update equations are

$$y_i^{t+1} = y_i^t + \lambda f_{ar}^L(\zeta_1^t)(y_i^t - y_j^t), \tag{67}$$

$$y_j^{y+1} = y_j^t - \lambda f_{ar}^L(\zeta_1^t)(y_i^t - y_j^t), \tag{68}$$

where $f_{ar}^L$ is the attraction-repulsion shape given by

$$f_{ar}^L(\zeta) = -\frac{K(C - 1 - \zeta^2)}{2\sqrt{1 + \zeta^2}(1 + C + \zeta^2)^2}. \tag{69}$$

The update provides contraction as long as $\zeta^2 < C - 1$ and $-1 < \lambda f_{ar}^L < 0$ (from Proposition 4.1). When $\zeta^2 > C - 1$, $f_{ar} > 0$, and by Proposition 4.2 the update equations provide expansion. The values of $K$ and $C$ determine whether attraction or repulsion dominates the dynamics. In LocalMAP implementation, both $C$ and $K$ are set to 10 (Fig. 16(b), and the strength of attraction is higher than PaCMAP (Fig. 14(a)). As a result, when $\zeta > 3$, the nearest neighbors face repulsion, causing pairs bridging two clusters to separate. The value of $K$, working as a scaling parameter for the forces, regulates this separation.

Using Proposition 4.1, $\lambda f_{ar}^L(0) \geq -\frac{1}{2}$ gives the maximum values of $K$, and the maximum attraction possible by $f_{ar}^L$, without flipping the placements of the pairs. (We would want to avoid flipping the pairs at the LocalMAP optimization to preserve the ordering from PaCMAP; otherwise, it may inhibit cluster separation.) This simplifies (69) to $K \leq \frac{\lambda(1+C)^2}{C-1}$, which for $C = 10$ and $\lambda = 1$ gives $K \lesssim 13.44$. Moreover, at a higher value of $K \gtrsim 13.44$, the repulsive forces dominate, and the clusters become more compact, but objective of LocalMAP fails ( the bridge between clusters persists in Fig. 16(c) for $K = 20$). On the other hand, as $K$ decreases, attractive forces dominate (because repulsive forces are too low), and the embedding shows the breaking up of existing clusters (Fig. 16(d) for $K = 7$, which mimics the one in Fig. 4(b)).

### G.4 $t$-DISTRIBUTED STOCHASTIC NEIGHBOR EMBEDDING ($t$-SNE)

$t$-SNE (Van der Maaten & Hinton, 2008) optimizes pairwise distances. The loss function is

$$\mathcal{L} = -\sum_{i,j} w_{i,j} \log \left( \frac{q_{i,j}}{\sum_{k \neq l} q_{k,l}} \right), \tag{70}$$

where $w_{i,j}$ is the weight of the pair. The original implementation of $t$-SNE considers all the pairs (not just nearest neighbors). This loss function decomposes into attraction and repulsion forces by

$$\mathcal{L} = \sum_{i,j} \left[ -w_{i,j} \log(q_{i,j}) + w_{i,j} \log \left( \sum_{k \neq l} q_{k,l} \right) \right]. \tag{71}$$

As previously, the first term provides the attractive forces and the second term provides the repulsive forces. While the attractive term is identical to that of UMAP (with $a = 1$ and $b = 1$) and is easy to

compute, the repulsive term is coupled among every pair and thus, is very costly. Using the same principles we applied for UMAP, we can write the update equations of t-SNE. Since the original $t$-SNE didn't rely on the nearest neighbor graph, the weight $w_{i,j}$, computed for all the pairs, is important in the update equations. The attractive update equations are

$$y_i^{t+1} = y_i^t + \lambda w_{i,j} f_a^{t-SNE}(\zeta_{i,j}^t)(y_i^t - y_j^t), \tag{72}$$

$$y_j^{t+1} = y_i^t - \lambda w_{i,j} f_a^{t-SNE}(\zeta_{i,j}^t)(y_i^t - y_j^t), \tag{73}$$

where $f_a^{t-SNE}$ is the attraction shape of $t$-SNE and $\zeta_{i,j} = ||y_i - y_j||_2$. The update equation for the repulsive parts are

$$y_i^{t+1} = y_i^t + \lambda \frac{w_{i,j}}{Z} \sum_k f_r^{t-SNE}(\zeta_{i,k}^t)(y_i^t - y_k^t), \tag{74}$$

$$y_j^{t+1} = y_j^t - \lambda \frac{w_{i,j}}{Z} \sum_l f_r^{t-SNE}(\zeta_{l,j}^t)(y_l^t - y_j^t), \forall j, j \neq i, \tag{75}$$

where $f_r^{t-SNE}$ is the repulsion shape of $t$-SNE and $Z = \sum_{k \neq l} q_{k,l}$. The functional forms of these shapes are

$$f_a^{t-SNE}(\zeta) = -\frac{2}{1+\zeta^2}, \text{ and} \tag{76}$$

$$f_r^{t-SNE}(\zeta) = \frac{2}{(1+\zeta^2)^2}. \tag{77}$$

From the attractive update Eqs. (72)-(73), $f_a^{t-SNE}$ follows Proposition 4.1 (with $0 < \lambda w_{i,j} f_a^{t-SNE} < -1$) and gives the minimum distance for contraction, $\zeta_{-1}$. The repulsive update is coupled with all the pairs and thus does not have a simple relation to the repulsion shape. Rather, enabled by our experience from TriMap's analysis, we can derive the following:

**Proposition G.3.** *The update Eqs. (74)-(75) provide an expansion if*

$$h_r^{t-SNE}(\zeta_{i,j}, v, \theta, \lambda, w_{i,j}) > 1, \tag{78}$$

*where*
$h_r^{t-SNE}(\zeta_{i,j}, v, \theta, \lambda, w_{i,j}) = (1 + 2\lambda \frac{w_{i,j}}{Z} f_r^{t-SNE}(\zeta_{i,j}))^2 + \frac{||v||_2^2}{\zeta_{i,j}^2} + 2\lambda \frac{w_{i,j}}{Z} f_r^{t-SNE}(\zeta_{i,j}) \frac{||v||_2}{\zeta_{i,j}} \cos\theta$,

$v = \lambda \frac{w_{i,j}}{Z} \left( \sum_{k, k \neq j} f_r^{t-SNE}(\zeta_{i,k})(y_i - y_k) + \sum_{l, l \neq i} f_r^{t-SNE}(\zeta_{l,j})(y_l - y_j) \right)$,
*and $\theta$ is the angle between $(y_i - y_j)$ and $v$.*

*Proof.* We require,

$$||y_i^{t+1} - y_j^{t+1}||_2^2 > ||y_i^t - y_j^t||_2^2. \tag{79}$$

From Eqs. (74) and (75), $y_i - y_j = (1 + 2\lambda \frac{w_{i,j}}{Z} f_r^{t-SNE}(\zeta_{i,j}))(y_i - y_j) + v$. Putting this value in Eq. (79), we obtain the desired inequality. $\square$

Since $t$-SNE's condition for repulsion (Eq. 78) resembles that of TriMap, the repulsion behavior will be the same. Thus, $t$-SNE's repulsive forces can give both attraction and repulsion.

Since, $t$-SNE's forces are scaled (by $w_{i,j}$ for attraction and $\frac{w_{i,j}}{Z}$ for repulsion), the attractive and repulsive forces are typically lower than that of UMAP. As a result, the algorithm generally uses large values for learning rate (e.g., $\approx 10^3$ in the Open-$t$-SNE package (Poličar et al., 2024)). Moreover, most $t$-SNE implementations require an 'early exaggeration' step where the attractive forces are multiplied by a constant value for the first few iterations. This causes points that are supposed to be closer but currently placed far apart to approach each other (inducing far-sightedness). On the other hand, if some points are very close ($\zeta_{i,j} \approx \zeta_{-1}$) but require separation, this early exaggeration trick achieves that as well. Thus, 'early exaggeration' plays a vital role in finding a consistent embedding in $t$-SNE and is an indispensable feature of the algorithm; especially when initialized randomly. (This trick, when applied throughout the optimization, makes $t$-SNE embeddings look closer to UMAP embeddings (Böhm et al., 2022).)

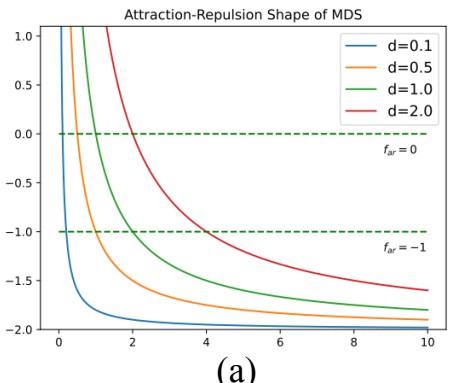 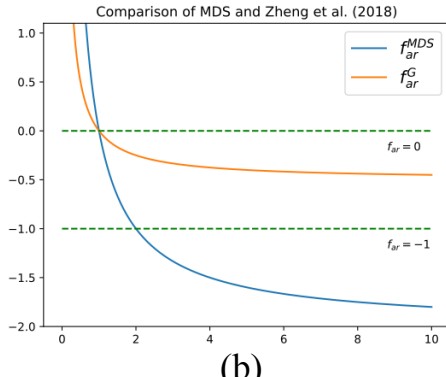

(a)    (b)

Figure 17: (a) Attraction-repulsion characteristic of MDS using $f_{ar}$. (b) Comparison of attraction-repulsion characteristic of MDS and the method in (Zheng et al., 2018) for $d = 1.0$.

### G.5 STOCHASTIC NEIGHBOR EMBEDDING (SNE)

SNE (Hinton & Roweis, 2002) is one of the oldest algorithms in this class. This follows the same formula of $t$-SNE (Eq. 71), but with $q_{i,j} = \exp(-||y_i - y_j||_2^2)$. This results in the attraction shape

$$f_a^{SNE}(\zeta) = -2, \tag{80}$$

that follows Proposition 4.1 (with $0 < \lambda w_{i,j} f_a^{SNE} < -1$) and the repulsion shape

$$f_r^{SNE}(\zeta) = 2\exp(-\zeta^2), \tag{81}$$

that follows proposition G.3 (by replacing the $t$-SNE symbols with SNE counterparts). The attraction shape is ill-posed and thus mainly relies on the values of learning rate ($\lambda$) and the weights ($w_{i,j}$) for contraction resulting in clusters overlapping each other even for small number of samples (called crowding problem (Van der Maaten & Hinton, 2008)), which $t$-SNE and the subsequent algorithms improve by moving away from the Gaussian kernel to a heavy-tailed one, i.e., Eq. (43).

### G.6 MULTIDIMENSIONAL SCALING (MDS)

Multidimensional scaling (MDS) (Borg & Groenen, 2007) typically does not use gradient methods, as they often fail to converge to good mappings; instead, it employs stress majorization. Nevertheless, few works discuss gradient methods (Kruskal, 1964; Zheng et al., 2018). We offer a brief treatment for this below. Particularly, Zheng et al. (2018) formulates a successful gradient descent-based MDS algorithm for graph drawing. We start with the cost function (we can ignore the weight $w_{i,j}$ without loss of generality to their approach, and for discussion, we can lump it into the learning rate):

$$\mathcal{L}^{MDS} = \sum_{i,j,i<j} (||y_i - y_j|| - d_{ij})^2. \tag{82}$$

The loss has a singleton term for each pair and does not have explicit attractive and repulsive terms. The term itself will provide both attraction and repulsion. Fortunately, the loss is separable into individual $(i, j)$ components, allowing us to analyze a single pair. The derivative is given by

$$\frac{\partial}{\partial y_i}(||y_i - y_j|| - d_{ij})^2 = 2\frac{||y_i - y_j|| - d_{ij}}{||y_i - y_j||}(y_i - y_j). \tag{83}$$

Thus, the attraction-repulsion shape of MDS becomes

$$f_{ar}^{MDS}(\zeta) = -2\frac{\zeta - d}{\zeta}, \tag{84}$$

where $\zeta = ||y_i - y_j||_2$. Here, the shape is attractive when $\zeta > d$. In this range, the shape is bounded within $[-2, 0]$ (Fig. 17(a)). However, this is not suitable for convergence as values only within $[-1, 0]$

contract, while the others expand. To make things worse, values lower than $-2$ cause the points to flip and expand. To work around this, Zheng et al. (2018) uses an ad-hoc formulation inspired by the force-directed graphs, given by

$$f_{ar}^G(\zeta) = -\frac{1}{2}\frac{\zeta - d}{\zeta}, \tag{85}$$

which works for an effective learning rate $\lambda \leq 1$. $f_{ar}^G$ is bounded within $[-0.5, 0]$ and thus, $\lambda \leq 1$ works. On the other hand, for $\zeta < d$, $f_{ar}^{MDS}(> 0)$ shows repulsive behavior. Overall, this attraction-repulsion interaction works best when the initialization is already close to a desired output. If one keeps optimizing for a pair, it will oscillate around the distance ($\zeta = d$) where $f_{ar}^G = 0$ (and consequently, we can have a notion of distance similar to $\zeta_{-1}$ of UMAP). No choice of learning rate reduces this distance to zero (in fact, this achieves the objective of MDS). Thus, the clusters are often fuzzy compared to methods like UMAP and PaCMAP (for relevant illustrations, see Lambert et al. (2022), de Bodt et al. (2025), and Kury et al. (2025)). Note that Lambert et al. (2022) converts the MDS loss function to a quartet stress (loss involving four samples) and a relative distance (distance divided by the sum of six distances in the quartet), enabling global regularization and reduced computation than the majorization approach.

## H    CONSTRUCTION OF THE HIGH-DIMENSIONAL GRAPH

Stochastic Neighbor Embedding (SNE) Hinton & Roweis (2002) underpins modern dimensionality reduction algorithms. It constructs a high-dimensional graph of the dataset $X = \{x_i \in R^n | i = 1, \ldots, N\}$ by the following system of equations:

$$w_{ij} = \frac{w_{j|i} + w_{i|j}}{2N}, \tag{86}$$

$$w_{j|i} = \frac{\exp(-||x_i - x_j||_2^2/2\sigma_i^2)}{\sum_{t \neq v}\exp(-||x_t - x_v||_2^2/2\sigma_t^2)}, \tag{87}$$

where $\sigma_i^2$ is chosen to match a user-defined value *perplexity*, $P$, defined as

$$P = 2^{H_i} \tag{88}$$

$$H_i = \sum_{j \neq i} w_{j|i} \log_2 w_{j|i}. \tag{89}$$

On the other hand, UMAP constructs its high-dimensional graph by the following system of equations relying on the k-nearest neighbor (k-NN) algorithm:

$$p_{i|j} = \begin{cases} \exp\left(-\frac{d(x_i, x_j) - \rho_i}{\sigma_i}\right) & \text{if } x_j \in \text{KNN}(x_i, k) \\ 0 & \text{otherwise} \end{cases}, \tag{90}$$

$$\rho_i = \min_{x_j \in \text{KNN}(x_i, k)} d(x_i, x_j), \tag{91}$$

where $\text{KNN}(x_i, k)$ is the set of $k$-nearest neighbors of the point $x_i$ and $\sigma_i$ is a scaling parameter such that $\sum_j p_{i|j} = \log_2(k)$. The graph is then symmetrized by a t-conorm:

$$p_{i,j} = p_{i|j} + p_{j|i} - p_{i|j}p_{j|i}. \tag{92}$$

PaCMAP uses just the k-NN graph for the affinities (all equal to 1) with a self-tuning distance measure (Zelnik-Manor & Perona, 2004),

$$d_{i,j}^2 = \frac{||x_i - x_j||^2}{\sigma_i \sigma_j}, \tag{93}$$

where $\sigma_i$ is the average distance between $x_i$ and its Euclidean nearest fourth to sixth neighbors. The purpose of this is the same as the corresponding $\sigma_i$ parameters in Eq. (87) and (90), despite all three being defined differently.

Regardless of these choices, the optimization, and by our analysis, the attraction and repulsion shapes are the primary drivers of the low-dimensional embedding.

# I   DETAILED RESULTS FOR VARYING $\lambda_a$ AND $\lambda_b$

In this section, we provide detailed results for the experiments in Fig. 1((h,i). We obtained these results, by changing the attraction and repulsion shapes of UMAP to that of the other methods. Figure 18 reproduces the results given in the main text. Figures 19-22 show the individual embeddings for each of the choices of $\lambda_a$ and $\lambda_r$ for each methods along with their initialization, a reference embedding when learning rate, $\lambda$, is annealed, and when either the $\lambda_a$ or $\lambda_r$ set to 0. For PaCMAP, which uses the concept of mid-near pairs, we show additional reference of mid-near pairs included as well. Figures 23- 27 and  Figures. 28- 32 provide results on FMNIST and single-cell transcriptomes dataset, respectively.

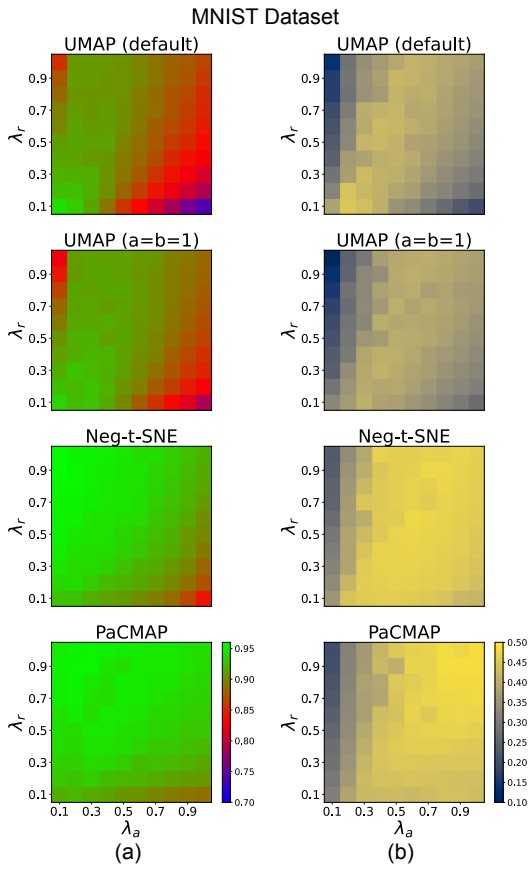

Figure 18: (a) Trustworthiness and (b) Silhouette score of different methods for the MNIST dataset.

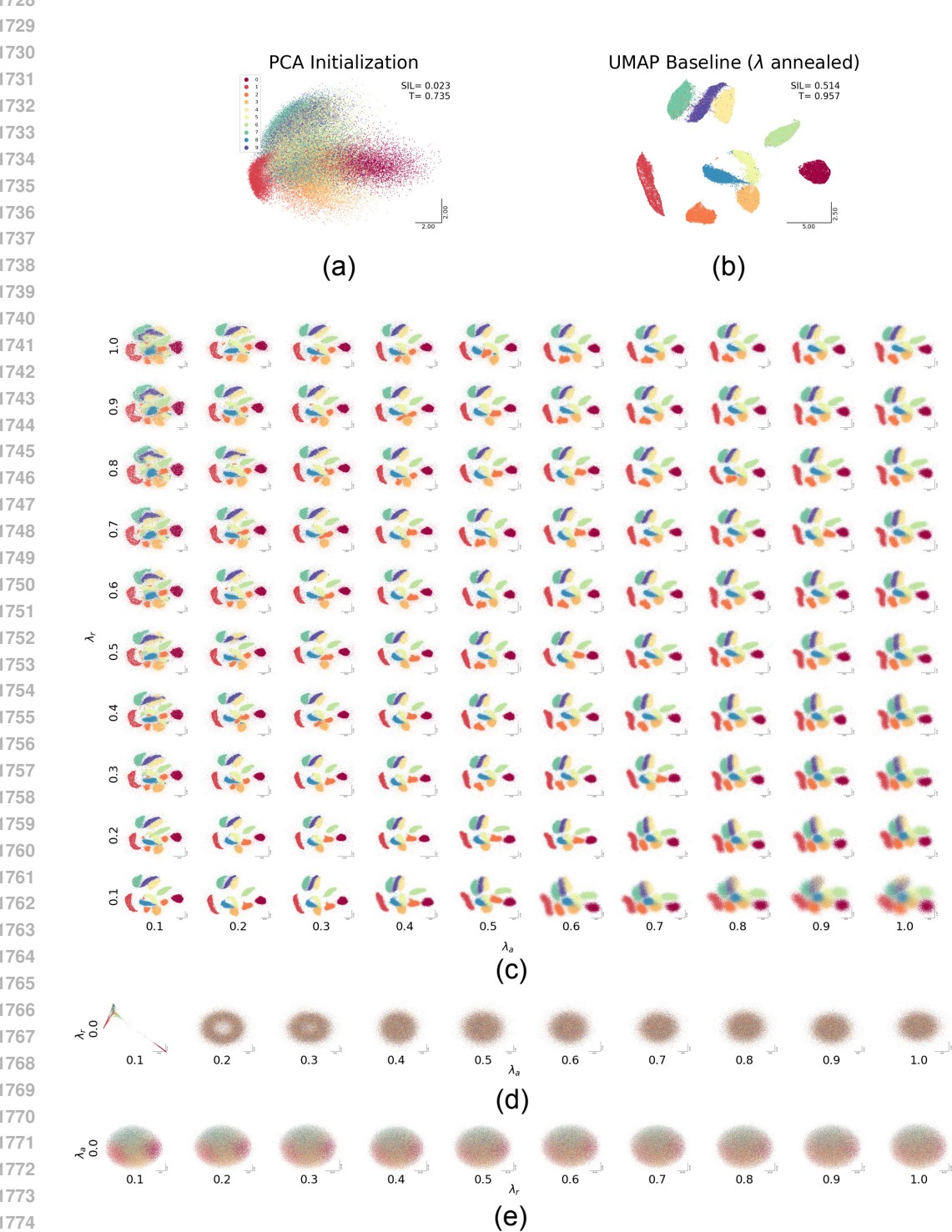

Figure 19: Varying $\lambda_a$ and $\lambda_r$ for the MNIST dataset using UMAP's attraction and repulsion shapes. (a) Initialization for the embeddings. (b) Baseline when $\lambda$ is annealed from 1. (c) The embeddings when $\lambda_a$ and $\lambda_r$ vary (without any annealing). (d) When $\lambda_r$ is set to 0 (no repulsion), the attractive force alone cannot produce any cluster. (e) Similarly, when $\lambda_a$ is set to 0 (no attraction), the repulsive force alone cannot produce any clusters.

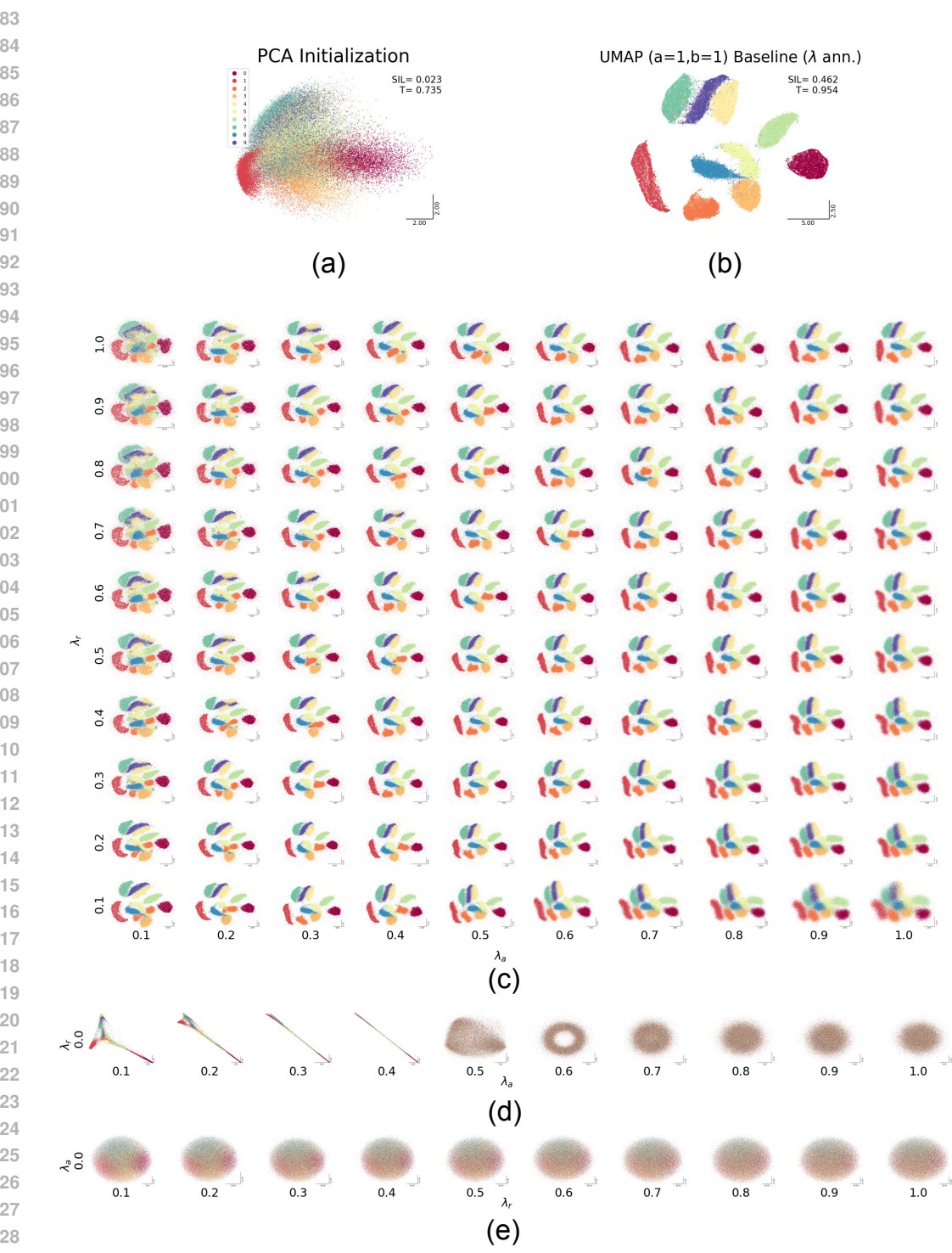

Figure 20: Varying $\lambda_a$ and $\lambda_r$ for the MNIST dataset using UMAP's attraction and repulsion shapes (by setting $a = 1$ and $b = 1$). (a) Initialization for the embeddings. (b) Baseline when $\lambda$ is annealed from 1. (c) The embeddings when $\lambda_a$ and $\lambda_r$ vary (without any annealing). (d) When $\lambda_r$ is set to 0 (no repulsion), the attractive force alone cannot produce any cluster. (e) Similarly, when $\lambda_a$ is set to 0 (no attraction), the repulsive force alone cannot produce any clusters.

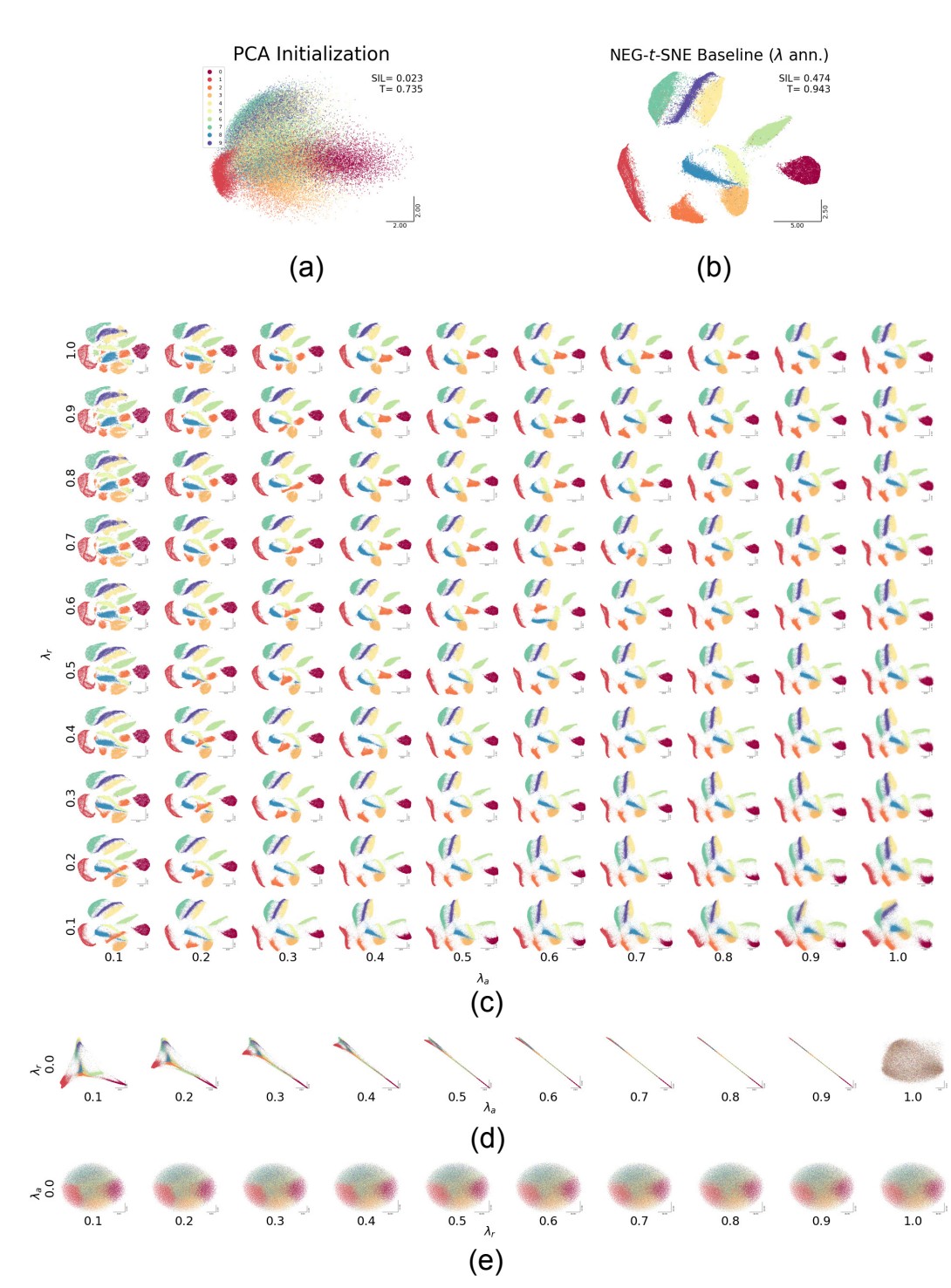

Figure 21: Varying $\lambda_a$ and $\lambda_r$ for the MNIST dataset using NEG-$t$-SNE's attraction and repulsion shapes. (a) Initialization for the embeddings. (b) Baseline when $\lambda$ is annealed from 1. (c) The embeddings when $\lambda_a$ and $\lambda_r$ vary (without any annealing). (d) When $\lambda_r$ is set to 0 (no repulsion), the attractive force alone cannot produce any cluster. (e) Similarly, when $\lambda_a$ is set to 0 (no attraction), the repulsive force alone cannot produce any clusters.

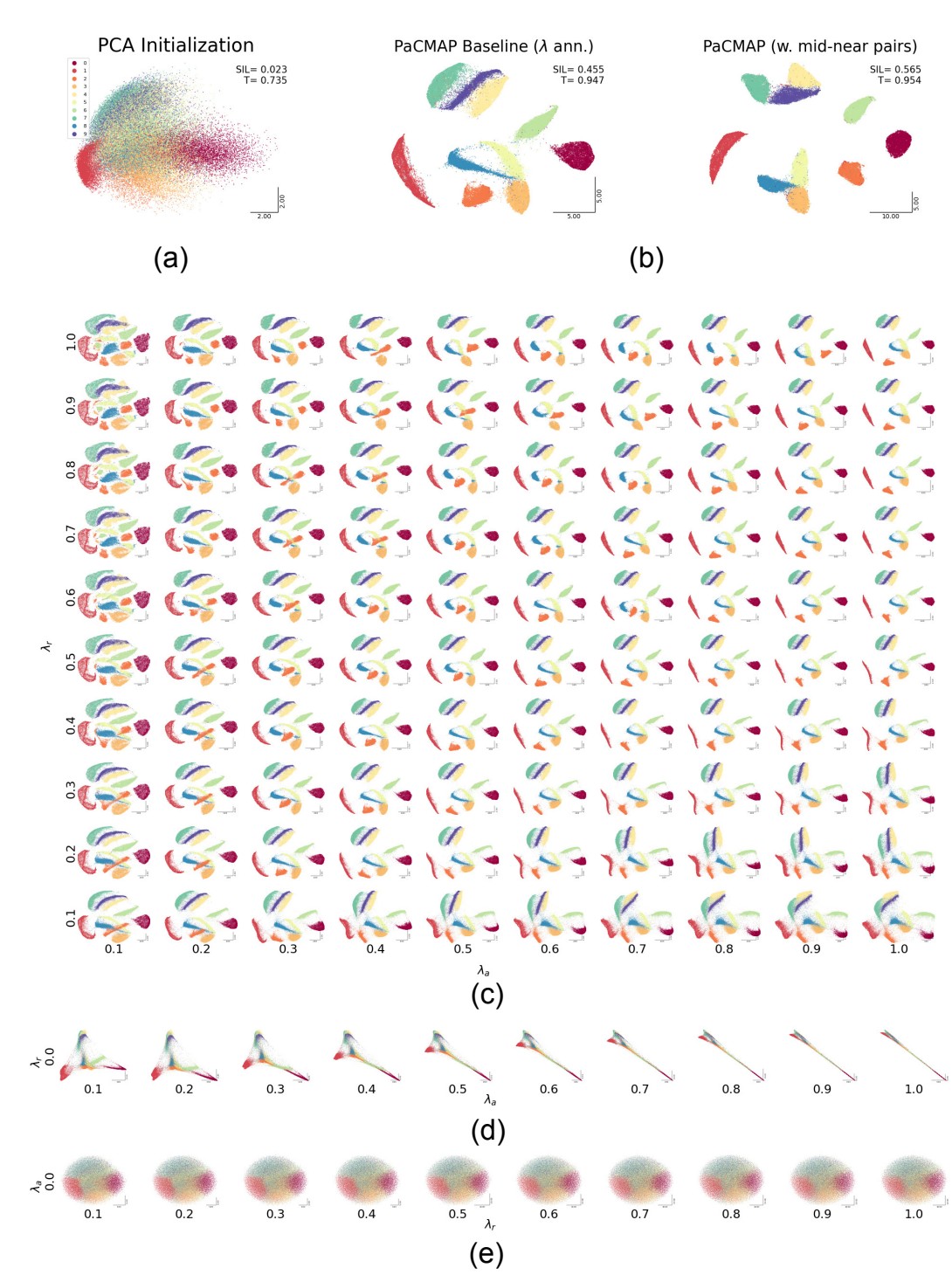

Figure 22: Varying $\lambda_a$ and $\lambda_r$ for the MNIST dataset using PaCMAP's attraction and repulsion shapes. (a) Initialization for the embeddings. (b) Left: Baseline when $\lambda$ is annealed from 1 (mid-near points are excluded to observe the interaction of attraction-repulsion alone), right: when mid-near points are considered. (c) The embeddings when $\lambda_a$ and $\lambda_r$ vary (without any annealing). (d) When $\lambda_r$ is set to 0 (no repulsion), the attractive force alone cannot produce any cluster. (e) Similarly, when $\lambda_a$ is set to 0 (no attraction), the repulsive force alone cannot produce any clusters.

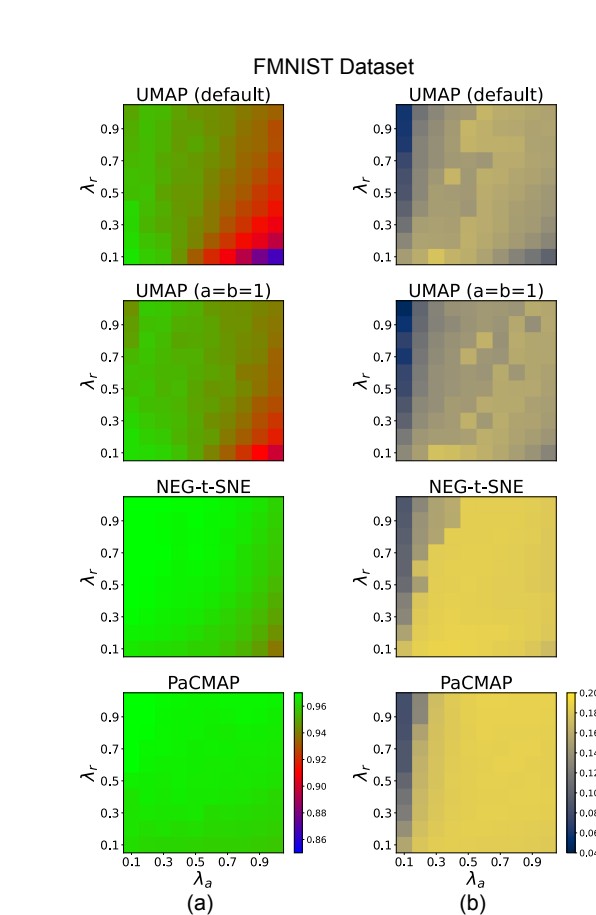

Figure 23: (a) Trustworthiness and (b) Silhouette score of different methods for the FMNIST dataset.

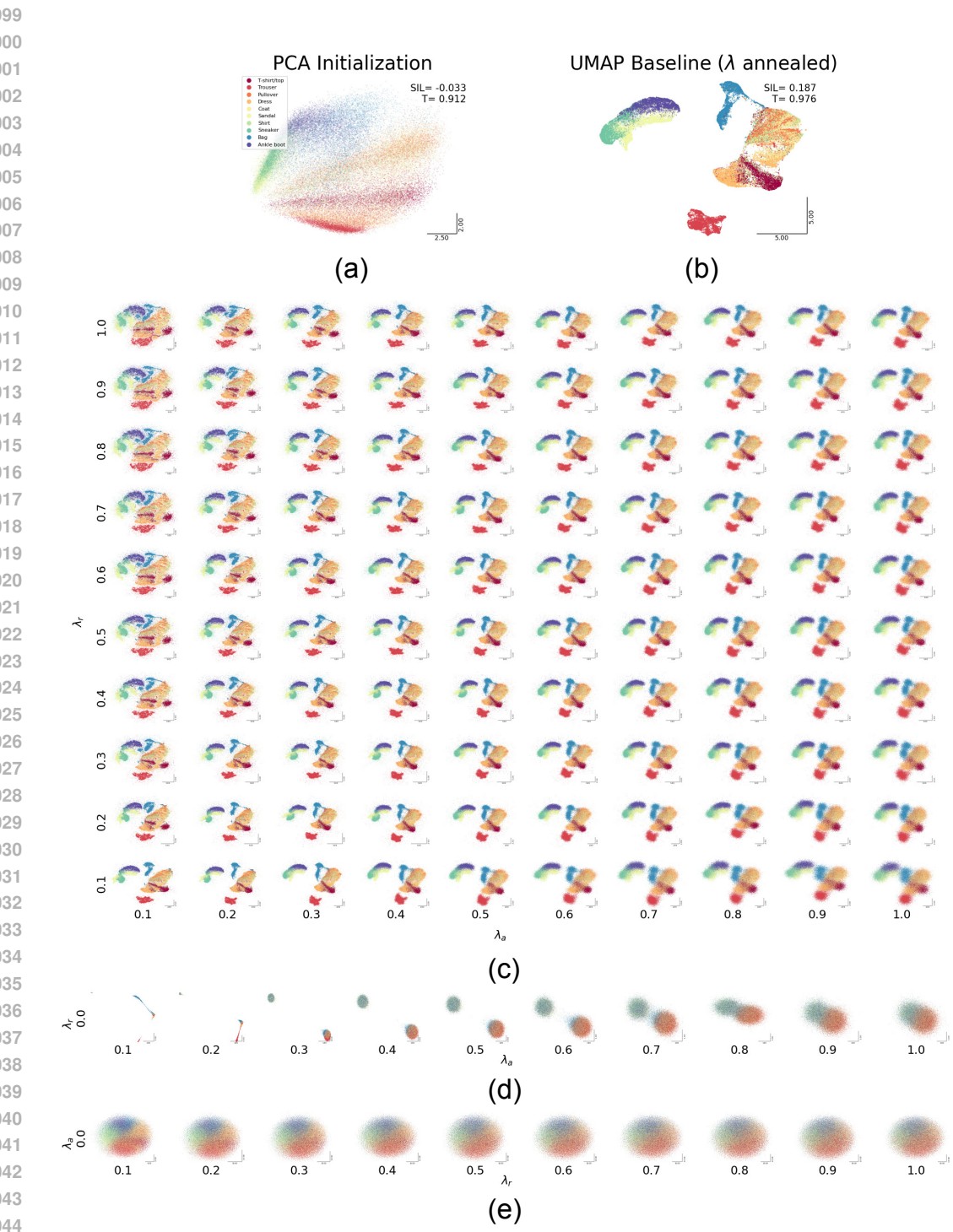

Figure 24: Varying $\lambda_a$ and $\lambda_r$ for the FMNIST dataset using UMAP's attraction and repulsion shapes. (a) Initialization for the embeddings. (b) Baseline when $\lambda$ is annealed from 1. (c) The embeddings when $\lambda_a$ and $\lambda_r$ vary (without any annealing). (d) When $\lambda_r$ is set to 0 (no repulsion), the attractive force alone cannot produce any cluster. (e) Similarly, when $\lambda_a$ is set to 0 (no attraction), the repulsive force alone cannot produce any clusters.

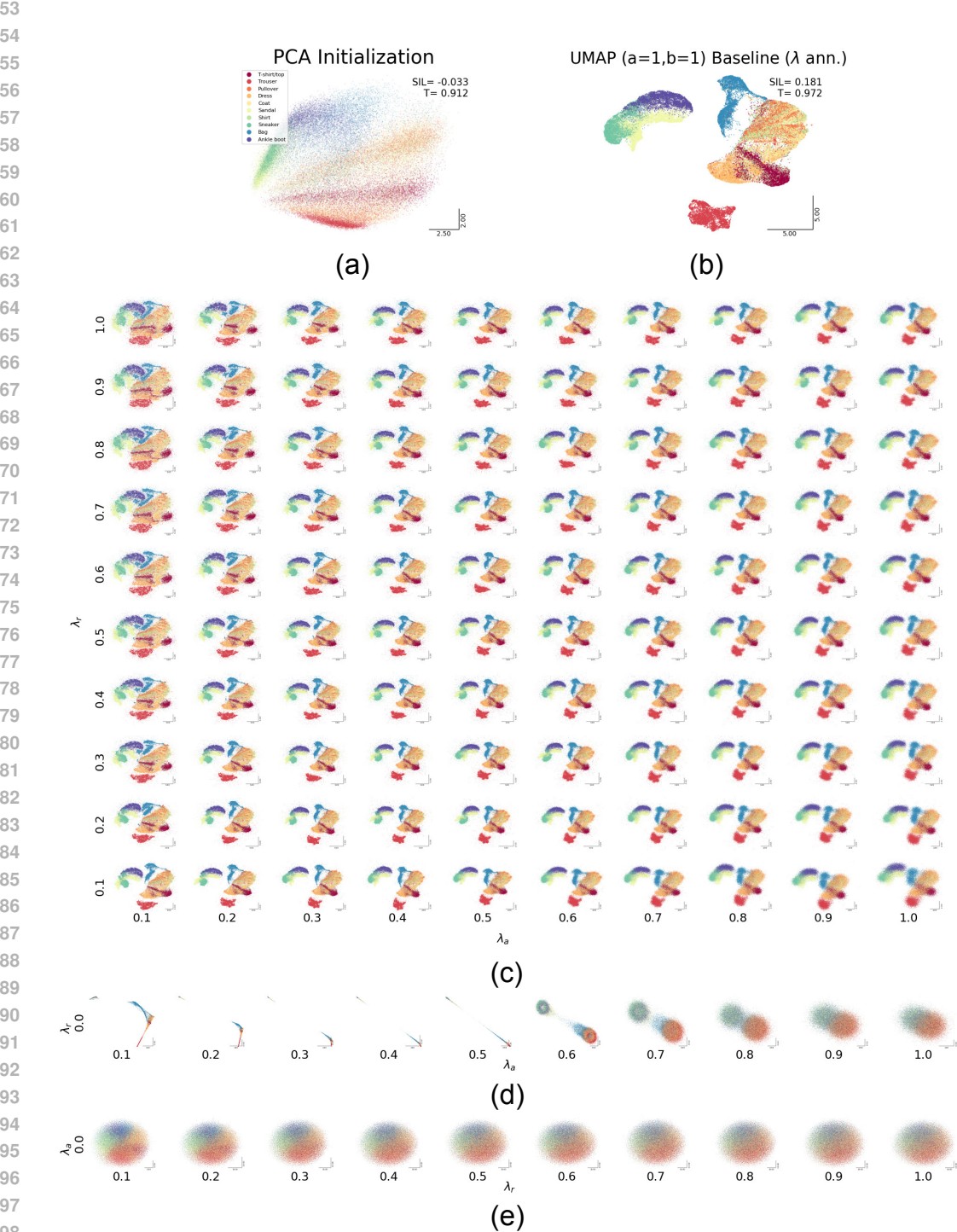

Figure 25: Varying $\lambda_a$ and $\lambda_r$ for the FMNIST dataset using UMAP's attraction and repulsion shapes (by setting $a = 1$ and $b = 1$). (a) Initialization for the embeddings. (b) Baseline when $\lambda$ is annealed from 1. (c) The embeddings when $\lambda_a$ and $\lambda_r$ vary (without any annealing). (d) When $\lambda_r$ is set to 0 (no repulsion), the attractive force alone cannot produce any cluster. (e) Similarly, when $\lambda_a$ is set to 0 (no attraction), the repulsive force alone cannot produce any clusters.

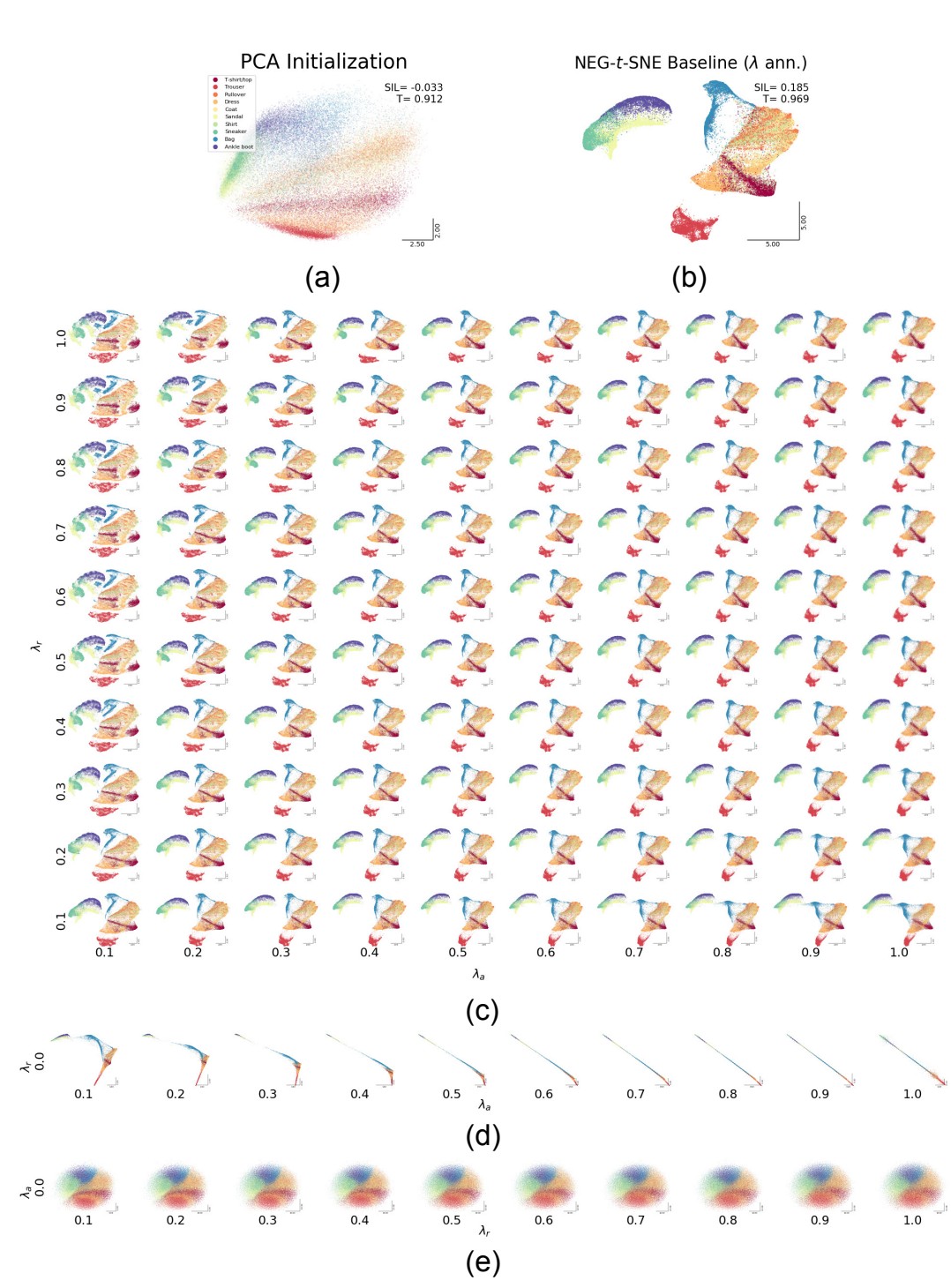

Figure 26: Varying $\lambda_a$ and $\lambda_r$ for the FMNIST dataset using NEG-$t$-SNE's attraction and repulsion shapes. (a) Initialization for the embeddings. (b) Baseline when $\lambda$ is annealed from 1. (c) The embeddings when $\lambda_a$ and $\lambda_r$ vary (without any annealing). (d) When $\lambda_r$ is set to 0 (no repulsion), the attractive force alone cannot produce any cluster. (e) Similarly, when $\lambda_a$ is set to 0 (no attraction), the repulsive force alone cannot produce any clusters.

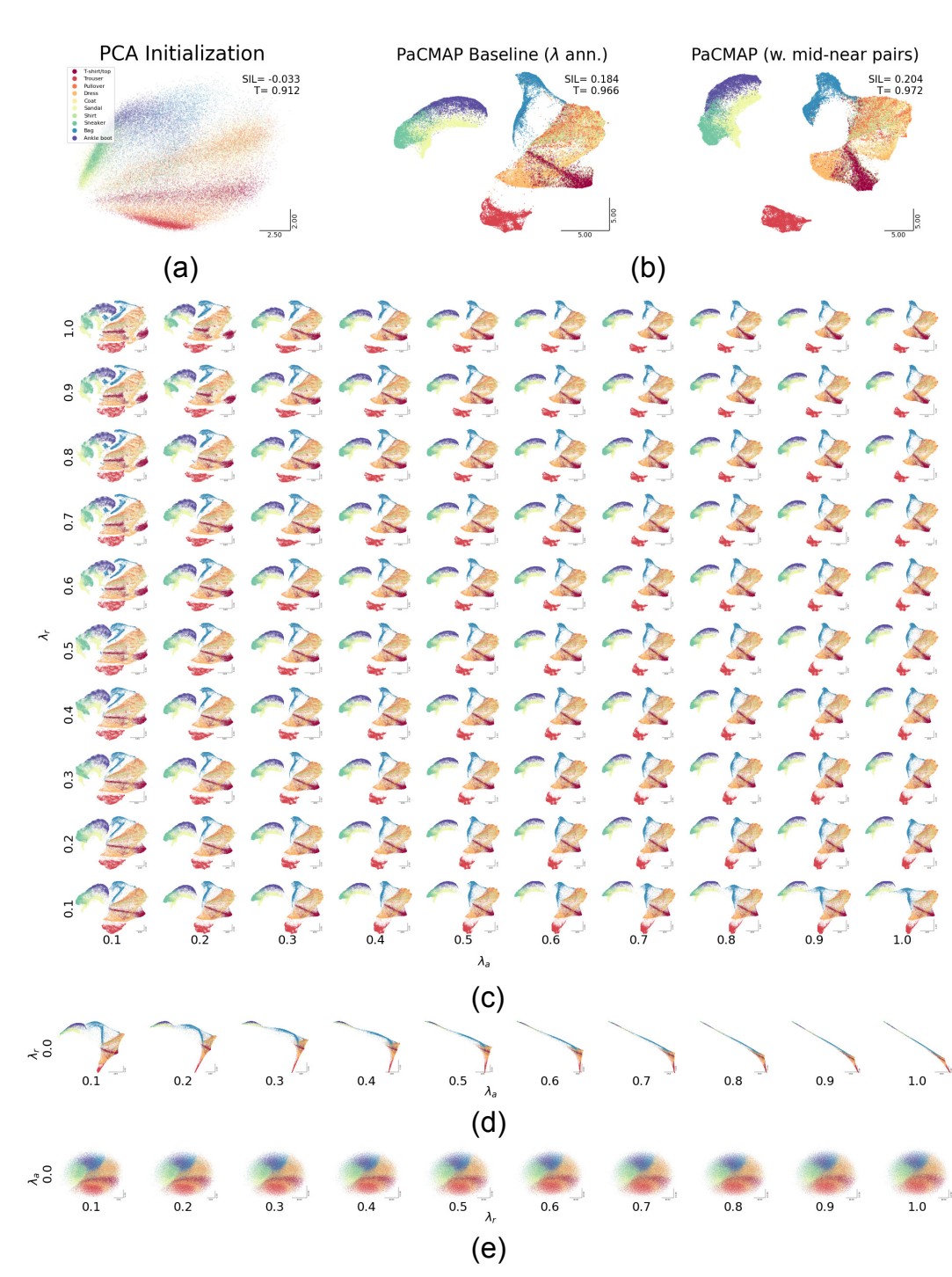

Figure 27: Varying $\lambda_a$ and $\lambda_r$ for the FMNIST dataset using PaCMAP's attraction and repulsion shapes. (a) Initialization for the embeddings. (b) Left: Baseline when $\lambda$ is annealed from 1 (mid-near points are excluded to observe the interaction of attraction-repulsion alone), right: when mid-near points are considered. (c) The embeddings when $\lambda_a$ and $\lambda_r$ vary (without any annealing). (d) When $\lambda_r$ is set to 0 (no repulsion), the attractive force alone cannot produce any cluster. (e) Similarly, when $\lambda_a$ is set to 0 (no attraction), the repulsive force alone cannot produce any clusters.

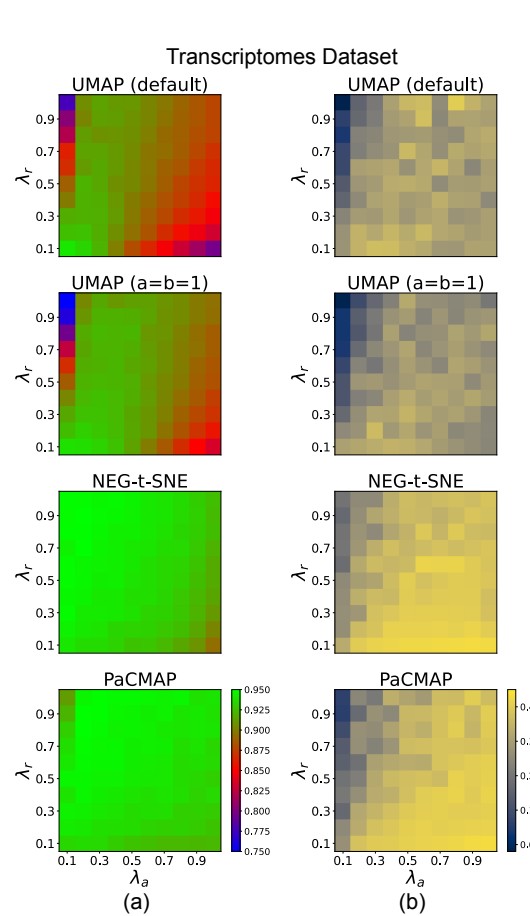

Figure 28: (a) Trustworthiness and (b) Silhouette score of different methods for the Single-cell transcriptomes dataset.

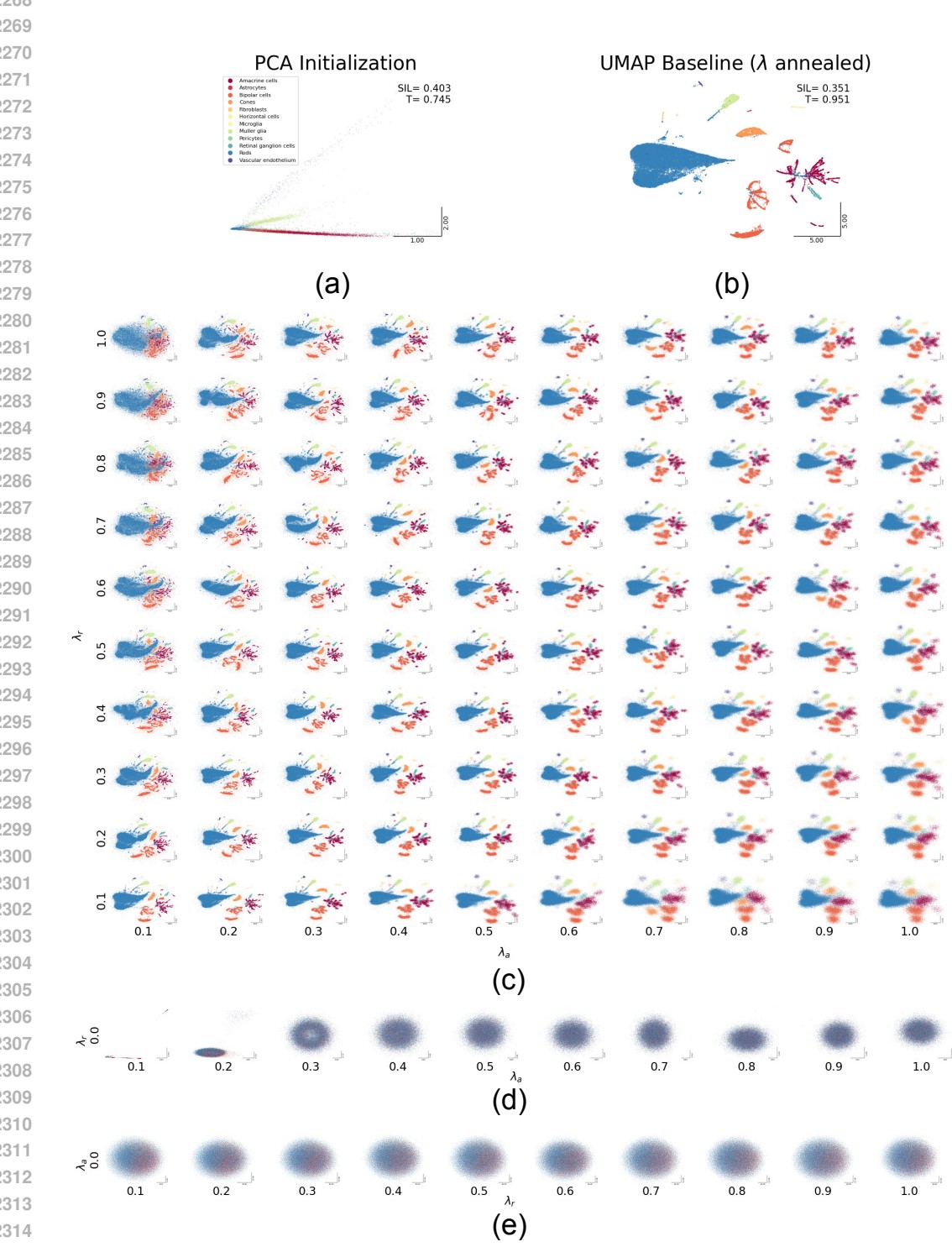

Figure 29: Varying $\lambda_a$ and $\lambda_r$ for the single-cell transcriptomes dataset using UMAP's attraction and repulsion shapes. (a) Initialization for the embeddings. (b) Baseline when $\lambda$ is annealed from 1. (c) The embeddings when $\lambda_a$ and $\lambda_r$ vary (without any annealing). (d) When $\lambda_r$ is set to 0 (no repulsion), the attractive force alone cannot produce any cluster. (e) Similarly, when $\lambda_a$ is set to 0 (no attraction), the repulsive force alone cannot produce any clusters.

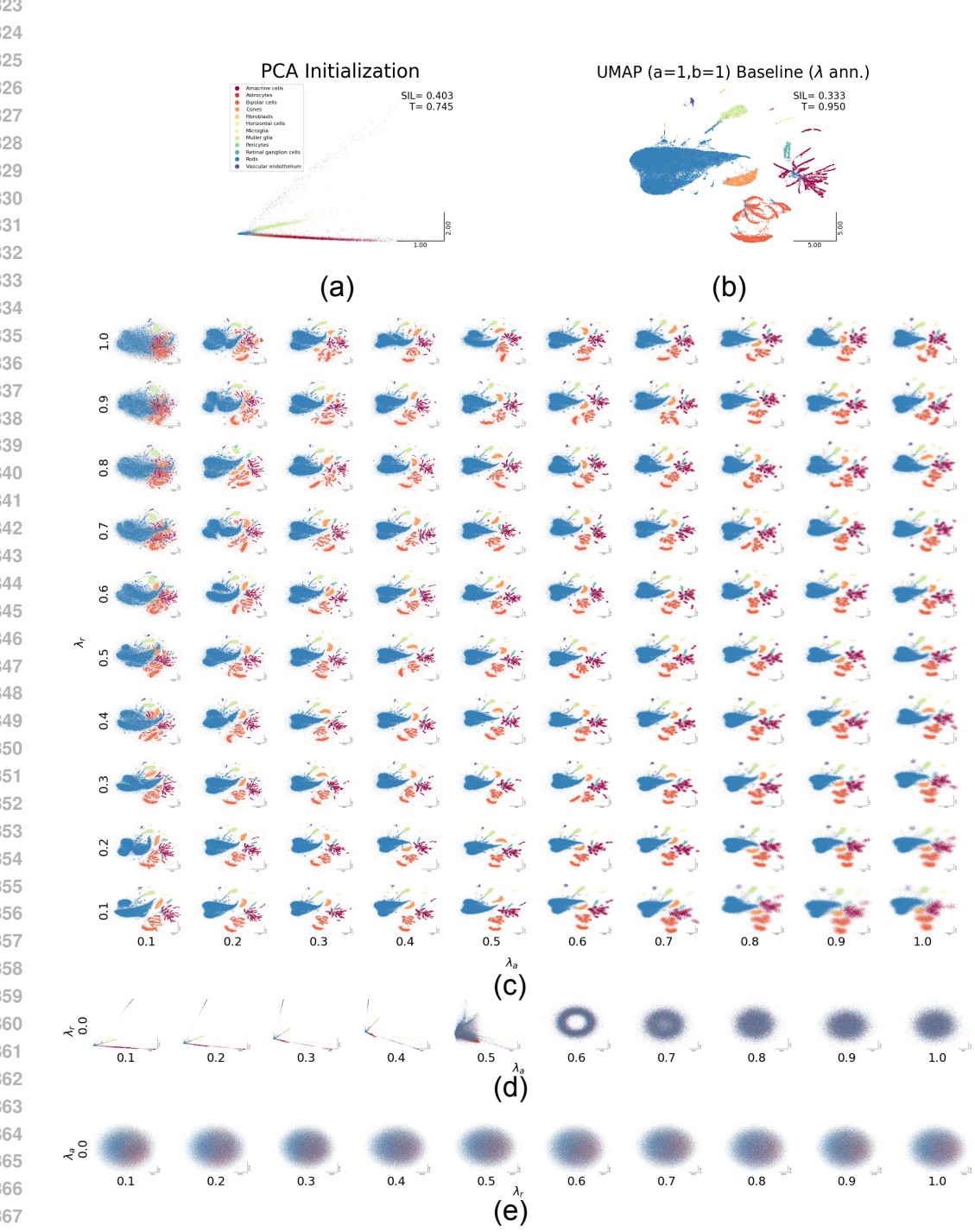

Figure 30: Varying $\lambda_a$ and $\lambda_r$ for the single-cell transcriptomes dataset using UMAP's attraction and repulsion shapes (by setting $a = 1$ and $b = 1$). (a) Initialization for the embeddings. (b) Baseline when $\lambda$ is annealed from 1. (c) The embeddings when $\lambda_a$ and $\lambda_r$ vary (without any annealing). (d) When $\lambda_r$ is set to 0 (no repulsion), attractive force alone cannot produce any cluster. (e) Similarly, when $\lambda_a$ is set to 0 (no attraction), repulsive force alone cannot produce any clusters.

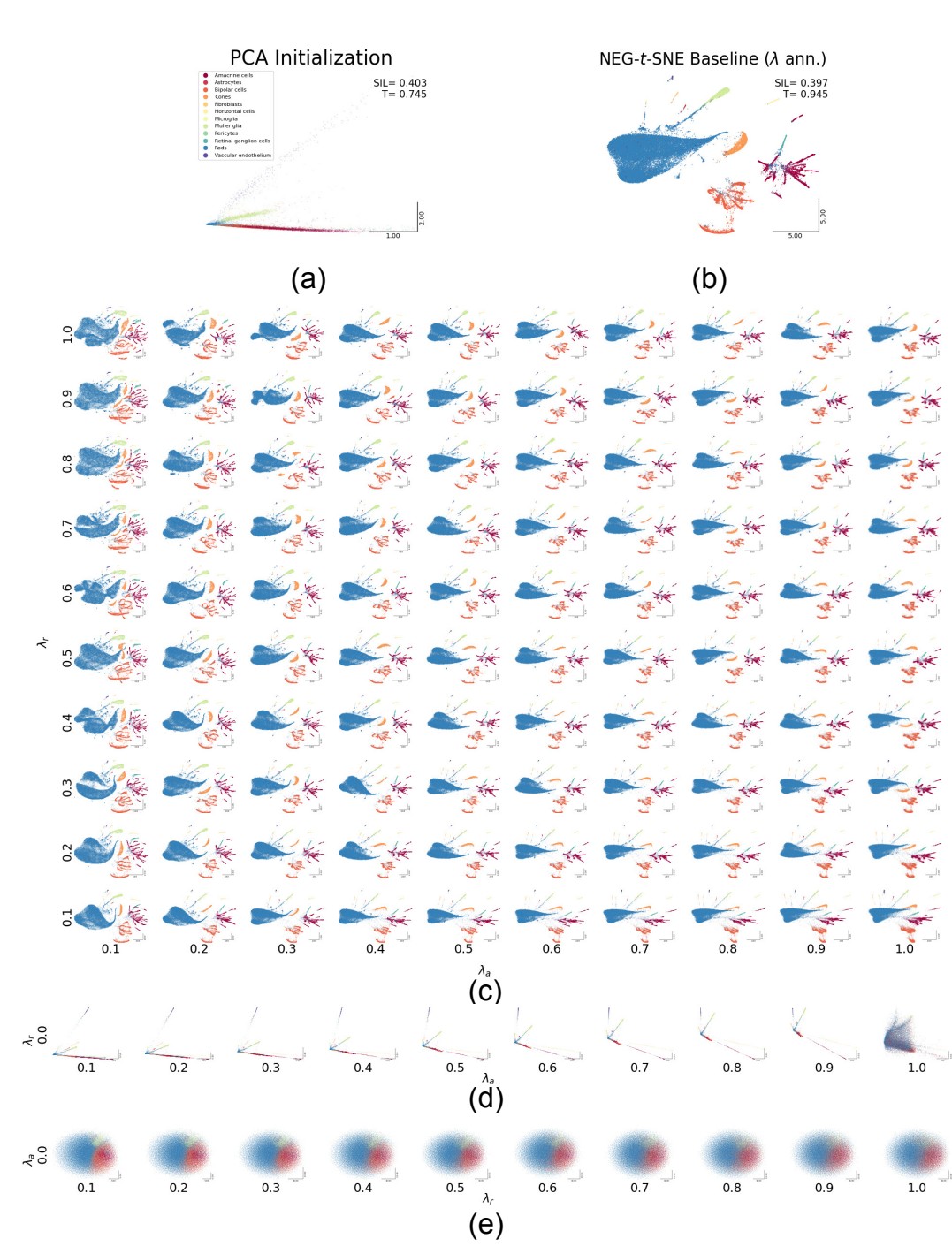

Figure 31: Varying $\lambda_a$ and $\lambda_r$ for the single-cell transcriptomes dataset using NEG-$t$-SNE's attraction and repulsion shapes. (a) Initialization for the embeddings. (b) Baseline when $\lambda$ is annealed from 1. (c) The embeddings when $\lambda_a$ and $\lambda_r$ vary (without any annealing). (d) When $\lambda_r$ is set to 0 (no repulsion), the attractive force alone cannot produce distinct clusters. (e) Similarly, when $\lambda_a$ is set to 0 (no attraction), the repulsive force alone cannot produce any clusters.

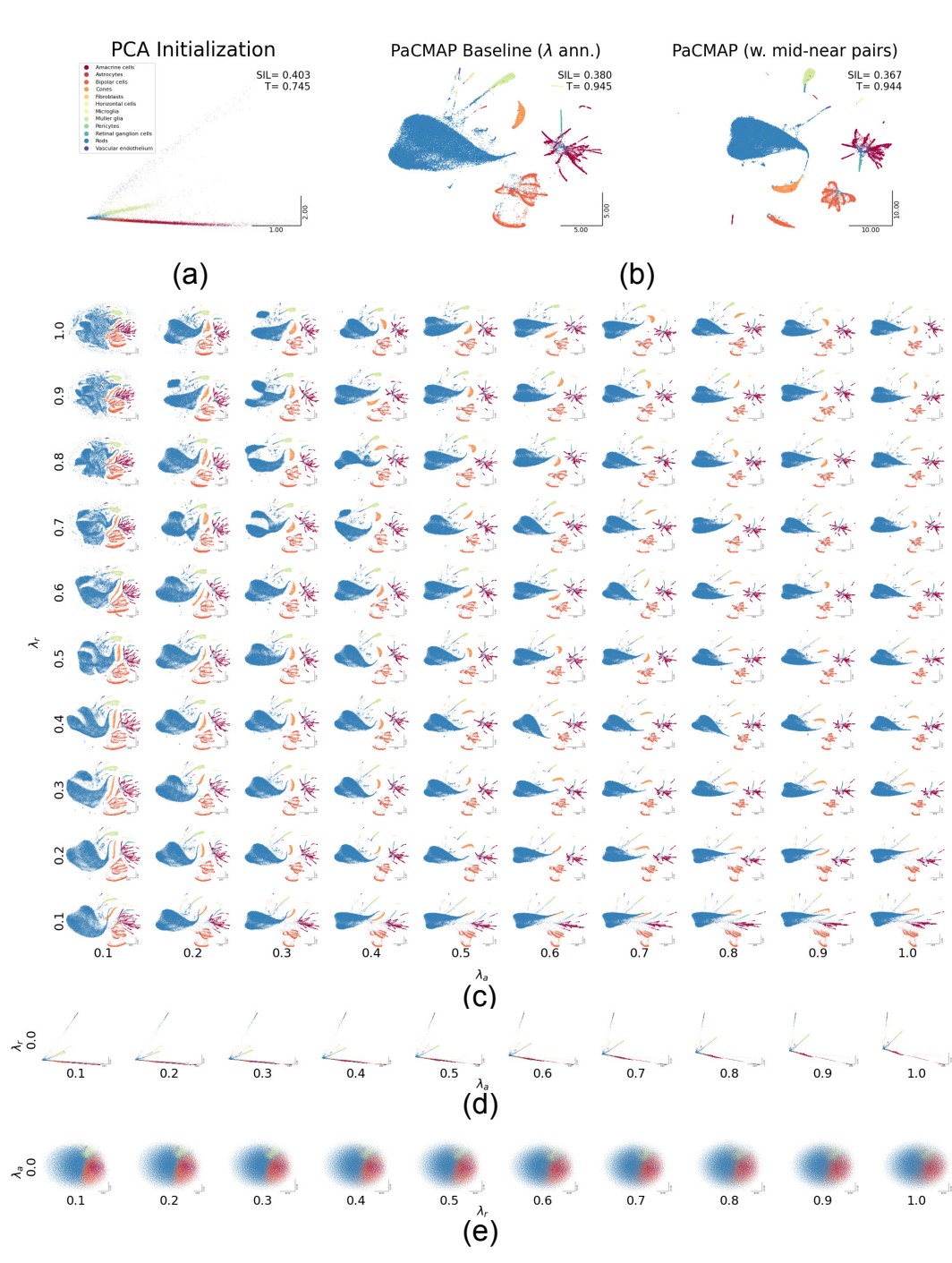

Figure 32: Varying $\lambda_a$ and $\lambda_r$ for the single-cell transcriptomes dataset using PaCMAP's attraction and repulsion shapes. (a) Initialization for the embeddings. (b) Left: Baseline when $\lambda$ is annealed from 1 (mid-near points are excluded to observe the interaction of attraction-repulsion alone), right: when mid-near points are considered. (c) The embeddings when $\lambda_a$ and $\lambda_r$ vary (without any annealing). (d) When $\lambda_r$ is set to 0 (no repulsion), the attractive force alone cannot produce any cluster. (e) Similarly, when $\lambda_a$ is set to 0 (no attraction), the repulsive force alone cannot produce any clusters.

## J   MIX AND MATCH APPROACH

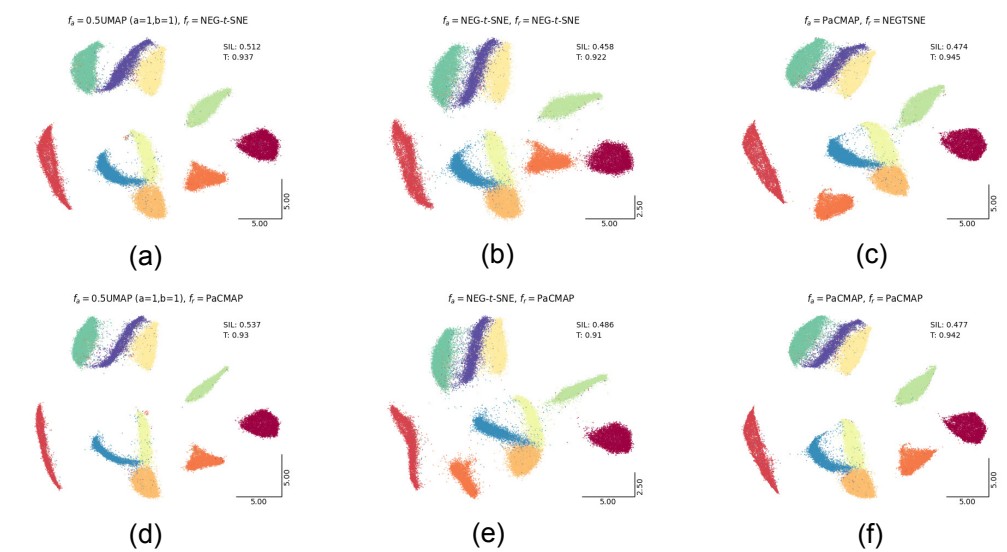

(a)        (b)        (c)

(d)        (e)        (f)

Figure 33: Embedding of MNIST by mixing and matching attraction and repulsion shapes with a constant learning rate ($\lambda = 1$). We used attraction shapes that follow Proposition 4.1 and repulsion shapes that are finite. Top row: UMAP, NEG-$t$-SNE, and PaCMAP's attraction shapes with NEG-$t$-SNE's repulsion shape. Bottom row: UMAP, NEG-$t$-SNE, and PaCMAP's attraction shapes with PaCMAP's repulsion shape.

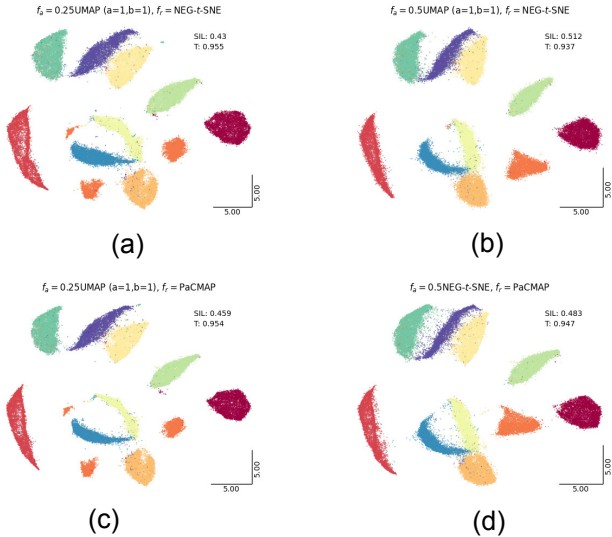

(a)        (b)

(c)        (d)

Figure 34: Embedding of MNIST by mixing and matching attraction and repulsion shapes with a constant learning rate ($\lambda = 1$). We kept the attraction shape confined to $[-0.5, 0]$ to prevent any flips during the attractive update. Top row: UMAP and NEG-$t$-SNE's attraction shapes with NEG-$t$-SNE's repulsion shape. Bottom row: UMAP and NEG-$t$-SNE's attraction shapes with PaCMAP's repulsion shape.

In our analysis, we considered the attraction and the repulsion shapes to be independent of each other. It thus suggests that we can combine attraction and repulsion shapes derived from different loss functions and principles. Figure 33 shows a few such examples. For this, we restricted the attraction shapes to $[-1, 0]$ (to work with Proposition 4.1) so that they become less sensitive to learning rates.

We considered finite repulsion shapes, hence mitigating the effect of large repulsive forces for smaller distances. We held the learning rate constant ($\lambda = 1$) and used the same optimization (of UMAP) for all of them, so that any changes arise from the shapes rather than learning-rate scheduling or optimization scheme. Overall, UMAP's attraction shape (by $f_a = 0.5 f_a^U$) makes better clusters (higher silhouette score), while PaCMAP's attraction shape provides better local structure (higher trustworthiness). The latter is primarily due to PaCMAP's attraction shape being within $[-0.5, 0]$, which prevents any flips during the updates. By confining UMAP and NEG-$t$-SNE 's attraction shape within $[-0.5, 0]$ (by $f_a = 0.25 f_a^U$ and $f_a = 0.5 f_a^N$, respectively), we can improve the local structure at the expense of clustering performance (Fig. 34).

## K  Implementation Details

For analysis, we implemented our own UMAP algorithm. We used `numba` (Lam et al., 2015) to compute an exact nearest neighbor graph (instead of an approximate one) with $k = 15$ and `scikit-learn`'s (Pedregosa et al., 2011) implementation of the PCA algorithm for PCA initialization. We used this to produce and quantify the embeddings in Figs. 1, 2, 3, 4, 6, 8, 9, and 10. We also used the same implementation when we changed the attraction and the repulsion shapes to those of the alternative methods (Figs. 18- 34, unless otherwise stated). The trustworthiness and silhouette scores were computed using the corresponding function from the `scikit-learn` package.

The mappings shown in Figs. 2, 3, 9, 10, and 15 are rotated to a reference embedding ((a) for each respective Figures). To achieve this, we performed Procrustes alignment of the embeddings by normalizing them (zero mean and unit norm) and then using SciPy's (Virtanen et al., 2020) `orthogonal_procrustes` method to extract rotation and scaling parameters.

To compare with Neg-$t$-SNE in Fig. 11, we used the original implementation of the contrastive embedding framework for both the UMAP and the Neg-$t$-SNE algorithms (available at https://github.com/berenslab/contrastive-ne).

PaCMAP and LocalMAP embeddings in Figs. 15, 16, 22 (b) (right), 27 (b) (right), and 32 (b) (right) were obtained using the official PaCMAP package (available at https://github.com/YingfanWang/PaCMAP).

The codes for displaying the embeddings of the PubMed dataset in Fig. 7 are adapted from https://github.com/berenslab/pubmed-landscape.

