# OpenReview forum: "The Shape of Attraction in UMAP: Exploring the Embedding Forces in Dimensionality Reduction"
_ICLR.cc/2026/Conference — Submitted to ICLR 2026_

### Official Review · Reviewer_N8BZ · 2025-10-27

**Soundness:** 3
**Presentation:** 1
**Contribution:** 1
**Rating:** 2
**Confidence:** 4

**Summary:**

UMAP is a widely-adopted neighbor embedding method, and understanding how its attractive and repulsive forces define the final embedding is core to improving reliability and usability in downstream tasks. This paper conducts numerical analysis on how these forces evolve under different circumstances and discusses the importance of learning-rate annealing and the asymmetry between attraction and repulsion. The authors provide code for both experiments and visualizations, supporting transparency and reproducibility.

**Strengths:**

- The writing is fluent, improving the readability of this paper..

- The code release is thorough, including experimental code and plotting scripts. This openness strengthens the paper’s scientific value and facilitates verification.

- The author has provided an extensive list of related works, covering most of the important progresses in this field.

**Weaknesses:**

- **Uncertain novelty relative to prior research.**
  - In section 4, the authors assert *a gap* in the literature regarding force dynamics, yet cited works (such as McInnes et al., 2018; Agrawal et al., 2021; Wang et al., 2021; Draganov & Dohn, 2023), as identified by the authors, already examine the evolution of attractive and repulsive updates, hence a distinction must be made to justify the novelty. The authors described their difference as *"Here, we treat the decompositions as independent functions that can take various forms"*, which is quite confusing to me and does not make a good point. In my opinion, the main difference here seems to be the integration of the learning rate into the discussion, so that we can discuss the actual update at each step. The significance of this point is unclear (see below), and it definitely needs more articulation.

  - Section 5.2: It should not be surprising to anyone in the dimensionality reduction field that the cluster formation is dominated by attraction and the repulsion makes the clusters more compact. To list a few examples just from the authors' own citations, similar ideas has been described in Narayan et. al 2020, Bohm et. al 2022, Wang et. al 2025. Besides, description regarding the design of LocalMAP seems to already discussed in Wang et. al 2025 Section 5.

  - Section 6: The authors built a connection between Neg-t-SNE and UMAP, which is a novel point, but the message is also unclear (see below). The authors' other acclaimed contribution, that the attractive and repulsive terms between different DR algorithms are interchangable, has been brought up before, see e.g. Wang et. al, 2021. As long as these terms are following the DR algorithm principles, they can form a good DR algorithm.

- **Lack of a clear overarching message.** While the experimental analysis is detailed, the narrative does not converge on a key takeaway.
  - Section 4 - 5.1: The authors observed that the existence of large updates in UMAP may flip the relative position of certain neighbors. While this observation can be interesting, the reader may wonder how this observation could translate into some real improvements over the dimensionality reduction quality. Comparing Fig 2 (b) and (d), I personally don't feel the authors' proposed change make any difference in the embedding quality, and this observation sort of invalidates the importance of Fig 2 (f-h) comparison. Similar comment could also be made to Fig 1, that for Neg-t-SNE and PaCMAP the default parameters seems to be achieving a good balance between the existing metrics. What **actionable** insight should an algorithm designer or user walk away with? For section 5.1, several conclusions restate design choices already justified in the original UMAP paper—for example, spectral-embedding-based initialization and learning-rate annealing—making the supposed improvements, if any, feel incremental rather than revelatory.

  - Section 6: While the authors built the connection between Neg-t-SNE and UMAP, it's unclear what is the major takeaway one could go from here. Shall we favor methods such as Neg-t-SNE and PaCMAP over the unstable UMAP? Is there statistical evidence to support such a claim? Many questions were left unanswered.


- **Scope feels narrow and presentation should be significantly reorganized**. The analysis focuses in the main text heavily on UMAP and the MNIST dataset. Framing within the broader ecosystem of neighbor-embedding methods could substantially elevate impact to the state required at top venues such as ICLR. It should be noted that, several analyses to other existing works and dataset reside in the appendix, but the main text does not have enough descriptions to these paragraphs. Ultimately, they are not only unlikely to shape reader perception, but also left in an unorganized state as they do not contribute to a coherent message. Integrating those results into the main narrative would considerably improve the overarching message and strengthen the paper’s contribution.

**Questions:**

My questions are mostly related to my confusion after reading this paper, which I have elaborated in the weaknesses section. Specifically:
- What would you like the reader to walk away after reading this paper? Compared to the default setting of UMAP, shall we increase $a$/$b$ or decrease it, to obtain a more faithful and interpretable result? Shall we increase the far-sightedness in the force gradient? Shall we favor other methods such as Neg-t-SNE/PaCMAP/LocalMAP? Does such choice improve the performance of the algorithm statistically?
- How would you like to separate your work against the existing ones in terms of novelty? Why is the proposed analysis more insightful, if the decomposition method discussed has already been adopted by many?

---

> ### Author Response · Authors · 2025-12-03
> **Response Regarding Novelty**
>
> The authors would like to thank the reviewer for finding the writing fluent and recognizing the strength of our code release and literature review. We provide a reply to the weaknesses and questions below:
>
> **Novelty relative to prior research:**
>
> **Section 4:** We agree that the significance of Section 4 was not articulated clearly and thank the reviewer for pointing this out. We provide a brief clarification below.
>
> While previous papers examine attractive and repulsive forces, to the best of our knowledge, they do not identify that, for practical choices of parameters, nominally attractive forces can in fact cause repulsion if not properly tuned (a consequence of our Proposition 4.1), nor that UMAP operates in a regime where this happens. The common narrative is that attractive forces cause attraction and repulsive forces cause repulsion; our analysis shows that this is not always the case and that practical algorithms must take this into account.
>
> More precisely, McInnes et al. (2018) derive the gradients needed to implement UMAP but do not analyze when attractive updates expand or contract distances. Agrawal et al. (2021) write the force in the form $f_a(\zeta)||y_i-y_j|| \times (y_i-y_j)/||y_i-y_j||$ (i.e., magnitude $\times$ unit vector) but do not study the contraction/expansion regime of $f_a(\zeta)$. Wang et al. (2021) use “rainbow plots” of force directions to visualize the loss, but do not show or discuss that for UMAP, at sufficiently small distances, the direction of the “attractive” update reverses. Their analysis primarily focuses on limiting values of the forces (e.g., limits of $f_a(\zeta)(y_i - y_j)$) rather than the intermediate region where this sign change occurs. Draganov \& Dohn (2023) use attractive and repulsive components to relate UMAP to PCA and LLE, but their focus is on connections to classical DR methods rather than on per-step distance dynamics.
>
> In contrast, we explicitly use the form $f_a(\zeta)(y_i - y_j)$, treat $f_a(\zeta)$ (and similarly $f_r(\zeta)$) as shape functions, and derive Proposition 4.1 (and its analogue for $f_r$), which characterizes precisely when an update contracts or expands an edge. This viewpoint, rather than the mere inclusion of the learning rate, is the main distinction of our analysis from the cited works.
>
> **About inclusion of the learning rate:**
> We respectfully disagree that the main difference in our analysis is simply the inclusion of the learning rate. Proposition 4.1 already holds for a fixed learning rate (e.g., $\lambda = 1$); its contribution is to characterize when a nominally “attractive” update actually contracts or expands an edge via the condition $-1 < \lambda f_a(\zeta) < 0$. In other words, the key step is to treat $f_a$ and $f_r$ as shape functions governing contraction/expansion, not just to append $\lambda$.
>
> We do not claim that prior work could not, in principle, include the learning rate, but to the best of our knowledge, they do not derive results analogous to Propositions 4.1–4.2. For example, Wang et al. (2021) use $\lambda = 1$ for the UMAP rainbow plot (a regime where $\zeta_{-1}\simeq1$ - large), yet the resulting visualization does not reveal or discuss the sign/arrow flip of the “attractive” updates at small distances. Our work makes this phenomenon explicit and links it to both UMAP’s sensitivity to the learning rate and the behavior of clusters in practice.

---

> > ### Author Response · Authors · 2025-12-03
> > **Response Regarding Novelty #2**
> >
> > **Section 5.2:**
> > We agree that the high-level intuition that attraction promotes cluster formation and repulsion encourages compactness is familiar in the dimensionality reduction literature (e.g., Narayan et al., 2020; Böhm et al., 2022; Wang et al., 2025). Our goal in Sec. 5.2 is therefore not to claim this qualitative statement as novel, but to show, within a single attraction–repulsion shape framework, how one can systematically control these effects in UMAP-type methods.
> >
> > Concretely, our experiments demonstrate two separate “knobs.” First, by modifying only the attraction shape while keeping the repulsion shape fixed, we can increase the number of visible clusters while preserving their compactness. Second, by modifying only the repulsion shape while keeping the attraction unchanged, we can increase inter-cluster distance while keeping the within-cluster structure essentially unchanged. To our knowledge, this explicit decoupling and empirical demonstration within the same base algorithm has not been shown in the cited works.
> >
> > Regarding LocalMAP, our description is consistent with Wang et al. (2025, Sec. 5); what we add is a reinterpretation of LocalMAP in terms of attraction and repulsion shapes. In particular, we define strong-attraction and strong-repulsion zones (Fig. 16(c,d)) in the same attraction-repulsion shape framework used throughout the paper, and connect these zones back to the consistency analysis in Sec. 5.1 (Fig. 4). This provides a common lens for comparing UMAP, LocalMAP, and our shape modifications.
> >
> > Finally, we note that a separation of the ten MNIST digit clusters similar to the main qualitative claim of LocalMAP can also be obtained by appropriately tuning the attraction and repulsion shapes within UMAP (e.g., Fig. 3(g)). We revised Sec. 5.2 to clarify that its contribution is to articulate and demonstrate this shared shape-based control mechanism, rather than to assert that the basic roles of attraction and repulsion themselves are new.
> >
> > **Section 6:**
> > We agree that the general idea that attractive and repulsive terms from different DR algorithms can be interchanged is not new; it has been discussed in, e.g., Wang et al. (2021), Agrawal et al. (2021), and is also implicit in the force-directed graph drawing literature. We do not intend to present this general interchangeability as a main contribution of our work, and we do not mention it.
> >
> > Rather, in Section 6 and Appendix J, we use this idea through the lens of attraction–repulsion shapes. In particular, we combine the finite-range repulsion shapes of Neg-t-SNE and PaCMAP with attraction shapes from UMAP, Neg-t-SNE, and PaCMAP that satisfy the contraction condition of Proposition 4.1, and then study how these shape choices affect clustering (silhouette score) and local structure (Trustworthiness). Our empirical finding is that keeping $f_a \in [-1,0]$ improves cluster separation, while restricting $f_a \in [-0.5,0]$ improves local structure by avoiding flips.
> >
> > We revised Section 6 to make it clearer that our goal is to provide a unified shape-based framework for understanding and systematically combining these terms, rather than to claim the abstract possibility of interchangeability itself as novel.

---

> > > ### Author Response · Authors · 2025-12-03
> > > **Overarching Message and Scope**
> > >
> > > **Overarching message:**
> > >
> > > **Section 4 - 5.1:**
> > > We thank the reviewer for these thoughtful comments and for prompting us to clarify the practical takeaways of Sec. 4–5.1.
> > >
> > > Regarding Fig. 2(b) vs. 2(d), we aimed to compare typical good outcomes rather than cherry-pick extreme failures. For each method, we therefore show the best four out of 100 runs in terms of Procrustes distance to the PCA-initialized reference. Fig. 2(f–h) is designed to address exactly the reviewer’s concern about overall quality: the heatmaps summarize the full distribution of Procrustes distances over 100 runs, and show that the long-range attraction substantially reduces variability and brings random-initialization results closer to the PCA-initialized baseline across a wide range of settings.
> > >
> > > The purpose of Fig. 1 is to connect Proposition 4.1 to cluster behavior. Sec. 5.1 then uses this insight to provide actionable guidance for algorithm designers and users:
> > > (i) Choose learning rates and annealing schedules so that, for most neighbor distances, $\lambda f_a(\zeta)$ lies in the $(-1,0)$ band.
> > > (ii) Adding a controlled long-range attraction in early epochs can substantially reduce run-to-run variability without degrading standard quality metrics when initialized randomly.
> > >
> > > We agree that, for Neg-t-SNE and PaCMAP, the default parameters already strike a reasonable balance between metrics. This is expected as they satisfy proposition 4.1 for $\lambda\in[0,1]$. Our goal in Sec. 5.1 is therefore not to claim dramatic visual changes for these methods, but to show that their empirical robustness can be interpreted and tuned within the same shape-based framework: their default settings place most edges in a favorable contraction regime, which explains why additional modifications yield more incremental improvements there than for UMAP.
> > >
> > > **Section 6:**
> > > We thank the reviewer for this question and for prompting us to clarify the role of Section 6.
> > >
> > > Our intention in Section 6 is not to argue that one should simply favor Neg-t-SNE or PaCMAP over UMAP, nor to claim a universal ordering between these methods. Rather, the goal is to show that, once expressed in the attraction–repulsion shape framework, we see two ways UMAP is different from NEG-t-SNE: 1) NEG-t-SNE follows Prop 4.1, 2) It has finite repulsion. This connection helps explain why Neg-t-SNE (and PaCMAP) tend to be more stable, and suggests that some of these advantages can be transferred to UMAP by adjusting its shapes (as we do in Sec. 5.2 and Appendix J), rather than abandoning UMAP altogether.
> > >
> > > Regarding statistical evidence, Section 6 is deliberately exploratory and qualitative: it focuses on illustrating the structural relationship between the objectives and the resulting force shapes, and uses a small number of examples to make this connection concrete. We are not recommending one algorithm over the other.
> > >
> > > **Scope:**
> > >
> > > Regarding scope, we agree that the current framing in the main text emphasizes UMAP and the MNIST dataset, and we see how this can give the impression of a narrow focus. We intended to use UMAP and MNIST as a running example to keep the exposition concrete, while demonstrating broader applicability through additional methods and datasets in the appendix. In particular, Appendix~C–J already analyzes Neg-t-SNE, PaCMAP, TriMAP, LocalMAP, t-SNE, SNE, and MDS within the same attraction–repulsion shape perspective, and applies our insights to multiple datasets beyond MNIST (e.g., other image datasets, and single-cell data). We will revise Sections 4–6 to make this broader scope explicit and to better highlight that our propositions and shape-based analysis are method-agnostic and not restricted to UMAP.
> > >
> > > We believe these changes will address the concerns of the reviewer.

---

> > > > ### Author Response · Authors · 2025-12-03
> > > > **Answer to Questions**
> > > >
> > > > **Question Set 1:**
> > > > We appreciate this question. It indeed goes to the heart of what we would like readers to take away from the paper. Broadly, there are two messages:
> > > >
> > > > (i) Conceptual: Neighbor-embedding methods such as UMAP, Neg-t-SNE, PaCMAP, and LocalMAP can all be viewed through a common lens of attraction and repulsion shapes.
> > > >
> > > > (ii) Practical: This shape view yields concrete, but problem-dependent, guidelines for choosing parameters and variants, rather than a single universal “best” setting.
> > > >
> > > > **On UMAP and $(a,b)$:** The study regarding a/b demonstrates how changing only attraction and only repulsion shapes can influence the cluster property. If a practitioner wants more clusters with the same repulsion profile, the practitioner can change only $b$ in the attraction profile. Alternatively, if the practitioner wants the same clustering properties but wants to increase inter-cluster distance (to show better separation), the practitioner may change $b$ in the repulsion shape alone. We do not give any recommendation for $a$.
> > > >
> > > > On farsightedness: If randomly initialized and one wants to achieve an embedding that is closer to the standard initialization, increasing farsightedness is recommended.
> > > >
> > > > On choosing between UMAP, Neg-t-SNE, PaCMAP, and LocalMAP: We do not claim that one should universally favor Neg-t-SNE, PaCMAP, or LocalMAP over UMAP. Our paper is not about proving which algorithm is best, but rather what choices affect these algorithms and exploring this under our attraction-repulsion shape formalism.
> > > >
> > > > In summary, we would like readers to walk away with (a) a unified, shape-based way of thinking about neighbor embeddings, and (b) practical “knobs”: learning rate and curvature of $f_a$ and $f_r$—that can be adjusted to trade off stability, cluster separation, and local structure. We will revise the conclusion to state these takeaways explicitly and to better connect the parameter choices in Sec. 5–6 to these concrete recommendations.
> > > >
> > > > **Question Set 2:**
> > > > As discussed in our earlier responses and in the paper, prior works use a similar force decomposition but in a different context: they do not characterize when “attractive’’ or “repulsive’’ updates actually contract or expand points. Our analysis in Sec. 4 treats the decomposed terms as shape functions and, via Propositions 4.1 and 4.2, provides explicit conditions on $f_a(\zeta)$ and $f_r(\zeta)$ that govern contraction and expansion. We believe this dynamical characterization, and the design insights it yields in Sec. 5–6, is the main way in which our analysis is more insightful compared to the previous paper.

---

### Official Review · Reviewer_PzRp · 2025-10-29

**Soundness:** 3
**Presentation:** 3
**Contribution:** 3
**Rating:** 6
**Confidence:** 4

**Summary:**

This paper introduces a theoretical formulation of UMAP’s learning process via a pair of mathematical properties: attraction shape and repulsion shape. With these properties, the authors manage to interpret the general mechanism of how UMAP’s loss function works in two forces under different conditions. The authors further show the practical implications of attraction and repulsion shapes by showing how they can interpret and improve the consistency and cluster formation, and how they explain the connection between Neg-t-SNE and parametric UMAP.

**Strengths:**

- Provide a careful theoretical analysis with derivations of several key properties.
- Offer insights that help explain behaviors such as initialization effects and cluster formation.
- Indicate practical implications for DR design, e.g., encouraging consistency.

**Weaknesses:**

- Ad hoc shape tuning. In the experiments of consistency improvements (Sec. 5.1) and cluster formation (Sec 5.2), the attraction/repulsion curve is adjusted ad hoc. While this lowers Procrustes distance, potential trade-offs with broader DR quality (e.g., Trustworthiness, Continuity) are not evaluated.
- Insufficient parameter analysis. The attractive/repulsive forms hinge on parameters a and b, yet the paper shows only scattered results for fixed choices. A systematic sensitivity study—visualizing how a and b affect the shapes and offering selection guidance—would clarify their roles.
- Limited discussion of alternative consistency methods. Related approaches, such as aligned-UMAP, are not adequately discussed. A comparison and a clear statement of the advantages and costs of the shape-guided method would strengthen the case.
- Poor readability and organization. The writing is dense, with many results and observations. Reorganizing the analysis with clearer signposting and bolded subheadings that highlight key takeaways would improve accessibility.

**Questions:**

(1) The core contribution of this paper is the detailed analysis of UMAP’s gradient update based on the attraction & repulsion shape coefficients. This can be viewed as a stepwise analysis of the local updates. But I wonder whether these shapes can shed light on UMAP's overall convergence property, especially across different settings of a and b? Do the oscillations the authors observe in some conditions cause unstable optimization?
(2) Beyond the cluster formation, the interplay between the attraction and repulsion forces also impacts the preservation of global and local structure. Can the attraction/repulsion shapes help explain some phenomena about this?

---

> ### Author Response · Authors · 2025-12-03
> **Response to Reviewer PzRp**
>
> The authors would like to thank the reviewer for the positive tone. We are glad that the reviewer identified the theoretical analysis, discovered insights, and provided practical implications as strengths.
>
> We clarify the weakness below:
>
> **Shape tuning:** In the revised manuscript, we have added Table 1 to show the potential trade-off with broader DR quality. Overall, we see that the run-to-run consistency improves a lot using the method in section 5.1. And with our modified/composite shapes, the representation becomes closer to that of the PCA/Spectral initialized one, suggesting this initialization is not just ad-hoc but likely an ideal one.
>
>
> **Parameter analysis:** We change the values of $a$ and $b$ when necessary (to show independent control of attraction and repulsion in Section 5.2, and to compare to NEG-t-SNE in Section 6). Besides, a systematic sensitivity study on the MNIST algorithm may not be transferable to other datasets of different sizes and dimensionality.
>
> **Alternative consistency methods:** Aligned-UMAP is a method for aligning multiple datasets with shared components in them. This is out of scope for our paper.
>
> **Organization:** We thank the reviewer for raising this concern. In the revised manuscript, we have included additional text to clarify the takeaways from each section.
>
>
> **Answer to Questions:**
>
> **Question 1:** We agree that our analysis primarily focuses on local behavior with global implications.
>
> We can connect $a$ and $b$ to the value of $\zeta_{-1}$ from attraction. A lower $\zeta_{-1}$ causes better convergence. On the other hand, curvature of the shape profile is of interest: if it is too steep and reaches zero quickly, the convergence becomes slow/unstable.
>
> The repulsion behavior is more guided by random sampling and thus, having a fail-safe mechanism or protection against large repulsive forces is desirable through some numerical stability parameter or finite repulsion shapes (e.g., NEG-t-SNE, PaCMAP). The shape should also decrease quickly so that the points that are already far apart do not get unnecessarily far away. In section 5.2, we experimented by adding a small value (order of $1e-4$). A value on the order of $1e-3$ or higher causes instability in convergence (example: https://bashify.io/i/rmldpw_epsilon ).
>
> **Question 2:** We thank the reviewer for the question. The global structure is controlled by the tail behavior of the attraction shape. Results in Section 5.1 show that, to manifest a consistent global structure from the forces, we need to add far-sightedness into the attraction shape (for random initialization). Repulsion shape, on the other hand, controls the global inter-cluster distance (Section 5.2).

---

### Official Review · Reviewer_AykA · 2025-10-31

**Soundness:** 3
**Presentation:** 4
**Contribution:** 3
**Rating:** 8
**Confidence:** 4

**Summary:**

The paper provides an in-depth analysis of the attractive and repulsive forces present in the optimization step of UMAP. The authors first process and simplify the update functions of the point coordinates in the projection space, introducing two functions, $f_a$ and $f_r$ which control contraction and repulsion. Using this tool, they go on and generate the following insights:

- The first function, $f_a$, which is often associated with attraction, can actually act repulsively by flipping points and moving them further apart. $f_a$ and $f_r$ are also plotted for several methods and different learning rates. Lowering the learning rate lowers the distance below which $f_a$ becomes repulsive. Methods where $f_a$ is only attractive by design seem to be more robust to changes of the learning rate.
- The consistency of the projection under random initialization is studied using the Procrustes distance as a measure of projection similarity. It is show that by increasing the attractive force, the projections become more consistent presumably because the force between distant points is large.
- The formation and compactness of clusters is studied. Evidence is provided that the former is caused by attraction while the latter by repulsion.
- Different methods are compared in light of the approach of using $f_a$ and $f_r$. It is shown that Neg-t-SNE are the same in terms of their loss terms.

**Strengths:**

- Very clean exposition and nice focus on the single topic of the role of attraction/repulsion in dimensionality reduction algorithms.
- Though a lot the ingredients of the paper already appeared in previous work (e.g. empirical attraction/repulsion function graphs), the way they are used and put together creates a novel view of the topic.
- The insights can help further optimize existing methods and provide a solid understanding on how new methods can setup their projection optimization.
- Follows the nice practice of including plenty of projections and plots for different datasets in the supplementary. This increases credibility and also makes the paper a nice reference.

**Weaknesses:**

- Only three datasets where used to support the different arguments, two of them quite similar in nature, MNIST and FMNIST. Though it is enough for visualization and demonstration purposes, the evidence feels a bit more qualitative than quantitative. Usually papers in the domain use almost 10 datasets covering different domains. To give an example of how this could have strengthened the evidence, in the study of cluster formation one could have correlated the attraction shape with the Silhouette score among datasets to support the connection of repulsion with cluster formation. One could also consider such ablations on synthetic datasets.

**Questions:**

- Around line 257 I think there is a typo, you use $\zeta^{-1}$ instead of $\zeta_{-1}$

---

> ### Author Response · Authors · 2025-12-03
> **Response to Reviewer AykA**
>
> We thank the reviewer for the very positive and thoughtful assessment of our work, and for highlighting both the clarity of the exposition and the potential impact of the attraction/repulsion–shape perspective. We respond to the specific concerns below.
>
> **Weakness:**
> We agree that we focused on only three datasets (MNIST, FMNIST, and Single Cell analysis). In the revised manuscript, we have added two additional datasets: another transcriptome data (Shekhar et al, 2016) and the 20Newsgroup (20NG) dataset. 20NG does not have any clusters. Table 1 (newly added) shows that with our modifications to the attraction shape, we can increase the silhouette score (for random initialization).
>
> **Typo:** Thanks for detecting the typo. We have corrected it.

---

### Official Review · Reviewer_3K6U · 2025-10-31

**Soundness:** 2
**Presentation:** 1
**Contribution:** 2
**Rating:** 2
**Confidence:** 5

**Summary:**

This paper discusses various aspects of the attractive and repulsive forces in UMAP and related visualization methods. It uncovers that for nearby points attraction might even increase their distance, explaining why learning rate annealing is crucial for UMAP. The authors also propose adding a long-range attractive force during the first optimization epochs to achieve a more consistent global layout, even from random initializations. Finally, they explore using different similarity kernels for attraction and repulsion in UMAP and compare UMAP's forces with other visualizations methods.

**Strengths:**

- S1: The finding about the expansive behavior of the attraction at small scales is interesting, novel and gives relevant pointers for the choice of the learning rate in UMAP.
- S2: The long-range attraction is an interesting contribution.
- S3: The extensive appendix extends some of the analysis to many other neighbor embeddings and is a useful resource.
- S4: Code is submitted, facilitating reproducibility.

**Weaknesses:**

**Major**

The paper presents many aspects about forces in neighbor embedding methods. While some of them are interesting, many feel underexplored and some are not novel. My suggestion is to expand the sections on expansive and long-range attraction in favor of the other sections. While this would improve the paper, I do not think that the novelty of the paper is sufficient for ICLR.

*W1: Presentation and exploration of UMAP's expansive attraction*:
While the expansive nature of UMAP's attraction is interesting, the phenomenon should be better explained and explored. The following would help the explanation in my mind: The expansion is a classical case of an overshooting gradient descent. The update size decreases more slowly than the distance, so that the update eventually overshoots. This happen only for $b<1$.
Figure 1 a takes up a lot of space, but conveys very little information, I would omit it, or at least omit the repulsive part and decrease the size for the attractive part.
Including some figures of UMAP with non-annealed learning rate (maybe even all four corner cases $\lambda_a, \lambda_r \in \\{0.1, 1\\}$) in the main paper and explaining them in more detail would help. Why can we deduce from this experiment that *only* the attraction necessitates annealing (the setting $\lambda_a = 0.1, \lambda_r=1.0$ still produces a "fuzzy" embedding). Based on these four plots, one could also explain the "failure modes" in Fig 1 h (top left and bottom right corner) qualitatively. It may also be useful to state that the distance $\zeta_{-1}$ around which the attraction would oscillate without annealing is large relative to typical UMAP plot diameters. Finally, it would help to state that by default, UMAP starts with $\lambda = 1$ and anneals this learning rate linearly to $0$ over training. Perhaps it might even be nice to show intermediate training stages, when clusters and their relative positions have already formed, but still appear fuzzy as $\lambda$ is still too large.

Moreover, the phenomenon should be explored more deeply: One could, for instance, let a single pair of points evolve according to UMAP's attraction from various starting distances and plot how its distance evolves (without learning rate annealing). I assume it would eventually oscillate around 1. Similarly, in a UMAP plot without annealed learning rate one could measure the average distance between the embeddings of k-nearest neighbors and check if it is close to 1. Or compare this mean neighbor distance between the annealed and non-annealed case.

*W2: Discussion and exploration of long-range attraction*
The proposed fix of adding $||y_i - y_j||^3$ to the attractive loss is quite similar to SNE, which uses attraction $|| y_i - y_j||^2$. It would be good to discuss this relation and perhaps even compare quantitatively. A small point is that in line 299 it is not $f_a^U$ going to zero that is problematic, but what is given in brackets ($|f_a^U| \cdot \zeta$ going to zero). But this is not equivalent, so "i.e." is not the correct choice here.

In general, this chapter should be made more quantitative. While the authors compute Silhouette scores, Trustworthiness, and Procrustes distances, their comparison would be much eased by putting mean values in a Table. Moreover, the relative positioning of clusters might be also captured well by metrics like Spearman correlation between distances (both relative to the high-dimensional and among different UMAP embeddings). A key questions is, of course, how the long-range attraction compares to informative initializations with PCA or LE. So, a results table should also contain metrics for PCA/LE-initialized embeddings. Are they not simply much more stable than UMAP with long-range attraction starting from random noise? Finally, the authors include experiments on other datasets in the appendix, but a tabular summary of these experiments in the main paper would also strengthen the empirical findings.

*W3: Cluster formation chapter is vague and less novel*
Overall, the novelty of this chapter seems limited since, e.g., Agrawal et al. (2021) also mix and match different attractive and repulsive forces and Kobak et al. (2019) (which the paper cites in this section) already discuss heavy-tailed neighbor embeddings, including the effect of $b$ on UMAP embeddings. While the present paper refines their analysis to heavy-tailed attraction being more important, claims such as "to resolve the mystery of cluster formation" are way too strong given the limited novelty. In addition, the paper does not investigate if the additional clusters found with heavy-tailed attraction are meaningful or even undesirable fragmentation. Finally, in line 371 the paper states that changing $a$ affects both repulsion and attraction. But a different $a$ really just rescales the final embedding without changing its appearance, since

$$1 / (1+ad\_{ij}^{2b}) = 1 / (1+(a^{1/(2b)}d\_{ij})^{2b}) = 1 /(1+\tilde{d}\_{ij}^{2b})$$

for the rescaled distance $\tilde{d}_{ij} = a^{1/(2b)}d\_{ij}$.

*W4: Discussion of Neg-t-SNE*
Relating Neg-t-SNE to UMAP in the context of expansive attraction is good and interesting. I particularly appreciate broadening its explanation for why learning rate annealing is required for UMAP to the attractive forces. However, I would strongly suggest to do so in the chapter on the expansive behavior of UMAP's attraction rather than two chapters later. The other statement in the Neg-t-SNE chapter, Prop. 6.2 on Neg-t-SNE's relation to parametric UMAP, is not novel, but can already be found in appendix E of Damrich et al. (2023).


**Minor**
*W5: Notation* The notation does not get introduced in many places. E.g. at the start of sec 3: $f, h, d, d_{ij}$ are not defined and in section 5 I do not understand why the notation changes from $f_a$ and $f_r$ to $f_a^U$ and $f_r^U$.

*W6: Related work* I would recommend not to include the spectral methods Laplacian Eigenmaps and Locally Linear Embedding in the middle of the paragraph on neighbor embeddings in the related work.

*W7: Flowery language* The language is in some place a bit to flamboyant for a research paper. E.g. the first sentence uses both "era" and "deluge", and the discussion section "demystifying".

*W8: Font size in many figures too small*



**Typos**

- line 235: approaches $-\infty$ not $\infty$

- line 484: Wrong citation Damrich & Hamprecht (2021) --> Damrich et al. (2023)

- line 1059 $(_a)$?

- Eq 86 / 87 Switch in notation between $w_{i \mid j}$ and $p_{i \mid j}$

- Abstract: forces among high-dim data point -->  low-dim data points

- Equality sign missing after $\zeta_{-1}$ in axis-label of figure 1d.

**Questions:**

- Q1: line 246: "only" UMAP and t-SNE deal with some issue. Only methods among which group?

- Q2: line 252: "only UMAP deals with large [repulsive] values for small distances" Is this a good thing, because non-neighbors get pushed far apart, or a bad thing because it leads to numerical instability and requires gradient clipping?

- Q3: line 294: What is "scale-invariant structure"? Was perhaps "initialization-invariant" structure meant?

- Q4: line 1060: Why does initializing $\lambda$ at $0.5$ satisfies Prop 4.1? It is still the case that $\zeta_{-1} \approx 0.5 > 0$ according to Fig 1d.

---

> ### Author Response · Authors · 2025-12-03
> **Response**
>
> We thank the reviewer for the detailed reading and review of the paper. We find it disheartening that the reviewer didn't find the insight presented in the paper on par. We give clarifications below:
>
> **Presentation and exploration of UMAP's expansive attraction:**
>
> This effect is not related to $b<1$ alone, and it can manifest in $b>1$ as well.  However, for $b>1$, the values are finite, and thus, we can set a $\lambda$ such that Proposition 4.1 is satisfied throughout. (image link: https://bashify.io/i/viDbsG )
>
> However, we do not discuss this in the paper as $b\leq1$ is the most common.
>
> **Distances among k-NN graph after embedding:**
>
> We ran the experiment for three datasets. Here are the results:
>
> | Dataset          | With Annealing | Without Annealing |
> |------------------|----------------|-------------------|
> | MNIST            | 0.56±1.70      | 1.71±1.80         |
> | FMNIST           | 0.39±0.94      | 1.52±1.24         |
> | Transcriptomes   | 0.48±1.16      | 1.67±1.54         |
>
> This is roughly an increase of ~1.20 from the annealed version.
>
> **Discussion and exploration of long-range attraction:**
>
> We fixed the point in Line 299. Now it reads: "Known as near-sightedness, this phenomenon is evident in the attraction shape, where $|f_a^U|$ diminishes towards zero as the distance increases (since $|f_a^U|=o(1/\zeta)$, and thus, $\lim_{\zeta\to\infty} |f_a^U(\zeta)|\zeta=0$)."
>
> To make the section more quantitative, we have added Table 1, which includes the embedding quality metrics (Trustworthiness, Silhouette Score, and Spearman) as well as run-to-run consistency metrics (Spearman and Procrustes distance) for 5 different datasets.
>
> In this section, we specifically focused on random initialization and compared it to a standard initialization. We do not claim this is better than PCA. Rather, Figs. 2 f-h show that, with long-range attractions, the algorithms have a higher propensity to be similar to a PCA initialized one (as per our knowledge, in most practical cases, PCA and Spectral initialization produce similar embedding).
>
> Regarding SNE: while we understand what the reviewer is refereeing to, we would like to disagree that ours is similar to SNE (We have discussed SNE in the supplementary G.5). To our understanding, SNE does not add $||y_i-y_j||^2$ explicitly, rather it comes from the gradients ($-2||y_i-y_j||^2$, with $f_a^{SNE}=-2$).
>
> **Cluster formation chapter:**
>
> The goal of this chapter is to show how attraction and repulsion shapes can be individually controlled to achieve different levels of clustering and inter-cluster distance. This is not just a mix-and-match picked up from another paper. We have cited Kobak et al, 2021. We show the effect of $b$ in Figs. f,g,h to show how b affects both clustering and inter-cluster distance; relation to Kobak et al, 2021 ends here. Then we show how we can isolate the repulsion shape alone and modify it to just influence the inter-cluster distance. Then, we show in Fig. 4 how we can alter clusters by changing attraction alone, and how we can alter inter-cluster distance by changing attraction alone. For convenience, we chose $b$ as the parameter to change; however, this experiment is not related to Agrawal et al, 2021, or Kobak et al, 2021.
>
> **Regarding Varying $a$:**
>
> We appreciate reviewers' derivation regarding varying $a$: i.e., it is a scaling parameter. We mentioned that, "decreasing $a$ increases repulsion, but it decreases attraction at a faster rate (causing a worse case of near-sightedness)." We can see this in this figure: https://bashify.io/i/lbE4kw_vary_a.
> Initially, we focused on decreasing $a$ to increase the repulsion. But for the discussion, we plotted a wide range here (0.1 to 9.0) and showed a wide spectrum.
>
> The corresponding MNIST embeddings when the learning rate is annealed are: https://bashify.io/i/BQf5dg_annealed.
>
> For small $a$, the clusters start overlapping, and cluster formation is not optimal (mentioned in our paper). Thus, we decided not to show any result regarding $a$. For large $a$, however, the cluster formation is okay as learning rate annealing takes over and anneals $\zeta_{1}$ to zero.
>
> For completeness, let's look at the case when the learning rate is constant: https://bashify.io/i/SDVoYi_constant.
>
> In this case, we see fuzzy clusters as $a$ increases. It is because, $\zeta_{-1}$ increases along with $a$.

---

> > ### Author Response · Authors · 2025-12-03
> > **Response #2**
> >
> > **Discussion of Neg-t-SNE:**
> > We agree that we could add the NEG-t-SNE section 5. However, we wanted this section to focus mostly on UMAP alone and not involve formulas and excessive details of other methods, and rely on the attraction and repulsion propositions.
> >
> > We also want to have NEG-t-SNE in the main text, because to us, this was a very exciting observation. Thus, we have kept it as a separate section.
> >
> > Our Prop. 6.1 was regarding fixed $a=1$ and $b=1$, and not just equivalence of the methods. But we do see that, in heart, they are the same. To fix it, we have cited Damrich et al., 2023.
> >
> > **Minor:**
> >
> > **W5:** We have added the definitions. We changed $f_a$ to $f_a^U$ to specifically describe UMAP. For other attraction shapes we used different notations. Such as, for NEG-t-SNE, we used $f_a^N$.
> >
> > **W6:** We appreciate the suggestion to avoid placing Laplacian Eigenmaps and Locally Linear Embedding inside the main paragraph on modern neighbor embedding methods. We intended to highlight their use of the same k-NN graph construction that later underpins iterative algorithms such as LargeVis, UMAP, PaCMAP, and NEG-t-SNE.
> >
> > **W7:** We have corrected the flowery language and altered the sentences:
> >
> > From: The current era is characterized by a deluge of high-dimensional data.
> > To: Modern applications routinely generate high-dimensional data.
> >
> > From: Here, we demystified much of the dynamics underlying cluster formation.
> > To: Here, we have analyzed the dynamics underlying cluster formation.
> >
> > **W8:** We apologize regarding the small font. In the revision, we will increase the size of fonts.
> >
> > **Typos:** Thanks for detecting the typos. We have corrected them.
> >
> > **Questions:**
> >
> > **Q1:** "only UMAP and t-SNE deal with the issue" - the ones in Fig. 1(f)
> >
> > **Q2:** "only UMAP deals with large [repulsive] values for small distances" - at this point in the paper, we don't say whether it is good or bad. Later, in Section 6, we do mention that it causes instability that NEG-t-SNE solves and extend the story to attraction shape as well. In Appendix J, we exclusively focus on Finite repulsion shapes. We do not mention gradient clipping in the paper, but we agree that this is a feature that is required in UMAP.
> >
> > **Q3:** We wanted to say, structures that are invariant under different initialization. We have fixed it.
> >
> > **Q4:** This is because we are discussing the case when $a=1$ and $b=1$. For this case, UMAP's attracting shape $f_a^U(0)=-2$. Thus, by halving the attraction, the shape satisfies Prop 4.1.

---

### Meta-Review · Area_Chair_5XZk · 2025-12-30

**Summary:**

The primary concerns not addressed are the limited novelty and significance of the work, as highlighted by Reviewers 3K6U and N8BZ. After careful study of the paper, I concur with these reviewers. Indeed, the novelty of the analysis and results is limited compared to prior work. Moreover, the paper fails to provide concrete guidance on applying UMAP in practical scenarios. Consequently, the overall quality falls below the acceptance threshold for ICLR, and I cannot recommend acceptance.

**Reviewer Concerns:**

* **Reviewer 3K6U**'s major concern is about the novelty of the paper. Besides, the reviewer pointed out three weaknesses: 1) The presentation and exploration of UMAP's expansive attraction are insufficient; 2) More discussion and exploration of long-range attraction should be provided; 3) The cluster formation chapter is vague and less novel.
   * It seems that the rebuttal and revision have addressed Weaknesses 1 and 2.

* **Reviewer AykA**'s concern is that only three datasets were used to support the arguments.
    * The revision has addressed this concern by adding two more datasets in the experiments.

* **Reviewer PzRp** has the following concerns: 1) Broader DR quality measures (e.g., Trustworthiness, Continuity) haven't been used; 2) Insufficient parameter analysis; 3) Limited discussion of alternative consistency methods; 4) Poor readability and organization.
    * I think the rebuttal addressed concerns 2, 3, and 4, while concern 1 persists.

* **Reviewer N8BZ**'s major concerns are: 1) Uncertain novelty relative to prior research; 2) Lack of a clear overarching message; 3) The scope of the paper is a little narrow, and the presentation should be improved.
    * The rebuttal, I think, hasn't sufficiently addressed these concerns. For instance, on page 10 of the revised paper, the authors provided a few takeaways, but most of which are still vague.

**Reviewer Scores:**

I think Reviewer 3K6U (2) and Reviewer N8BZ (2) would raise their ratings to 4, at most, if they had been able to participate fully in the discussion. The other reviewers would maintain the positive assessments.

---

### Decision · Program_Chairs · 2026-01-26

Reject